# Octyl itaconate enhances VSVΔ51 oncolytic virotherapy by multitarget inhibition of antiviral and inflammatory pathways

The presence of heterogeneity in responses to oncolytic virotherapy poses a barrier to clinical effectiveness, as resistance to this treatment can occur through the inhibition of viral spread within the tumor, potentially leading to treatment failures. Here we show that 4-octyl itaconate (4-OI), a chemical derivative of the Krebs cycle-derived metabolite itaconate, enhances oncolytic virotherapy with VSVΔ51 in various models including human and murine resistant cancer cell lines, three-dimensional (3D) patient-derived colon tumoroids and organotypic brain tumor slices. Furthermore, 4-OI in combination with VSVΔ51 improves therapeutic outcomes in a resistant murine colon tumor model. Mechanistically, we find that 4-OI suppresses antiviral immunity in cancer cells through the modification of cysteine residues in MAVS and IKKβ independently of the NRF2/KEAP1 axis. We propose that the combination of a metabolite-derived drug with an oncolytic virus agent can greatly improve anticancer therapeutic outcomes by direct interference with the type I IFN and NF-κB-mediated antiviral responses.

The use of oncolytic viruses (OVs) in cancer treatment is an emerging therapeutic approach that has demonstrated significant antitumor actions and safety in phases I–III clinical trials[1–3]. OVs are tumor-specific, self-amplifying therapeutics that selectively kill cancer cells by direct oncolysis, shutdown of tumor vasculature, and stimulation of immune responses against tumor antigens[4–7]. The therapeutic potential of OVs was highlighted by the FDA approval of the oncolytic recombinant herpes virus talimogene laherparepvec (T-VEC) as a therapy for the treatment of inoperable malignant melanoma[8]. More recently, Japan approved teserpaturev, a third-generation, triple-mutated oncolytic herpes simplex virus 1 for the treatment of patients with malignant glioma offering the first ever OV therapy option to cancers that are not controlled with currently available treatments[9].

An essential aspect in the development of these biologically active agents is safety, thus therapeutic viruses are genetically attenuated to decrease viral pathogenicity. One of the outcomes showcased by preclinical and clinical trials is the heterogeneity in the therapeutic response to oncolytic virotherapy where a fraction of the patients remain resistant to the effect of the OVs[10–12]. This constitutes a critical roadblock to the broader use of viruses as clinical therapeutic

bioactive agents in the clinic. The identification of strategies that potentiate OV replication specifically within malignant tissues without harming non-malignant cells could substantially improve oncolysis and the overall efficacy of these therapeutic viruses.

The combination of OVs with pharmacological agents that can suppress antiviral immune responses and bolster OV efficacy is an area of increasing interest that has shown promise in improving clinical outcomes[12]. Previous studies have reported the in vitro and in vivo synergistic effects of different classes of molecules, including histone deacetylase inhibitors, mTOR inhibitors, microtubule-destabilizing agents, as well as alkylating antineoplastic agents. Generally, these molecules dampen the type I IFN response and increase OV replication within resistant malignancies[13–19]. Our earlier work also reported the use of sulforaphane, an electrophilic aliphatic isothiocyanate, as an OV bolstering agent through the activation of the transcriptional regulator nuclear factor erythroid 2-related factor 2 (NRF2) and the suppression of antiviral immunity[20]. Likewise, Selman et al. also reported the unconventional application of the FDA-approved metabolite-derived drug dimethyl fumarate (DMF), which resulted in higher anticancer properties of OV candidates[21].

✉e-mail: olagnier@biomed.au.dk

In recent years, another Krebs cycle derivative, itaconate, gathered a lot of scientific attention[22–24]. Original work from Lampropoulo et al. identified that itaconate affected macrophage metabolism and drove anti-inflammatory action through the inhibition of succinate dehydrogenase (SDH)[24]. Other chemical derivatives of itaconate including 4-octyl itaconate (4-OI), have also recently emerged as negative regulators of macrophage inflammatory responses[22,23]. 4-OI was shown to act on the transcriptional regulator NRF2 via the alkylation of its repressor KEAP1 to suppress inflammation and limit type I IFN responses[23]. In line with these findings, our previous work also demonstrated that 4-OI suppressed STING expression via NRF2 engagement, therefore limiting its type I IFN signaling potential[25]. Recently, two studies uncovered mesaconate and citraconate as endogenous positional isomers of itaconate with anti-inflammatory potential, thus expanding on the possible arsenal of metabolite-derived drugs with anti-inflammatory action[26,27].

Swain et al. demonstrated distinct biological activities of unmodified itaconate and itaconate derivatives by examining their metabolic, electrophilic, and immunologic profiles[28]. For instance, it was shown that 4-OI had a stronger electrophilic potential in contrast to itaconate, which correlated with its potent anti-inflammatory and interferon suppressive capacities. Conversely, itaconate treatment further increased LPS-induced IFNβ secretion highlighting the differences in biological actions of these molecules[28]. In line with its strong electrophilic potential, additional 4-OI targets including NLRP3, JAK1, and STING were identified[29–33]. Unexpectedly, we also showed that 4-OI induced a cellular antiviral program that potently inhibited a broad range of viruses including SARS-CoV-2, Herpes Simplex virus 1 and 2 (HSV-1/2), Vaccinia Virus, and Zika Virus, and that without promoting an inflammatory state[34]. This antiviral action was also observed by two other groups, reporting that 4-OI reduced influenza A virus replication by directly targeting the nuclear export protein CRM1[35,36].

In summary, the itaconate derivative 4-OI demonstrates paradoxical antiviral yet broadly immunosuppressive effects. At the same time, we have explored the concept of complementing the oncolytic virus VSVΔ51 with 4-OI to potentiate its replication and oncolytic action in tumor cells. We report here the pro-viral effect of 4-OI in vitro in tumor cell lines and 3D patient-derived colon tumoroids, in vivo in animal models as well as ex vivo in human organotypic brain tumor slices. Mechanistically, we demonstrate that 4-OI alters the RIG-MAVS, the NF-κB and the type I IFN signaling pathways independently of NRF2 through the alkylation of MAVS, IKKβ, and JAK1 to potentiate oncolytic virotherapy.

## Results

### Octyl itaconate enhances tumor-specific VSVΔ51 replication and oncolysis in vitro

In recent years, itaconate and its derivative 4-OI have been reported to display some antiviral action against a broad range of DNA and RNA viruses[34,37]. However, little is known about the possible pro-viral action of itaconate and its derivatives in tumoral cells. We first set out to examine the impact of endogenous itaconate, its naturally occurring isomers citraconate and mesaconate and three chemical derivatives, 4-OI, 1-octyl citraconate (1-OC) and 4-octyl citraconate (4-OC) (structures displayed in Fig. 1a) on the infectivity of oncolytic vesicular stomatitis virus Δ51 (VSVΔ51) in the renal adenocarcinoma cell line 786-O. A pro-viral effect of 4-OI was detected in 786-O cells, where a pretreatment of the cells with the molecule yielded a 3.5-fold increase in VSVΔ51 infectivity (Fig. 1b). While 4-OC also had some slight enhancing action on VSVΔ51 infectivity, none of the other metabolites tested including itaconate and its natural isomers displayed any pro-viral action in 786-O cells (Fig. 1b). To validate the uptake of itaconate and 4-OI in 786-O cancer line, cells were treated with the respective compounds prior to infection with VSVΔ51. Liquid chromatography-mass spectrometry (LC-MS) was applied to measure intracellular levels of the different metabolites (Fig. S1a, b). Consistent with the previous report[28], we

detected significant intracellular levels of itaconate, as well as intracellular 4-OI in the respective treatment conditions, however, itaconate was not shown to be accumulated following 4-OI exposure (Fig. S1a, b). Since itaconate was demonstrated to inhibit SDH[24], we also measured the intracellular accumulation of succinate after itaconate or 4-OI treatment in the presence or absence of the virus. Interestingly, we observed that only itaconate and not 4-OI treatment resulted in succinate accumulation (Fig. S1c). Of note, VSVΔ51 infection did not alter the levels of itaconate in the cells but moderately enhanced the accumulation of succinate (Fig. S1c).

In the same resistant cancer cell line, 4-OI potentiated VSVΔ51 infectivity in a dose-dependent manner at concentrations between 12.5 and 75 μM in 786-O cells (Fig. 1c). A similar trend towards promoting VSVΔ51 infection was also shown by increased viral RNA content upon 4-OI treatment (Fig. 1d). More broadly, 4-OI bolstered virus replication from 5-fold to more than 100-fold at the different multiplicity of infection (MOI) in a panel of human and murine cancer cell lines (colon, breast, lung, pancreas, kidney, skin, brain) displaying variable susceptibility to VSVΔ51 (Fig. 1e–g). In contrast to cancer lines, 4-OI only negligibly increased VSVΔ51 infectivity in primary normal human umbilical vein endothelial cells (HUVECs) and primary fibroblasts from healthy donors (Fig. S2a). A side-by-side comparison with DMF, another metabolite-derived drug that promotes viral oncolysis[21], revealed a greater capacity of 4-OI to bolster VSVΔ51 infectivity in tumoral cells, especially at low micromolar concentrations (Fig. S2b). Another significant observation between the pro-viral action of 4-OI vs DMF is the relatively low impact of 4-OI on primary HUVECs and healthy fibroblasts when DMF was still strongly enhancing VSVΔ51 infection in normal non-transformed cells (Fig. S2c). To test the action of 4-OI on other oncolytic viruses, different combinations of virus and 4-OI treatments were applied to cancer cells, and viral growth was evaluated with various readouts. 4-OI treatment did not affect Sindbis virus and Reovirus replication (Fig. S3a, b) but reduced the proliferation of Measles virus and Vaccinia virus JX-594 (Fig. S3c, d) as previously reported in ref. 34.

We further assessed the oncolytic action of VSVΔ51 in combination with 4-OI. 786-O monolayers infected with VSVΔ51 solely or in the presence of 4-OI were stained with Hoechst dye and overlaid with methyl cellulose-containing medium (1%) to define the size of virus replication foci. 48 h after infection, cells were stained with crystal violet solution to validate the formed plaques (Fig. 1h). The addition of 4-OI led to 2.5-fold increase in the average plaque diameter of VSVΔ51 showed by fluorescence imaging at 24 h (Fig. 1i) and resulted in a massive monolayer disruption visualized by crystal violet staining at 48 h (Fig. 1h). A similar effect in tissue integrity alteration was observed with calcein green staining in 786-O cells infected with VSVΔ51 in the presence of 4-OI (Fig. 1j). To evaluate the cytotoxicity from the 4-OI + VSVΔ51 combination compared to VSVΔ51 challenge alone, immunofluorescence microscopy and flow cytometry analysis were conducted in CT26WT and 786-O cells (Fig. 1k–m). Increased signal for cleaved caspase 3 in CT26WT cells challenged with VSVΔ51 was detected only in 4-OI treated group (Fig. 1k). In addition, the VSVΔ51 + 4-OI combination significantly decreased CT26WT and 786-O viability at 30 h post-infection (Fig. 1l, m). The bolstering and pro-oncolytic effect of 4-OI on VSVΔ51 was transferable to bystander cells as the transfer of supernatant-containing virus from 4-OI-treated cells (Fig. S4a) enhanced viral infectivity (Fig. S4b, c) and cancer cell killing (Fig. S4d) in non-stimulated recipient CT26WT cells. Together, our data indicate that 4-OI can increase the replication and oncolytic potential of VSVΔ51 in both mouse and human cancer cell lines.

### 4-OI enhances VSVΔ51 infection ex vivo in murine tumor cores and improves its therapeutic outcomes in vivo

As we demonstrated a significant enhancement in VSVΔ51 replication with 4-OI in two murine colon (CT26WT) and sarcoma (76-9) cell lines

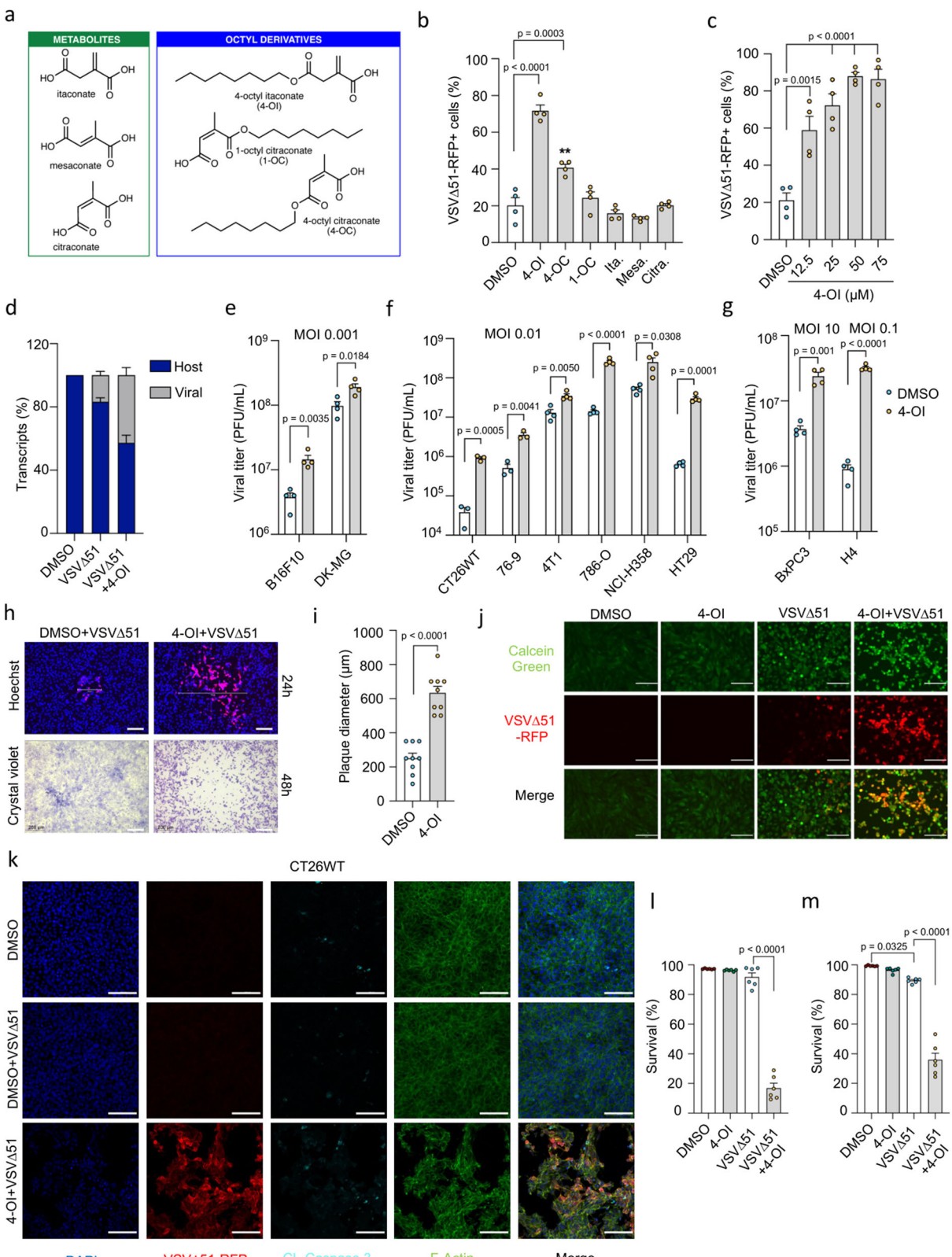

in vitro (Fig. 1f), we also assessed the treatment outcome ex vivo in mouse-derived tumor and normal tissues. To establish solid tumors, BALB/c and C57BL/6 mice were implanted subcutaneously with CT26WT or 76-9 cells, respectively. Tumor cores, as well as cores from healthy brains, lungs, spleens, and muscles, were subsequently extracted and infected with VSVΔ51-GFP in the presence or absence of 4-OI at various concentrations (Fig. 2a–e). 4-OI potently increased the

replication of the virus in CT26WT and 76-9 cores by roughly 10-fold and 3-fold respectively but did not affect the virus levels in healthy tissue cores (Fig. 2c–e). To expand on these findings, we assessed the 4-OI-mediated potentiation of VSVΔ51 under in vivo conditions. CT26WT tumors were established in BALB/c mice and received a single intratumoral (i.t.) injection of 4-OI 24 h prior to i.t. infection with luciferase-expressing VSVΔ51 (VSVΔ51-Luc). We found that 4-OI

**Fig. 1 | 4-OI promotes VSVΔ51 infection and oncolysis. a** Structures of itaconate, isomers and octyl-derivatives **b** 786-O cells pretreated with octyl-derivatives (80 μM) or metabolites (25 mM) for 24 h, then infected with VSVΔ51 (MOI of 0.01). Virus-infected cells quantified by flow cytometry at 17 h post-infection. **c** Flow cytometry analysis of virus-infected cells in 786-O cells treated with increasing 4-OI concentrations at 17 h post-infection. **d** Host *vs* viral RNA ratio in VSVΔ51-infected 786-O cells (MOI of 0.01) with or without 4-OI (75 μM). **e**–**g** Cancer lines pretreated with 4-OI (125 μM or 75 μM for 786-O cells) for 24 h, then infected with VSVΔ51 at varying MOIs. Viral titers determined from supernatants 24 h post-infection. **h** 786-O cells pretreated with 4-OI (75 μM) for 24 h and infected with VSVΔ51 (MOI of 0.0001), followed by plaque imaging. Scale bars, 200 μm. **i** Plaque diameters measured 24 h after infection. **j** 786-O cellular layer integrity assessed by Calcein green staining after treatment with 4-OI (75 μM) and VSVΔ51 infection (MOI 0.01) for 24 h. Scale bars, 100 μm. **k** CT26WT cells treated with 4-OI (125 μM) for 24 h post-infection with VSVΔ51 (MOI 0.01) for 48 h. Cleaved caspase 3 in cyan blue, nuclei in dark blue stained with DAPI, and actin filaments with phalloidin in green. Scale bars, 100 μm. **l, m** CT26WT and 786-O cells pretreated with 4-OI (125 μM) and (75 μM), respectively, for 24 h, then infected with VSVΔ51 at a MOI of 0.01. Percentage of viable cells determined by flow cytometry at 30 h post-infection. Data are means ± SEM of two independent experiments in duplicates in (**b, c, e**–**g**) (except for CT26WT and 76-9, from one experiment in triplicates); one experiment in triplicates for (**d**); one experiment in multiple replicates for (**l**); and two experiments in triplicates for (**l, m**). Images are from one experiment in (**h**), one representative experiment out of two in (**j**), and one out of two in (**k**). Statistical significance indicated by one-way ANOVA for (**b, c, l, m**); and two-tailed Student's *t*-test for (**e**–**g, i**). Source data provided in a Source Data file.

potentiated VSVΔ51-Luc luminescence by 2.5-fold within the tumors one day after virus administration (Fig. 2f, g). 48 h after infection, tumors were extracted, and the virus titer was quantified by plaque assay. A 6.6-fold increase in VSVΔ51-Luc titers was observed in tumors in situ (Fig. 2h).

We finally evaluated the therapeutic potential of combining 4-OI with VSVΔ51 in vivo. Following the establishment of CT26WT tumors grown on the flank of BALB/c animals, mice were injected i.t. with 4-OI for 24 h prior to infection i.t. with VSVΔ51. The treatment regimen was applied twice. Mice given the combined treatment were more successful in controlling tumor burden as tumor volumes were notably smaller compared to other treatment groups (Fig. 2i). The combination therapy also prolonged the survival of animals compared with VSVΔ51 as a monotherapy (*$p = 0.0351$ VSVΔ51 *vs* VSVΔ51 + 4-OI) (Fig. 2j). Notably, 7/8 (87.5%) animals receiving the VSVΔ51 + 4-OI treatment achieved complete remission whereas only 3/8 animals (37.5%) did in the VSVΔ51 treatment group (Fig. 2k). The cured CT26WT-bearing mice that had received the combination regimen subsequently controlled tumor growth (Fig. 2k) and became immune to rechallenge with the same cancer cells (Fig. 2l). These results demonstrate that 4-OI in combination with VSVΔ51 confers an improved therapeutic efficacy and provide a specific antitumor immunity compared with animals receiving either monotherapy.

## 4-OI potentiates oncolytic virotherapy in 3D patient-derived colon tumoroids and organotypic human brain tumor slices

Oncolytic virotherapies are usually examined in vitro in cancerous cell lines, in vivo in xenograft or syngeneic mouse models, or ex vivo in cancer patient's biopsy cores, however only a limited number of studies have attempted to evaluate the efficacy of OV therapies in pathologically relevant 3D tumoroid models that are becoming more clinically predictive and ethically favorable compared to murine in vivo animal models in cancer research[38,39]. Here, we set to investigate our combination therapy in a panel of colorectal tumor organoids (TO) derived from multiple patients and in their matching normal organoids (NO)[40] (Table S1). Virus-encoded RFP expression was increased in patient-derived colon cancer organoids pretreated with 4-OI in comparison to the untreated control, as visualized by confocal spinning disk microscopy in tumoroids stained with membrane permeable nuclear dye Hoechst (Fig. 3a). In line with our findings in normal primary cells and normal murine tissue cores, 4-OI did not promote virus spread in the matching normal colon organoids (NO P1) that remained refractory to the infection by VSVΔ51. The percentage of VSVΔ51-infected cells increased from 5.2% with VSVΔ51 alone to 62% in combination with 4-OI in the tumoroids (TO P1), while infectivity remained below 5% in both treatment groups for the normal organoids (NO P1) (Fig. 3b). A similar increase in viral RNA was also measured by qPCR from the combination of VSVΔ51 with 4-OI in the tumoroids (TO P1) (Fig. 3c). To further assess whether the combinatorial treatment of VSVΔ51 and 4-OI would result in a similar trend across a panel of different patient-derived colon tumoroids, we developed a reliable microscopy-based assay to quantify viral infection within each culture well.

To ensure a quantitative detection of VSVΔ51-RFP in tumoroids by microscopy, we coupled it with flow cytometry analysis of RFP-positive cells. For that, we chose two tumoroid lines that displayed drastically different susceptibility to VSVΔ51 (P1 and P12) (Fig. S5a–d). Using the microscopy-based approach, an infection time of 48 h was selected for further testing on a larger panel of patient-derived 3D tumoroids (Fig. S5e). 4-OI treatment significantly bolstered VSVΔ51 infectivity in 8 out of the 12 patient-derived lines tested (P1–P12), highlighting the existence of patient variability in response to the virus and/or to the sensitization by 4-OI (Fig. 3d–f). Quantification of viral RNA by qPCR in P5 and P6 confirmed the increased level of infection in response to 4-OI and validated the relatively low levels of virus infection in the matching normal organoids (Fig. S5f). Finally, to further assess the oncolytic potential of VSVΔ51 in the presence of 4-OI, some tumoroid lines were introduced with GFP/luciferase reporter that allowed us to measure tumoroids survival at 96 h after infection. Combined treatment with 4-OI resulted in a decrease of luciferase activity (survival) for the tested lines (Fig. 3g). The effect was especially impressive for two of the lines (P3 and P4) where the survival went below 5% after 4 days of virus infection and 4-OI treatment. Additionally, we assessed the gene expression of two apoptotic markers by qPCR (PUMA (*BBC3*) and NOXA (*PMAIP1*)) in NO *vs* TO for three different patients (P1, P5 and P6) (Fig. S5g, h). Consistently, the TO from the different patients treated with the combination 4-OI + VSVΔ51 displayed higher levels of apoptotic markers compared to NO. Together these results indicate the capacity of 4-OI to promote VSVΔ51 infection and tumor cell killing in multiple patient-derived 3D colon cancer organoids.

The use of patient-derived tumoroids as models for testing oncolytic virotherapies is faced with certain limitations including the lack of innervation and blood vessels, immune cells, or tumor stroma. To overcome these issues, we tested the virotherapy in combination with 4-OI on excised biopsies from patients with primary or metastatic brain cancer undergoing neurosurgery (Table S2). Tumor material was sliced and cultured as organotypic structures as described previously[41]. Slices were treated with 4-OI and further infected with VSVΔ51 (Fig. S6a). 4-OI increased the infectivity of VSVΔ51 in two out of the three patient samples tested (Fig. S6b, c). Overall, both primary and metastatic brain tumors were sensitized by 4-OI to VSVΔ51 infection, especially in material from melanoma brain metastasis.

## 4-OI promotes viral infection independently of NRF2 and KEAP1

4-OI is known to activate NRF2[23,25], and we, therefore, measured the increased stabilization of this transcription factor in 4-OI-stimulated 786-O tumoral cells, knockout (KO) or not for NRF2. As expected, 4-OI treatment of 786-O control cells led to an increase of NRF2 protein levels which was absent in NRF2 KO cells (Fig. 4a). Recent studies have shown that KEAP1 and other targets of 4-OI can be detected by using an alkynylated analog (4-OI-alk)[42]. To confirm the KEAP1 interaction with 4-OI in cancer cells, 786-O cells were treated with the 4-OI-alk probe for

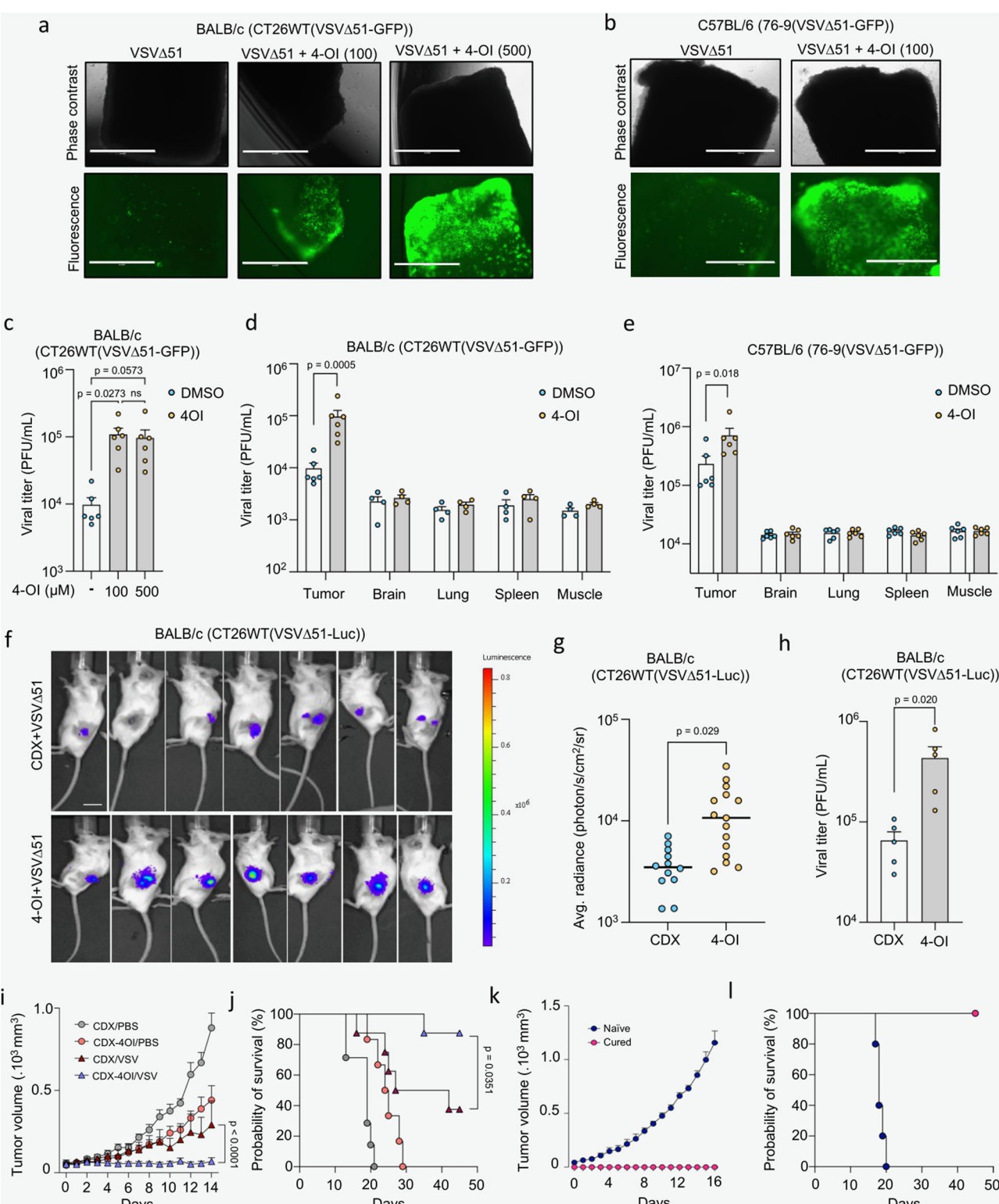

4 or 24 h, which demonstrated enrichment in KEAP1 upon CuAAC conjugation to biotin azide and subsequent streptavidin-based affinity pull-down (Fig. 4b). Of note, labeling of 4-OI-alk towards KEAP1 is decreased after 24 h which could be explained by the relatively short half-life of KEAP1 protein[43]. Finally, using bulk RNA sequencing to examine the NRF2-dependent gene expression profile in 4-OI-stimulated 786-O control and NRF2 KO cells, we observed an induction of NRF2-driven genes (such as *HMOX1, TXNRD1, NQO1, AKR1B10*) in control cells, which was absent in NRF2 KO cells (Fig. S7a).

Altogether, the data presented supports previous reports that 4-OI is a potent NRF2 inducer via the alkylation of KEAP1.

We next sought to assess the mechanism of 4-OI-mediated enhancement of VSVΔ51 infection by evaluating the biological activity of the combinatorial treatment in NRF2 KO cells. Surprisingly, the number of viral RNA transcripts did not differ comparing control *vs* NRF2 KO cells, hence 4-OI was still able to increase VSVΔ51 infection in the absence of NRF2 (Fig. 4c). This NRF2-independent increase in VSVΔ51 infectivity by 4-OI was confirmed using immunoblotting of VSV

**Fig. 2 | 4-OI enhances VSVΔ51 infection ex vivo in murine tumor cores and improves its efficacy in vivo. a–c** BALB/c-derived CT26WT tumor cores pretreated with 4-OI at different concentrations for 4 h before VSVΔ51-GFP challenge ($3 \times 10^4$ PFU). Representative fluorescence images (scale bars, 1000 μm) shown in (**a**) with viral titers determined from supernatants at 24 h post-infection in (**c**). C57BL/6-derived 76-9 tumor cores pretreated with 4-OI (100 μM) for 4 h before VSVΔ51-GFP challenge ($3 \times 10^4$ PFU). Representative fluorescence images (scale bars, 1000 μm) in (**b**). **d, e** CT26WT implanted subcutaneously in BALB/c (**d**) and 76-9 cells in C57BL/6 mice (**e**), Tumors explanted and cored with surrounding healthy tissues, pretreated with 4-OI (100 μM) or DMSO before VSVΔ51-GFP infection ($3 \times 10^4$ PFU). Viral titers were determined 48 h post-infection. **f–h** CT26WT tumor-bearing BALB/c mice intratumorally treated with vehicle or 4-OI (25 mg/kg/dose) for 24 h before VSVΔ51-luciferase challenge ($10^8$ PFU). Bioluminescent images taken and luminescence quantified 24 h post-infection (**f, g**), and viral titers determined at 48 h post-infection (**h**). **i, j** Tumor volume (**I**) and survival (**j**) monitored after intratumoral injection of 4-OI prior to VSVΔ51 challenge, treatment regimen was repeated twice ($n = 7$ in CDX-PBS group; $n = 6$ in CDX-4-OI/PBS group, $n = 8$ in CDX-VSVΔ51 and CDX-4-OI/VSVΔ51 groups). **k, l** Tumor volume (**k**) and survival (**l**) monitored after reimplantation of CT26WT cells in cured animals from CDX-4-OI/VSVΔ51 group from (**c**) and naïve mice. $n = 5$ animals per group. Data are depicted as means ± SEM in (**c–g, i, k**). Data points in (**c–e**) are from 4–6 animals. Data in (**g**) are from two independent experiments performed on 12–15 animals and from one experiment on 5 animals in (**h**). Pictures are from one representative experiment out of two in (**a, b**), and from 7 representative animals out of 12–15 per group in (**f**). Statistics indicate significance by one-way ANOVA for (**c**); two-way ANOVA for (**d, e, I**); two-tailed Student's *t*-test for (**g, h**); log-rank (Mantel–Cox) test for (**j**). Source data are provided as a Source Data file.

viral proteins, and by RFP detection with fluorescence microscopy and flow cytometry in VSVΔ51-RFP-infected cells (Fig. 4d–f). Given that the NRF2 transcription factor is highly involved in redox homeostasis and the metabolic plasticity of cancer cells[44], we thought to validate our findings with transient silencing of NRF2 in 786-O cells to avoid any compensatory mechanisms or clonal bias that could have emerged from the permanent lack of NRF2 in this cell line. Transient depletion of NRF2 using electroporation of Cas9 and NRF2 targeting guide RNA (*NFE2L2*[gRNA]) did not prevent 4-OI bolstering effects on VSVΔ51 infectivity as observed by flow cytometry and expression of viral proteins by immunoblotting (Fig. 4g, h). To unequivocally exclude the possibility that the NRF2/KEAP1 axis is involved in the 4-OI-mediated increase of VSVΔ51 in cancer cells, KEAP1 was removed using a similar transient CRISPR/Cas9 gene editing strategy. As expected, ablation of KEAP1 led to the induction of the NRF2-regulated protein AKR1B10 (Fig. 4i). However, the absence of KEAP1 did not reduce the capacity of 4-OI to increase VSVΔ51 infection in 786-O cells (Fig. 4i, j). Finally, to investigate whether some of the biological activity of 4-OI could be dependent on the generation of reactive oxygen species (ROS), cells were pretreated with the ROS scavenging agent N-Acetyl-L-Cysteine (L-NAC) prior to 4-OI and virus challenge. The addition of L-NAC did not alter the pro-viral action of 4-OI on VSVΔ51 infectivity (Fig. 4k). Altogether, these data indicate that 4-OI operates independently of the NRF2/KEAP1 axis and ROS to promote VSVΔ51 infection in cancer cells.

## 4-OI inhibits antiviral immunity
LDL receptor (LDLR) serves as one of the major entry receptors for VSV in human and murine cells[45]. We evaluated whether modulation of the LDLR surface protein level (Fig. S8a) and increased viral entry could be part of the mechanism driving 4-OI pro-viral action in 786-O cells. 4-OI treatment did not increase LDLR surface levels (Fig. S8b). Additionally, KO of LDLR in 786-O cells did not alter the capacity of 4-OI to promote VSVΔ51 infectivity (Fig. S8c), thus excluding the modulation of LDLR protein level as a possible mechanism driving 4-OI biological effects on VSVΔ51 infectivity.

To gain further understanding of the possible mechanism involved in 4-OI mediated potentiation of VSVΔ51, we performed bulk RNA sequencing analysis on VSVΔ51-infected 786-O cells. Strikingly, a landscape of antiviral genes being differentially expressed dominated in the initial analysis (Fig. 5a, b and S7b). Indeed, VSVΔ51 infection induced a large cross-section of antiviral genes (IFITs, IFITMs, OASs, ISGs) and different pro-inflammatory cytokines and chemokines (CCL5, CXCL10, IFNB1) that were all downregulated by 4-OI treatment (Fig. 5a, b and Fig. S7b). Gene ontology (GO) term analysis highlighted the enrichment of signaling pathways related to the early and late responses to OVs in cancer cells including interferon gamma and alpha response, TNF signaling via NF-κB and apoptosis that were all being affected by the 4-OI treatment (Fig. 5c). Consistent with our RNA sequencing analysis, the induction of antiviral immunity was impaired in 4-OI-stimulated VSVΔ51-infected cells as shown by the inhibition of IRF3 and STAT1 phosphorylation and the suppression of antiviral proteins such as ISG15 or IFIT1 (Fig. 5d, e). Contrary to the pro-viral action with VSVΔ51, 4-OI had an antiviral effect on wild-type VSV (wtVSV) infection in 786-O cells reducing its replication greater than 5-fold (Fig. 5f). As wtVSV is reported to robustly interfere with type I IFN production[46], the immunosuppressive effect of 4-OI would therefore be redundant since wtVSV largely affects cellular antiviral responses itself. 4-OI also inhibited Vaccinia virus, Measles virus, and Reovirus-induced antiviral responses (Fig. S9a, b). However, this inhibition did not lead to an increased infectivity of these different viruses (Fig. S3). Instead, 4-OI even had an antiviral action against Vaccinia virus and Measles virus (Fig. S3). Finally, in line with the moderately increased virus infectivity in normal HUVECs (Fig. S2a), the levels of antiviral proteins such as ISG15 and IFIT1 were only slightly altered in response to 4-OI treatment (Fig. S9c). Conversely, DMF, which greatly promoted VSVΔ51 infection in HUVECs (Fig. S2c), led to a strong suppression of virus-induced immune responses in this cell type (Fig. S9d).

Since we previously showed that KO of NRF2 or KEAP1 did not limit the capacity of 4-OI to promote VSVΔ51 infection in cancer cells (Fig. 4), we further investigated whether the inhibition of the antiviral response to the virus by 4-OI was also modulated independently of the NRF2/KEAP1 axis. Using 786-O NRF2 KO cell line or cells transiently silenced for NRF2 or KEAP1 (786-O *NFE2L2*[gRNA] and *KEAP1*[gRNA]), we demonstrated that 4-OI impaired the expression of the antiviral proteins IFIT1 and ISG15 and reduced the release of CXCL10 in the supernatants of virus-infected cells in the absence of NRF2 or KEAP1 (Fig. 5g–i). Since VSV is an RNA virus engaging the RIG-MAVS signaling axis, we tested whether 4-OI could also inhibit the response to a sequence-optimized RIG-I agonist (M8)[47,48]. Predictably, 4-OI treatment reduced RIG-I signaling following M8 stimulation in both control and NRF2 KO cells as demonstrated by the inhibition of IFIT1 and ISG15 induction, and the reduction in IRF3 dimerization (Fig. 5j). These data altogether indicate the strong inhibition of virus-mediated immune responses by 4-OI at the RIG-I/MAVS level.

## 4-OI dampens innate antiviral immunity in vivo but does not affect the distribution of immune cells in the tumor
We next evaluated the effect of 4-OI treatment on immune parameters in vivo. CT26WT-bearing mice were treated with 4-OI, VSVΔ51, or the combination (Fig. 6a) as described in Fig. 2. Again, tumor growth was controlled with the combination treatment confirming the relevance of this independent experiment (Fig. 6b). Five days after the last VSVΔ51 injection, mice were sacrificed to collect tumor, spleen, and tumor-draining lymph node (T-DLN) for further analysis. RT-qPCR on tumor bulk RNA indicated that 4-OI dampened the antiviral and inflammatory responses induced by VSVΔ51 (Fig. 6c), confirming the in vitro data (Fig. 5). Next, we evaluated the effect of the different treatments on the immune cell populations in tumor, spleen, and T-DLN by flow cytometry (Fig. 6d–f, Fig. S10 and S11). Knowing that

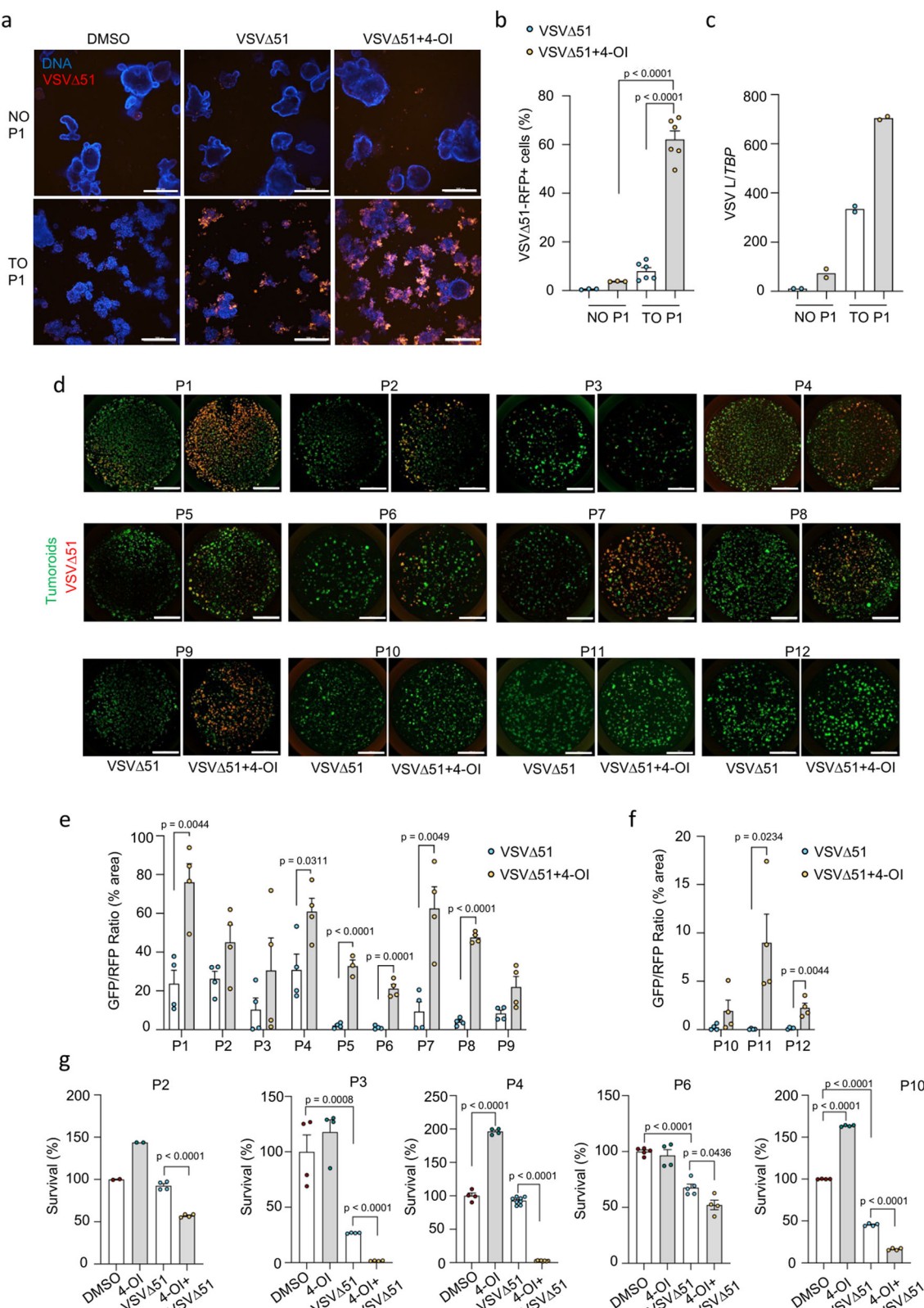

CT26 is an immunologically "hot" tumor model[49], we found it important to assess whether treating tumors with 4-OI and/or VSVΔ51 would influence the presence of immune subsets in the tumor microenvironment (TME) as well as the lymphoid organs. Flow cytometry analysis of infiltrating T-cell subsets in tumors showed that treating tumors i.t. with either or both agents did not influence the density of T-cells (Fig. 6d) in the TME nor the activation of T-cell subsets in both

the tumor and lymphoid organs apart from enrichment of KLRG1+ effector T-cells in the spleen and tumor of VSVΔ51-treated animals (Fig. S11). In addition, no major changes in myeloid subsets were found in TME and spleen although there is a trend towards a clustering of non-myeloid CD45+, CD3−, MHC-II+ cells in the T-DLN after single and combinational treatment with 4-OI (Fig. 6e, f). Overall, our data indicates no negative influence of 4-OI on infiltrating lymphocytes nor

**Fig. 3 | 4-OI promotes selective replication of VSVΔ51 in colon tumor organoids. a** Confocal imaging of colon normal (NO) and tumor organoids (TO) pretreated with 4-OI (125 µM) for 24 h, then infected with VSVΔ51-RFP (1 × 10⁶ pfu/well). Images taken two days post-infection using Hoescht dye as a DNA staining agent. Scale bars, 300 µm. **b** Flow cytometry counts of infected RFP-positive cells in enzymatically digested colon NO and TO organoids from patient 1 (P1) at two days post-infection with VSVΔ51-RFP (1 × 10⁶ pfu/well), with or without 4-OI (125 µM). **c** qPCR analysis of VSV L gene expression normalized to a housekeeping gene (TBP) in colon NO and TO organoids from patient 1 (P1) at one-day post-infection with VSVΔ51 (1 × 10⁶ pfu/well), with or without 4-OI (125 µM). **d** Imaging setup in a 48-well plate to evaluate the infectivity of VSVΔ51-RFP infection (1 × 10⁶ pfu/well) solely and in combination with 4-OI in colon cancer organoids from 12 patients (P1 to P12) at 48 h post-infection. Organoid area evaluated either by GFP transduction or calcein green staining. Scale bars, 3000 µm. **e, f** Representative values from the imaging setup expressed as a percentage of RFP fluorescence from VSVΔ51 normalized to total GFP area from tumor organoids. **g** Remaining luciferase activity to assess in vitro cytotoxicity of VSVΔ51 alone or combined with 4-OI towards luciferase-expressing colon tumor organoids from different patients at 4 days of post-infection is presented as "survival". Data are depicted as means ± SEM from two independent experiments performed in biological triplicates from one patient in (**b**), from one experiment performed in biological duplicates from one patient in (**c**), from one experiment performed in biological triplicates on tumoroids from 12 individual patients in (**e, f**), from one experiment performed in biological quadruplicates from 5 individual patients in (**g**). Pictures are from one representative experiment out of two in (**a**), and from one experiment performed on all individual patient material in (**d**). Statistics indicate significance by one-way ANOVA for (**b, g**), and two-tailed Student's t-test for (**e, f**). Source data are provided as a Source Data file.

systemic manipulation of the main T-cell and myeloid populations especially in this model that already has a significant baseline T-cell infiltration. The dampening of antiviral and inflammatory responses after combinational treatment might primarily benefit the oncolytic properties of VSVΔ51 oncolysis.

## 4-OI targets MAVS and JAK1 to suppress antiviral immunity

We next sought to dissect the molecular mechanism(s) of 4-OI-mediated suppression of the antiviral and inflammatory response to VSVΔ51. To further determine at what level in the pathway 4-OI interferes with RIG-I/MAVS signaling, the active CARD domain containing the form of RIG-I (RIG-IN), the MDA5 sensor, the antiviral adapter protein MAVS, the TBK1 kinase, or the active form of IRF3 [IRF3(5D)] were expressed in the presence of 0 (−), 62.5 (+) or 125 (++) µM of 4-OI. Immunoblot analysis represents the dynamics of endogenous IFIT1 and RIG-I regulation involved in the type I IFN pathway. All expression constructs resulted in an induction of IFIT1 and RIG-I protein levels compared to the transfected cells with the control plasmid expressing only GFP (Fig. 7a). In this experiment, IFIT1 and RIG-I expression induced by MDA5, RIG-IN or MAVS were inhibited by 4-OI, whereas TBK1 and IRF3(5D)-mediated induction was not affected by 4-OI treatment (Fig. 7a). These results indicate that 4-OI inhibits the IFN antiviral response downstream of MAVS and upstream of TBK1.

Chemical proteomic profiling experiments using 4-OI-alk have showed that the compound is able to ligand a large group of proteins in murine cells[42]. Because 4-OI can modify cysteine residues on different proteins such as KEAP1, NLRP3, STING, or JAK1 and alter their function, we hypothesized that proteins involved in the RIG-I/MAVS signaling could be direct targets of 4-OI as well. Indeed, MAVS was previously identified as a covalent binder of 4-OI-alk[42]. Utilizing another published dataset of iodoacetamide-desthiobiotin (IA-DTB)-based competitive cysteine profiling with 125 µM 4-OI[29], we looked for potential competing targets in the RIG-I/MAVS pathway in THP1 human cells. Interestingly, MAVS was one of the identified proteins, with a single cysteine, C283, being weakly competed by 4-OI (Fig. 7b, c). As previously demonstrated, KEAP1 was shown to be significantly modified by 4-OI at C288 thus serving as a strong positive control (Fig. 7b). To confirm MAVS interaction with 4-OI in 786-O cancer cells, cells were stimulated with 4-OI-alk for 4 or 24 h, followed by click chemistry with biotin azide and enrichment of biotinylated proteins. We observed an enrichment in 4-OI-alk labeling of MAVS, demonstrating the protein-drug interaction. Adding non-alkyne-tagged 4-OI to compete with 4-OI-alk led to a reduced interaction between MAVS and 4-OI-alk thus demonstrating the specificity of this interaction (Fig. 7d).

Considering the interaction between 4-OI and MAVS, we next sought to point out whether 4-OI disturbs the RIG-I/MAVS and/or the MAVS/TBK1 interaction. For that, HEK293 cells were transfected with Flag-MAVS and GFP-RIG-I or GFP-TBK1 expression plasmids in the presence or absence of 4-OI. Following Flag immunoprecipitation, a 3.7-fold reduction in the interaction between MAVS and TBK1 in the presence of 4-OI was observed where only a 1.6-fold reduction was seen between MAVS and RIG-I (Fig. 7e). We further tested the importance of MAVS in driving some of the effects elucidated by 4-OI on VSVΔ51 replication. Interestingly, transient knock-down of MAVS phenocopied the effect of 4-OI, which then could not boost the infectivity further in 786-O cells (Fig. 7f-g). In line, knocking down IRF3, a transcription factor acting downstream of MAVS also recapitulated the findings observed in the absence of MAVS, increasing the sensitivity to the virus which could not be further enhanced by 4-OI (Fig. S12a, b). However, 4-OI did increase VSVΔ51 infectivity in cells that were impaired for NLRP3, an already reported target of 4-OI[30] (Fig. S12c, d). Further, a targeted mutation was introduced in MAVS, specifically converting the cysteine residue at position 283 to alanine. Affinity pull-down and ISRE-Luciferase experiments were conducted to study the interaction of 4-OI with the mutated MAVS protein and its contribution to the dampening of 4-OI-mediated antiviral response. The findings indicated a partial reduction of 4-OI-alk labeling of MAVS C283A compared to MAVS wt (Fig. 7h). However, C283A mutation on MAVS led to complete impairment of 4-OI-driven inhibition of ISRE-mediated response (Fig. 7i). Finally, co-IP experiments were carried out to selectively study the interaction involving the mutated MAVS protein and TBK1 in presence or not of 4-OI. 4-OI was no longer capable of reducing MAVS-TBK1 interactions in cells expressing MAVS C283A compared to cells expressing MAVS wt (Fig. 7j).

The first wave of antiviral immunity is constituted by the production of type I IFNs. This initial wave then gets amplified through IFNAR receptor signaling followed by the expression of proteins with antiviral action[50]. Since we demonstrated that 4-OI can interfere with the early MAVS/IRF3 response (Fig. 7), we also set to test whether 4-OI could alter the effects of type I IFN signaling. To investigate this, 786-O cells were pretreated with IFNβ in the presence or absence of 4-OI before challenge with VSVΔ51 (Fig. S13a). Treatment of cells with 4-OI antagonized the antiviral action of type I IFN on VSVΔ51 infection (Fig. S13b). Furthermore, 4-OI dose-dependently reduced type I IFN signaling as shown by the impairment of STAT1 phosphorylation and the inhibition of ISG15 induction (Fig. S13c). This inhibition of STAT1 phosphorylation by 4-OI was largely maintained in NRF2 KO cells (Fig. S13d). 4-OI was recently reported to target the critical kinase JAK1 hence inhibiting type I/II IFN signaling[29]. We confirmed the binding of 4-OI-alk to JAK1 in our cellular model (Fig. S13e). In addition, using CRISPR/Cas9 gene editing, we showed that the transient KO of JAK1 highly enhanced VSVΔ51 infection in 786-O cells and 4-OI could not promote it further (Fig. S13f, g). Altogether these data indicate that 4-OI promotes VSVΔ51 infection in cancer cells through the direct targeting of two critical proteins in the interferon signaling pathway, MAVS and JAK1.

## 4-OI targets IKKβ and promotes viral infection through NF-κB inhibition

In parallel with its impact on antiviral immunity, complementary bioinformatic analysis identified NF-κB as another factor possibly

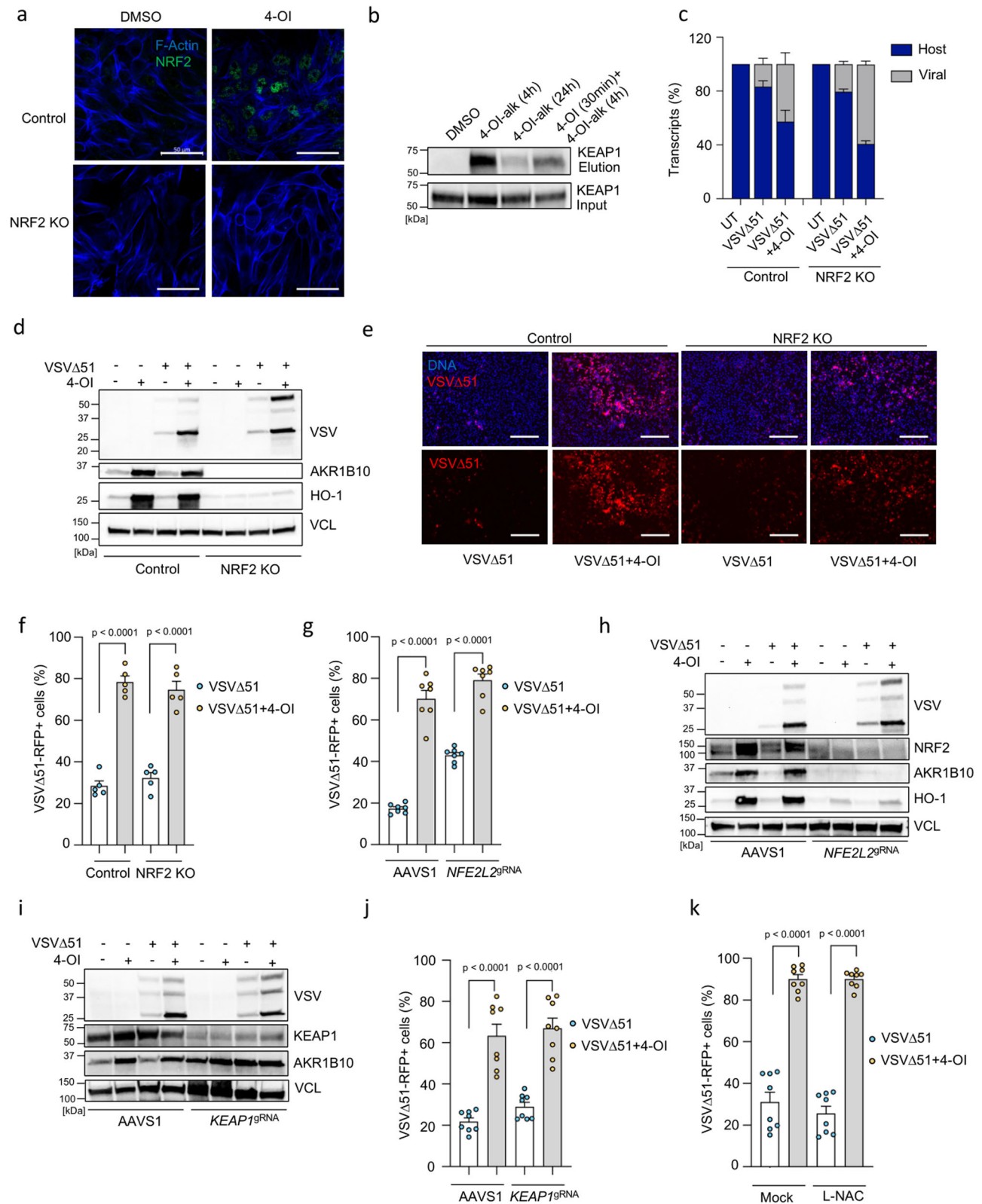

impacted by 4-OI following VSVΔ51 infection (Fig. 8a). Critically, the modulation of NF-κB activity has previously been reported to influence the susceptibility of cancer cells to VSVΔ51 infectivity[21,51]. Upon VSVΔ51 infection, 4-OI inhibited the nuclear translocation of RELA (p65) (Fig. 8b and S14a, b) and reduced the release of IL-6 in the supernatants of virus-infected cells (Fig. 8c). IL-6 production in response to LPS was earlier reported to be impaired by 4-OI in macrophages through the

ATF3-IκBζ axis[22]. We thus investigated whether this regulatory system could be at play in driving some of the 4-OI effects on VSVΔ51 infection. Knocking down ATF3 neither affected the basal VSVΔ51 infectivity levels nor altered the capacity of 4-OI to promote virus infection (Fig. S15a, b). Additionally, VSVΔ51 infection did not upregulate IκBζ protein levels thus fully excluding the ATF3-IκBζ axis as a possible mediator of the biological effects observed with 4-OI on VSVΔ51.

**Fig. 4 | 4-OI enhances VSVΔ51 infectivity independently of NRF2 and KEAP1 in cancer cells. a** NRF2 protein levels in control and NRF2 knockout (KO) 786-O cells treated with 4-OI (75 μM) for 24 h using confocal microscopy. Blue: actin filaments, green: NRF2. Scale bars, 50 μm. **b** KEAP1 levels analyzed in 786-O cells treated with alkynated 4-OI (4-OI-alk) (125 μM) for 4 or 24 h with or without non-alkynated 4-OI (125 μM) by anti-KEAP1 immunoblotting. **c** Viral RNA content assessed by RNA sequencing in VSVΔ51-infected (MOI of 0.01) control and NRF2 KO 786-O cells with or without 4-OI (75 μM) pretreatment. **d–f** Immunoblot analysis in control and NRF2 KO cells pretreated with 4-OI (75 μM) before VSVΔ51 challenge (MOI of 0.01) (**d**). Fluorescence microscopy showing VSVΔ51-RFP spread and cellular layer integrity with Hoechst stain overlay (Scale bars, 300 μm) (**e**) and quantification of infected cells by flow cytometry at 17 h post-infection (**f**). **g–j** Quantification of virus-infected cells by flow cytometry in 786-O cells transiently KO for NRF2 (**g**) or KEAP1 (**j**). Immunoblot analysis in NRF2 KO (**h**) and KEAP1 KO cells (**i**) pretreated with 4-OI (75 μM) before VSVΔ51 challenge (MOI of 0.01). **k** 786-O cells incubated with L-NAC (1 mM) for 3 h before 4-OI challenge (75 μM) for 24 h, then infected with VSVΔ51-RFP (MOI of 0.01). Quantification of infected cells by flow cytometry at 17 h post-infection. Data are means ± SEM from two independent experiments in biological duplicates and triplicates in (**f**), and in biological triplicates and quadruplicates in (**g**). Two experiments in biological quadruplicates in (**j**) and (**k**). Images from one representative experiment in triplicates in (**a**) and (**e**). Data in (**b**) and (**d**) from one representative of three independent experiments. Data as means ± SEM from one experiment in biological triplicates in (**c**). Statistics indicate significance by one-way ANOVA for (**f, g, j, k**). Vertical stacks of bands are not derived from the same membrane in (**d, h, I**). Source data provided in the Source Data file.

To further pinpoint at what level in the NF-κB signaling axis 4-OI inhibited the response, we again turned to published data on covalent interactors of 4-OI. IKKβ, one of the critical kinases involved in NF-κB signaling, was found to be strongly enriched by 4-OI-alk[42] and by IA-DTB competitive profiling[29]. We found C179 and C464 in IKKβ to be significantly competed by 4-OI treatment (Fig. 8d, e), indicating these might be the target residues. We further validated the interaction between IKKβ and 4-OI-alk in cancer cells by pull-down and immunoblotting experiments (Fig. 8f). The addition of non-tagged 4-OI competed with 4-OI-alk and reduced the interaction between IKKβ and 4-OI-alk thus demonstrating the specificity of this interaction (Fig. 8f). Moreover, 4-OI-alk exhibited low binding to closely related kinases, such as IKKγ and IKKε, in comparison to IKKβ (Fig. 8f).

In order to get some structural insight on potential binding sites and mode of interaction of 4-OI with human IKKβ, we modeled the Michael addition reaction of 4-OI with six cysteine residues that were identified by MS and pull-down experiments (Figs. 8g, S16, S17). We used two crystal structures of human IKKβ covering residues 1–664 (PDB ID: 4KIK)[52] and 701–746 (PDB ID: 3BRV)[53] for the covalent docking studies. Analysis of the crystal structures indicated that all targeted cysteines were located near the outer surface of IKKβ. 4-OI, however, displayed rather high binding energies, especially to the binding sites containing Cys179, Cys412, and Cys463 (Table S3), the top three bound cysteines in IKKβ from the competitive proteome profiling (Fig. 8e). Cys179 is situated within the activation loop of IKKβ between the phosphorylation sites necessary for IKKβ activation. 4-OI showed favorable binding to Cys179, where its carboxyl group can be engaged in two hydrogen bonds with Lys171 and Thr180 (Fig. 8h). Besides concealing the role of Cys179 for IKKβ activity, octyl-itaconation of Cys179 would affect the conformational plasticity of the activation loop and may sterically hinder substrate binding. Cys12 and Cys179 belong to the N-terminal kinase domain. Cys412, Cys464, and Cys524 are situated in the α-helical scaffold/dimerization domain mediating homo- or heterodimer formation, and Cys716 is found at the C-terminal region involved in the interaction with the regulatory sub-unit NEMO of the IKK complex. Covalent docking indicated that binding of 4-OI at the Cys12 site can be stabilized by a hydrogen bond between the carboxyl group and Cys12, whereas the octyl tail is likely accommodated in a hydrophobic trough formed by Ala14, Trp15, Thr37, Pro80, and Met83 (Fig. S16a, b). Cys412 is located in a hydrophobic pocket suitable for binding small molecules. Inevitably, 4-OI displays the highest binding scores for this site (−8.2 to −10.0 kcal/mol, Table S3). The carboxyl group of 4-OI established a network of electrostatic interactions with Asp248, Leu249, Asn250, Cys412, and Arg419. The additional hydrogen bond between the C4-carbonyl group and Ser409 was observed (Fig. S16c, d). Cys464 belongs to the leucine zipper region (458–479) of the α-helix α2s. 4-OI is predicted to establish multiple hydrogen bonds through its carboxyl and C4-carbonyl moieties with Ser463, Cys464, and Lys467. Moreover, hydrophobic interactions between the octyl chain and Met456 were detected (Fig. S16e, f). Cys524 is found within a loop between helices

α3s and α4s. The carboxyl group of 4-OI is expected to be involved in an ionic interaction with Arg526, while the octyl motif is oriented to an adjacent shallow hydrophobic groove between the α3s and α6s helices (Fig. S16g, h). Cys716 is situated in the N-terminal segment of the C-terminal α-helix mediating NEMO binding. 4-OI could participate in electrostatic interactions via its carboxyl and octyl functionalities with Cys716 and Glu720, respectively (Fig. S16i, j). Interestingly, the super-position of the predicted 4-OI binding pose to Cys716 and structure of IKKβ–NEMO complex clearly showed major steric clashes between the octyl motif of 4-OI and Gln67, Ser68, Ile71, and Arg75 of NEMO (Fig. S17), suggesting that the covalent binding of 4-OI to Cys716 may disrupt the IKKβ interaction with NEMO.

Altogether, the results of proteomics and covalent docking of 4-OI in human IKKβ indicate that 4-OI can modulate IKKβ on multiple levels including modification of the critical Cys179, or interference with protein–protein interactions. In line with all the modeling predictions, 4-OI inhibited IKKβ-induced NF-κB luciferase activity in both control and NRF2 KO cells (Fig. 8h), as well as reduced the phosphorylation of IKKα/β, IκBα and RELA (p65) (Fig. 8i). Finally, impairment of IKKβ and RELA (p65) by CRISPR/Cas9 gene editing phenocopied the effect of 4-OI which then could not boost the viral infectivity further (Fig. 8j, k and Fig. S15c, d). Recently, itaconate and its derivative 4-OI were reported to target the DNA dioxygenase TET2 to suppress LPS-induced NF-κB-related genes[54]. However, knocking down TET2 did not alter the capacity of 4-OI to promote VSVΔ51 infectivity in 786-O cells, thus excluding a possible involvement of TET2 in driving some of the biological effects observed here (Fig. S15e, f). Additionally, a point mutation was applied to IKKβ, specifically altering the cysteine residue at position 179 to alanine. NF-κB luciferase assay was performed to investigate the impact of the C179 mutation on modulating the capacity of 4-OI to suppress inflammatory response. The results highlighted the notable involvement of the cysteine residue at position 179 in influencing the modulation of the NF-κB-mediated inflammatory response in response to 4-OI (Fig. 8l). Together, the data indicate that 4-OI disrupts NF-κB signaling hence promoting VSVΔ51 infection in cancer cells through IKKβ alkylation.

## Discussion

Resistance to OV therapy can be multifactorial but one of the recurrent observations is the persistent residual antiviral immunity observed in cancer cells that is limiting the penetrance and the oncolytic action of therapeutic viruses especially for VSVΔ51[55]. Here, we demonstrate that the itaconate derivative 4-OI enhances the replication and the oncolytic action of VSVΔ51 in resistant cancer cells and primary patient-derived colon tumoroids or organotypic brain tumor slices. We additionally demonstrate that the combination of 4-OI with VSVΔ51 enhances therapeutic outcomes in a murine colon tumor model where VSVΔ51 is ineffective solely. Mechanistically, we show that 4-OI can overcome antiviral immune responses by simultaneously targeting the RIG-MAVS-IRF3, the IKKβ-NF-κB and the JAK1-STAT1 signaling pathways through the direct alkylation of MAVS, IKKβ, and JAK1, respectively.

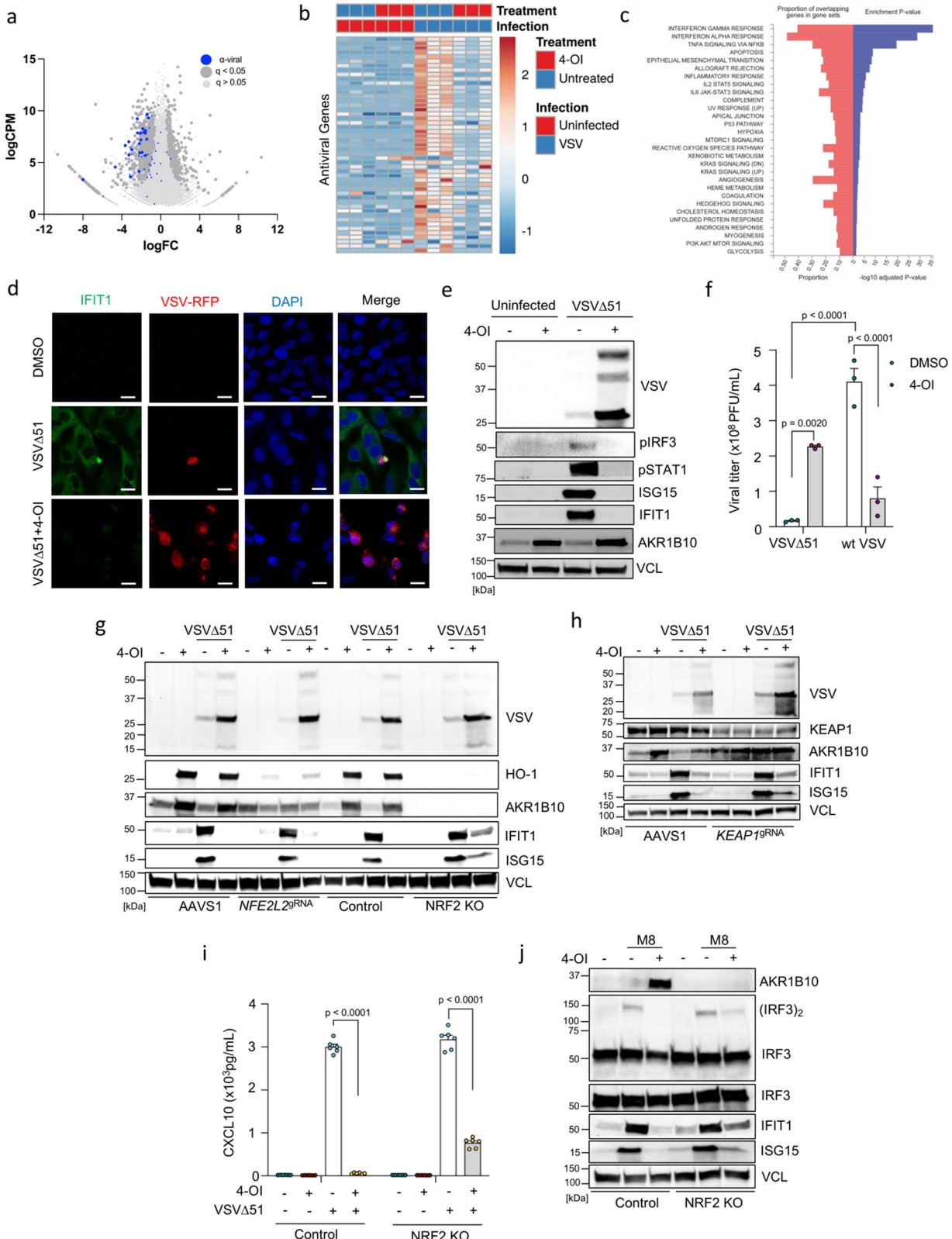

Octyl itaconate was recently demonstrated to activate the transcription factor NRF2 through the covalent engagement of its inhibitor KEAP1, hence, leading to the impairment of type I IFN and pro-inflammatory cytokine release[23,25]. The induction of the anti-oxidative transcription factor NRF2 has long been known to have a potent anti-inflammatory action[56]; however, it is only recently that NRF2-inducing compounds such as 4-OI were shown to suppress inflammation independently of NRF2 via the direct targeting of signaling molecules such as STING[32], JAK1[29], or NLRP3[30]. Consistent with this, we demonstrate that the enhancement of VSVΔ51 by 4-OI operates largely independently of the NRF2/KEAP1 axis. Nonetheless, we are not excluding the fact that other NRF2 activating compounds may also be beneficial to promote oncolytic virotherapy replication through NRF2-mediated suppression of

**Fig. 5 | 4-OI impairs VSVΔ51-induced antiviral immune responses. a–c** 786-O cells pretreated with 4-OI (75 μM) for 24 h and infected with VSVΔ51 (MOI 0.01) for 17 h. RNA sequencing analysis emphasizing on antiviral genes (blue dots) in the volcano plot (**a**), differentially expressed interferon-stimulated genes (ISGs) in the heat map (**b**), and top KEGG pathways affected by 4-OI during viral infection (one-sided hypergeometric test, Benjamini–Hochberg method was applied to adjust the *p*-value for multiple testing) (**c**). **d, e** 786-O cells pretreated with 4-OI (125 μM) for 24 h and infected with VSVΔ51-RFP (MOI 0.01) for 24 h. IFIT1 levels assessed by fluorescence microscopy (**d**), and Western blot performed on cell lysates for antiviral proteins (**e**). **f** 786-O cells pretreated with 4-OI (75 μM) for 24 h infected with wild-type VSV (wt VSV) or VSVΔ51 at MOI 0.01. Viral titers determined 24 h post-infection. **g–i** Control and NRF2 KO 786-O cells, as well as 786-O cells transiently KO for NRF2 or KEAP using CRISPR/Cas9, pretreated with 4-OI (75 μM) for 24 h and infected with VSVΔ51 (MOI 0.01) for 17 h. Immunoblots in (**g, h**). CXCL10 release measured by ELISA from supernatants in (**i**). **j** Control and NRF2 KO 786-O cells pretreated with 4-OI (75 μM) for 24 h and stimulated with the RIG-I agonist M8 (3.5 ng/mL) for 5 h. Western blot performed on cell lysates. Data are from one experiment performed in triplicate in (**a–c**). Images are from one experiment in (**d**). Data are from one representative experiment performed at least three times in (**e**). Data are depicted as means ± SEM from one experiment performed in triplicate in (**f**). Data are from one representative experiment out of three in (**g**), out of two in (**h**) and (**j**). Data are depicted as means ± SEM from two experiments performed in triplicate in (**i**). Statistics indicate significance by two-way ANOVA for (**f, i**). Vertical stacks of bands are not derived from the same membrane in (**e, g, h, j**). Source data provided as a Source Data file.

antiviral responses, as previously reported by our group with the isothiocyanate sulforaphane[20].

Dimethyl fumarate (DMF), an FDA-approved drug used in the treatment of multiple sclerosis, has been reported to potentiate oncolytic virotherapy through inhibition of NF-κB activity and limitation of type I IFN response[21]. Like 4-OI, DMF is also a Krebs cycle-derived metabolite described for its NRF2-inducing activity[21]. A side-by-side comparison with DMF showed a greater capacity of 4-OI to bolster VSVΔ51 infectivity in vitro in tumoral cells, especially at low micromolar concentrations. This difference in pro-oncolytic action between 4-OI and DMF could be explained by their slightly different mode of action. DMF was shown to promote VSVΔ51 replication by inhibiting NF-κB and impair type I IFN production but did not affect IRF3 transcriptional activity[21]. Here, we demonstrate that 4-OI targets three critical proteins MAVS, IKKβ, and JAK1, hence fully abrogating the early and late immune responses to the virus. Importantly, the wt matrix protein has the capacity to inhibit type I IFN response and suppress NF-κB activity while the same mutant protein in VSVΔ51 loses its effect[46,57]. Notably, the pro-viral action of 4-OI is restricted to VSVΔ51 and had no such effect on wt VSV. Paradoxically, we even observed an antiviral action of 4-OI on the wt virus, which was previously reported for other RNA viruses including SARS-CoV-2, Zika virus, or Influenza A virus[34,37]. 4-OI also displayed antiviral effects on other oncolytic viruses including Vaccinia virus and Measles virus. Future studies will be needed to characterize the pro *vs* antiviral mode of action of 4-OI on these different RNA and DNA oncolytic viruses.

We demonstrate that 4-OI inhibits antiviral responses by decreasing the production of type I IFN and the expression of antiviral proteins via the modulation of the RIG-I/MAVS, the IKKβ/NF-κB, and the JAK1/STAT1 pathways. Mechanistically, 4-OI covalently binds cysteine residues in MAVS, IKKβ, and JAK1, hence affecting their functionality. In support of our findings, JAK1 was recently shown to be modified by 4-OI leading to the inhibition of alternative macrophage activation following IL4/IL13 stimulation[29]. We report here two targets of 4-OI, MAVS, and IKKβ. While our data show that 4-OI targeted MAVS at Cysteine (Cys)283 and prevented the interaction between MAVS and RIG-I hence limiting IRF3 transcriptional activity, we were not able to speculate on the precise model of action of 4-OI as a crystal structure of MAVS in that specific region is not available. Cys283 however was reported to be a cleavage site of MAVS by proteases from the hepatitis C virus, resulting in the inhibition of RIG-I pathway signal transduction[58], thus suggesting that Cys283 is an important residue to drive immune responses to viral infection. Using mutational analysis, we were able to confirm the importance of Cys283 in binding to 4-OI. Furthermore, this cysteine residue was shown to be critical in driving some of the biological effects of 4-OI in the shutdown of the RIG-I/MAVS signaling axis. Of note, we did observe some residual binding of the overexpressed C283A MAVS to 4-OI-alk in affinity pull-down assay. This could be explained by the fact that Cys283 is not the sole amino acid residue modified by 4-OI on MAVS. Indeed, we used a cysteine profiling approach to identify alkylated residues by 4-OI which does

not exclude the possibility that other amino acid residues could also be modified by 4-OI on MAVS.

4-OI also alkylates IKKβ at several cysteine residues with Cys179 being the predominant one. Interestingly, OV infection can be improved in vitro by using the IKKβ inhibitor TPCA-1[21], as we have shown here with 4-OI. Mechanistically, our modeling experiments indicate that 4-OI can affect IKKβ on multiple levels including modification of the critical Cys179, or interference with protein–protein interactions especially with the adapter protein NEMO, a critical adapter molecule bridging the innate antiviral and inflammatory responses[59]. Using the same mutational analysis approach, we also confirmed the importance of Cys179 in the control of NF-κB-mediated inflammatory responses and validated the requirement of Cys179 in controlling some of the biological effects of 4-OI in the negative regulation of NF-κB pathway. Altogether, 4-OI represents an attractive small molecule that can promote viral replication of highly IFN-sensitive viruses such as VSVΔ51 in resistant cancer cells through targeted inhibition of antiviral and inflammatory pathways.

We conducted a series of experiments in which we evaluated the levels of various immune markers and antiviral proteins using immunoblotting in primary HUVECs from different donors upon VSVΔ51 infection, both with and without 4-OI. At the given dosage of 4-OI, only a moderate reduction in antiviral markers was observed in primary HUVECs, explaining the limited replication of VSVΔ51 in these cells (Fig. S9c). The stronger suppression of antiviral immunity by 4-OI in cancer cells can be attributed to several factors, including the molecule's penetrance in cancer versus normal cells, the binding affinity of different targets (such as MAVS, JAK1, IKKβ) to 4-OI in normal/cancer cells, and the magnitude of the immune response to VSVΔ51 that the drug must overcome in normal/cancer cells. These parameters are crucial in influencing the observed outcome of virus replication in normal/cancer cells.

Itaconate derivatives are currently holding a lot of promise as immunomodulatory drugs[23,29,30]. However, their general anti-inflammatory profile might argue against their use as anticancer agents. In the current work, local administration of 4-OI did not affect the composition and activation of CD8+ effector T-cells in the immunologically hot tumor microenvironment nor systemically, suggesting that 4-OI would not impair immunotherapy in this model. Dimethyl itaconate was shown to prevent colorectal cancer by dampening the inflammation in the tumor immune microenvironment[60], and 4-OI was recently demonstrated to inhibit the development of ovarian cancer by blocking the communication between cancer and immune cells[61]. Considering the data that we are presenting; this opens the door for further exploration of itaconate derivatives in tumor biology, particularly for cancer immunotherapies.

Oncolytic virotherapy is a promising form of anticancer therapy that is unfortunately often weakly effective due to the occurrence of virus-resistant tumor cells. A recent study by Bhatt et al. developed a mathematical model to study the 3D spatial dynamics of virus-sensitive *vs* virus-resistant tumor cells[62]. In this model and as previously

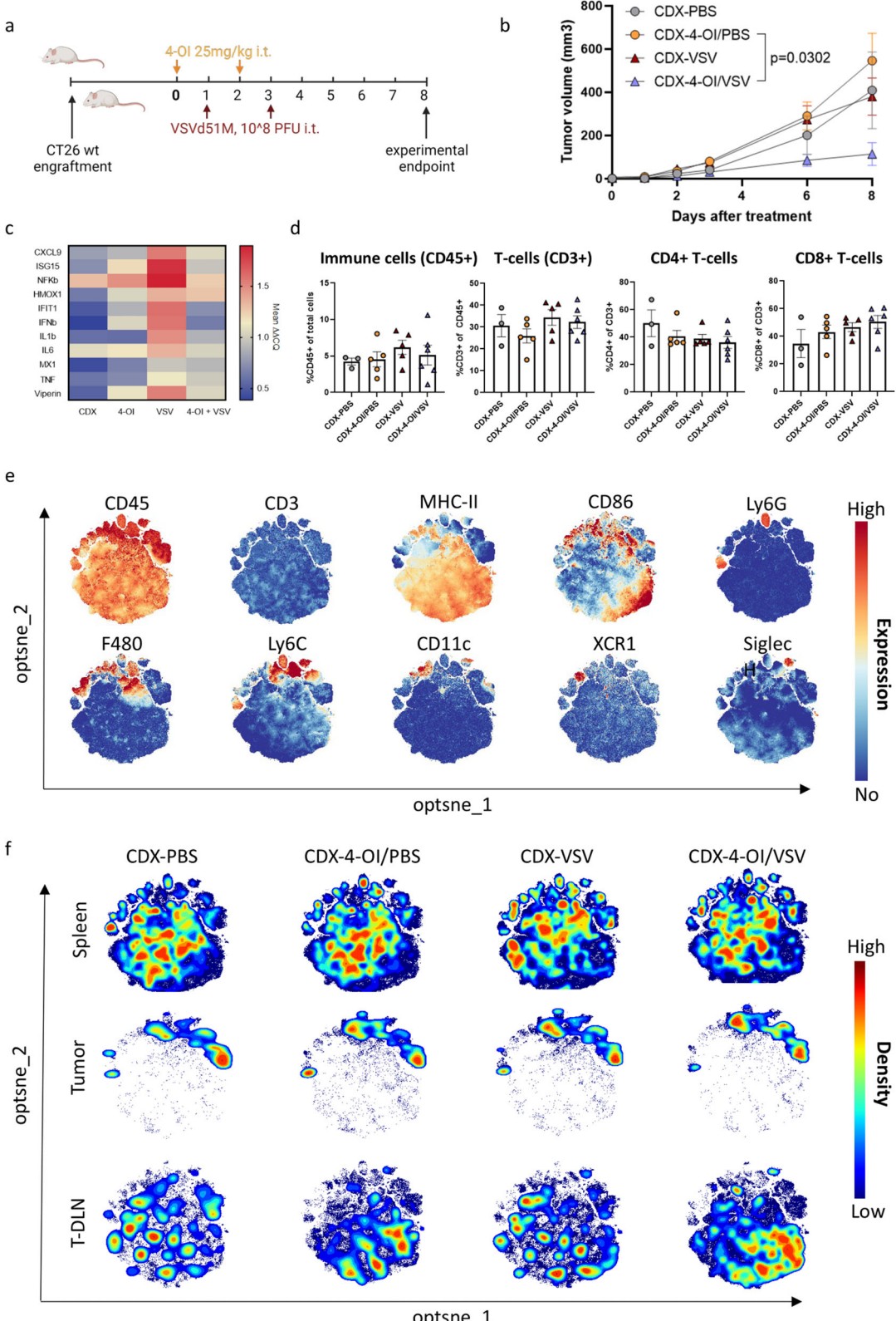

reported in Berg et al.[63], the addition of a third dimension profoundly altered the dynamics of virus–tumor interaction and significantly reduced the probability of the virus eliminating the tumoral cells in 3D *vs* 2D models. Oncolytic virotherapies are usually studied in vitro in two-dimensional models or in vivo in syngeneic and xenograft animal models and only a few studies using patient-derived 3D models as preclinical tools have been performed to test the efficacy of oncolytic

virotherapies in combination with drug sensitizers. Importantly, the treatment with 4-OI promotes VSVΔ51 infection in a large panel of 3D patient-derived colon cancer tumoroids. We complemented this approach using organotypic brain tumor slices, where innervation, blood vessels, and immune or stromal cells were preserved. We observed the same enhancing capacity of 4-OI in these physiologically relevant 3D tumoroid and organotypic models. Additionally, 4-OI

**Fig. 6 | 4-OI dampens innate antiviral immunity in vivo but does not affect the distribution of immune cells in the tumor. a** BALB/c mice were injected subcutaneously with $1 \times 10^5$ CT26WT tumor cells. Treatment via intratumoral (i.t.) injections with vehicle (40% CDX) in PBS or with 4-OI (25 mg/kg/dose) in 40% CDX in PBS for 24 h prior challenge with VSVΔ51 expressing firefly luciferase ($10^8$ PFU) was given as indicated by arrows. Mice were euthanized seven days after the first VSVΔ51 injection for analysis. **b** Mean ( ± SEM) tumor growth of the groups followed the start of the treatment regimen ($n = 3$ in the CDX-PBS group; $n = 5$ in the CDX-4-OI/PBS group, $n = 6$ in CDX-VSV and CDX-4-OI/VSV groups). Statistics on tumor volumes at day 8 indicate significance by one-way ANOVA followed by Šídák's multiple comparisons test on tumor sizes at day 8. **c** Relative expression analyzed by RT-qPCR of RNA isolated from bulk tumor samples. $n = 3$ in the CDX-PBS group; $n = 5$ in the CDX-4-OI/PBS group, $n = 6$ in CDX-VSV and $n = 4$ CDX-4-OI/VSV groups (two samples excluded due to insufficient tumor material) **d** Flow cytometry data indicating the distribution of main T-cell populations within tumors upon different treatments analyzed by manual gating in FlowJo software. Mean ± SEM is displayed and compared per treatment group. Each data point represents one animal. $n = 3$ in the CDX-PBS group; $n = 5$ in the CDX-4-OI/PBS group, $n = 5$ in CDX-VSV (one sample lost during acquisition), and $n = 6$ CDX-4-OI/VSV groups. **e** Expression intensity profile of myeloid markers on clustered live, CD45+, CD3− cells from merged samples ($n = 60$, 3 different organs) to distinguish the regional expression of single myeloid markers. Relative expression is indicated by color where red indicates high expression and blue represents no expression within the cluster. **f** Opt-SNE cluster plots of live, CD45+, CD3− cells from indicated organs and treatment group ($n = 3$ in the CDX-PBS group; $n = 5$ in the CDX-4-OI/PBS group, $n = 6$ in CDX-VSV and CDX-4-OI/VSV groups). Cell density is indicated by color where red indicates high density and blue indicates low density within the cluster. Analysis generated using OMIQ software. (**a**) was created using BioRender.com. Source data are provided as a Source Data file.

empowered the oncolytic capacity of VSVΔ51 in vivo in the resistant syngeneic murine colon cancer model CT26.

Overall, we show that the itaconate derivative 4-OI bolsters the oncolytic action of VSVΔ51 both in cancer cell lines and state-of-the-art 3D tumoroid or organotypic brain tumor slices. The fine-tuning of immune responses using anticancer proto-drug-derived metabolites and the use of oncolytic viruses already in clinical trials offers a clear platform for the clinical testing of such a potent and promising combination therapy.

## Methods

Animal experiments are conformed to the requirements of the Danish Experimental Animal Expectorate, the institutional Animal Welfare Body of the LUMC. They are also conformed to the University of Ottawa Animal Care and Veterinary Service guidelines in Canada. The study on patient-derived material was approved by the institutional review board of the UCT and the Ethical Committee at the University Hospital Frankfurt and by the Central Denmark Region Committee for Health Research Ethics.

### Cell lines and culture conditions

CT26WT, B16F10, and NCI-H358 cells were a kind gift from Martin R. Jakobsen (Aarhus University). HT-29 cells were provided by Lasse S. Christensen (Aarhus University) and Vero cells from Søren R. Paludan (Aarhus University). DK-MG and H4 cells were generously given by Anna-Liisa Levonen (Virtanen Institute, Kuopio, Finland). 4T1 and HEK 293T cells were a gift from Tommy Alain (University of Ottawa). 786-O cells were purchased from ATCC. All cell lines except 786-O, DK-MG, and NCI-H358 were cultured in high-glucose Dulbecco's modified Eagle's medium supplemented with 10% heat-inactivated fetal bovine serum (Gibco), 2 mM L-glutamine (Gibco) and 1% penicillin/streptomycin (Gibco). 786-O, DK-MG, and NCI-H358 cells were cultured in RPMI-1640 medium (both from Sigma) using the same supplements. All cell lines were incubated at 37 °C in a 5% $CO_2$ humidified incubator (Thermo Scientific) and verified to be mycoplasma-free (Eurofins genomics, Germany). Experiments with the human BxPC3 cells were performed in the Hiscott laboratory (Pasteur Institute, Rome). Cells were obtained from ATCC and cultured in RPMI-1640 medium (Euroclone) supplemented with 10% heat-inactivated fetal bovine serum (FBS) (Gibco) and 1% antibiotics (Euroclone) and verified to be mycoplasma negative. Experiments with the PANC1 cell line were performed in the Alain laboratory (Ottawa University/CHEO).

### Generation of NRF2 KO 786-O cells

786-O CRISPR-edited NRF2 KO cells were produced by transfecting cells with pLentiCRISPR-v2 (a gift from Dr Feng Zhang, Addgene plasmid #52961) containing single-guide (sg) RNAs directed against the KEAP1-binding domain within the NFE2L2 locus (TGGAGGCAA GATATAGATCT). After 2 days of puromycin selection, cells were

clonally selected by serial dilution, and positive clones were identified as previously described[64]. Control cells referred to as 786-O Control are the pooled population of surviving cells transfected with an empty pLentiCRISPR-v2 vector treated with puromycin.

### Electroporation of 786-O cells with gRNA and Cas9 protein

786-O knockout lines were generated based on the recommendations of the manufacturer (Synthego) using the 4D-Nucleofector (Lonza) program CM-138. Briefly, cells were detached using TrypLe Express (Gibco) and $8 \times 10^5$ cells were nucleofected with a complex of 24 μg of Cas9 Nuclease V3 protein (ID technology) + 12.8 μg of sgRNA (Synthego, USA) in OptiMEM media (Gibco). After nucleofection cells were resuspended in prewarmed RPMI medium and incubated for 48 h until further experiments. Sequences of the synthetic guide RNA used are as follows; *AAVS1* (served as a control) - G*G*G*GCCACUAGGGACAGGAU, *ATF3* - U*A*G*CCCCUGAAGAAGAUGAA, *JAK1* - A*A*A*AGGACAAGGCCUCCUC GU, *NLRP3* - G*A*A*UCCCACUGUGAUAUGCC, LDLR - U*G*C*AUUCCC GUCUUGGCAC, *KEAP1* - G*C*C*UGGGAUCUGGCUGCAUG, *IRF3* - U*G*C ACCAGUGCCUCGGCCC, *MAVS* - U*C*A*GCCCCAGAGCCAUCCCA; *IKBKB* – U*A*G*GCUGACCCACCCCAAUG, *TET2* - U*G*G*UUCUAUCCU-GUUCCAUC. To generate *NFE2L2* knockout, 2 guides were combined at a 1:1 ratio (6.4 μg of each sgRNA) to improve efficacy - A*U*U*UGAUU GACAUACUUUGG and G*C*G*ACGGAAAGAGUAUGAGC.

### Primary cells and culture conditions

Primary fibroblasts were a kind gift from Søren R. Paludan (Aarhus University) and were cultured in high-glucose Dulbecco's modified Eagle's medium supplemented with 10% heat-inactivated fetal bovine serum (Gibco), 2 mM L-glutamine (Gibco) and 1% penicillin/streptomycin (Gibco).

Primary, single-donor early passage HUVECs were kindly provided by Dr. Joanna Kalucka (Department of Biomedicine, Aarhus University, Aarhus, Denmark) and were obtained from Lonza, cultured at 37 °C, 5% $CO_2$ humidified incubators, and regularly tested for mycoplasma. HUVECs were maintained in M199 medium (1 mg/mL D-glucose) (Thermo Fisher Scientific) supplemented with 20% fetal bovine serum (FBS) (Biochrom BmgH), 2 mM L-glutamine (Thermo Fisher Scientific), Endothelial Cell Growth Supplement (ECGS)/ Heparin (PromoCell), 100 IU/mL penicillin and 100 mg/mL streptomycin (Thermo Fisher Scientific). For performed experiments, HUVECs were seeded in endothelial cell basal medium (EGM2) (PromoCell) supplemented with endothelial cell growth medium supplement pack (PromoCell). In all experiments, HUVECs were always used as single-donor cultures and were used between passages 2 and 4.

### Human colon cancer organoid/tumoroid model

Resection samples from colorectal cancer patients were provided by the University Cancer Center Frankfurt (UCT). All materials and associated data (including age, sex, MMR status, tumor localization,

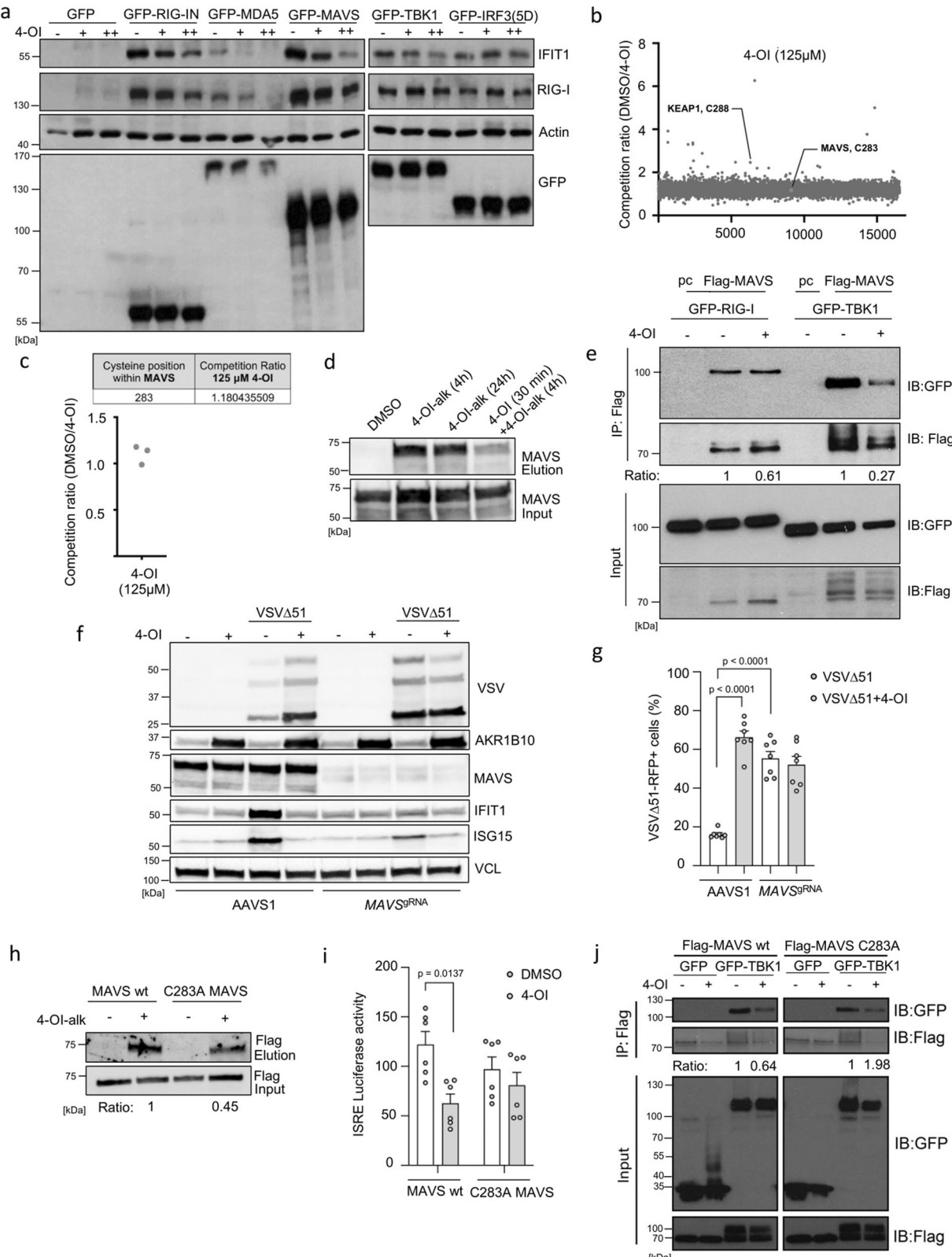

UICC classification, and prior radio-/chemotherapy) were collected after pseudonymization as part of the interdisciplinary Biobank and Database Frankfurt (iBDF) and the study was approved by the institutional review board of the UCT and the Ethical Committee at the University Hospital Frankfurt (Ethics vote: 4/09; project-number SGI-10-2022). Informed consent was obtained by the participants and included permission to publish personal health information.

Patients were not compensated to participate in the study. The organoid biobank used in this proposal has been established from a randomly selected patient cohort, which closely represents the clinical incidence of colorectal cancer. Therefore, the anticipated results will be representative for both male and female subjects although the incidence of colorectal cancer worldwide is higher in men with a relative distribution of about 54% vs. 46% in women which

**Fig. 7 | MAVS modification by 4-OI suppresses antiviral immunity. a** HEK293 cells pretreated with increasing 4-OI concentrations (75 and 125 μM) and transfected with GFP-tagged plasmids. Immunoblotting analyzes antiviral proteins. **b, c** Data from Runtsch et al.[29]. THP1 cells treated with 4-OI (125 μM), prior challenge with IA-DTB, cell lysis, and measurement of modified cysteines by LC-MS. **d** 786-O cells treated with 125 μM of 4-OI-alk (4 or 24 h) with or without 4-OI (125 μM), followed by click chemistry and biotin enrichment. Samples before and after enrichment analyzed by anti-MAVS immunoblotting. **e** HEK293 cells pretreated or not with 4-OI (125 μM) and transfected with Flag-tagged MAVS together with GFP-tagged RIG-I or TBK1 as indicated. Whole-cell extracts prepared and immunoprecipitated with anti-Flag antibody M2; immunoprecipitated complexes or 5% input run on SDS-PAGE and probed with anti-GFP/anti-Flag antibodies. **f, g** Transient MAVS KO 786-O cells treated with 4-OI, then infected with VSVΔ51, analyzed by Western blot (**f**) and flow cytometry (**g**). **h** HEK293 cells transfected with a plasmid encoding Flag-tagged MAVS wt or mutant Flag-tagged MAVS C283A for 24 h.

Subsequently, cells treated with 125 μM of alkynated 4-OI (4-OI-alk) for 4 h, followed by click chemistry and biotin enrichment. Samples before and after enrichment analyzed by anti-Flag immunoblotting. **i** HEK293 cells stimulated with 4-OI prior to transfection with MAVS wt or mutant C283A MAVS, analyzed by ISRE promoter luciferase activity. **j** HEK293 cells transfected with MAVS wt or mutant together with TBK1, treated with 4-OI, analyzed by immunoprecipitation and immunoblotting. Data from one representative experiment in (**a**), three independent experiments in (**b, c**), one representative experiment performed three times in (**d**), one experiment in (**e, h, j**). Data from one representative experiment out of two in (**f**). Data depicted as means ± SEM from two independent experiments performed in triplicates and quadruplicates in (**g**), two independent experiments in triplicates in (**I**). Statistical significance by one-way ANOVA for (**g**) and two-way ANOVA for (**I**). Vertical stacks of bands are not derived from the same membrane in (**a, e, f, j**). Source data provided in Source Data file.

is likely linked to differences in other risk factors (e.g. lifestyle, awareness, screening).

Tumor organoids were generated from tissues as described in ref. 40 (see Table S1). In parallel, normal organoids were generated from tumor-adjacent normal regions. Their normal status was validated by growth arrest in the absence of Wnt3a to exclude tumor cell contamination. Patient-derived organoids were established and cultured as previously described[65]. Organoids were embedded in basement membrane extract (BME) (Cultrex UltiMatrix Reduced Growth Factor BME, Bio-techne). Medium contained advanced DMEM/F12 supplemented with 10 mM Hepes, 1× Glutamax, 1× penicillin/streptomycin, 2% B27, 12.5 mM N-acetylcysteine, 500 nM A83-01 (R&D Systems), 10 μM SB202190 (Sigma-Aldrich), 20% R-spondin 1 conditioned medium, 10% Noggin conditioned medium, 50 ng/mL human EGF (Peprotech). For culture of normal organoids Wnt surrogate (35 ng/mL, # N001-0.5 mg, ImmunoPrecise) was added. Organoids were split every week by mechanical disaggregation at a 1:4 or 1:3 ratio, washed from the remaining matrix, pelleted, and seeded in a fresh BME. After BME had solidified at 37 °C for 30 min, prewarmed complete media was added. 10 μM Y-27632 (Selleckchem) was added to the culture medium after splitting, and the medium was refreshed every second day.

## Human neurosurgical specimens and ethical compliance

All procedures with human brain tumor tissue and data were approved by the Central Denmark Region Committee for Health Research Ethics (official name in Danish: De Videnskabetiske Komitéer for Region Midtjylland); journal number: 1-10-72-82-17 and conducted in accordance with the ethical principles of the World Medical Association Declaration of Helsinki[66]. Surgical specimens were obtained at Aarhus University Hospital (Denmark) from patients undergoing resection of primary or secondary brain tumors. The study specimens were surplus to diagnostic requirements. All patients included in the study provided informed consent for participation. Consent included permission to publish personal health information, including sex, age, diagnosis, and medical center. Patients were not compensated to participate in the study. Three patients were included in the study (Table S2). The sex of study participants was assigned based on the sex-specific individual Danish civil registration number and was considered in the study design or patient enrollment. Diagnoses were validated for all patients based on pathology examination and standardized WHO criteria[67]. Baseline characteristics, diagnoses, and relevant genetic profiles are given in Table S2.

The tumor specimens were approximately 0.5–1 cm³ each yielding approximately 9–18 viable slices for viral infection. Resection was performed without cauterization and minimal manipulation to maintain organotypic structure. The excised specimens were placed immediately in a sterile container filled with ice-cold artificial cerebral spinal fluid (aCSF) cutting solution comprised (in mM): 75 Sucrose, 84

NaCl, 2.5 KCl, 1 NaH$_2$PO$_4$, 25 NaHCO$_3$, 25 D-glucose, 0.5 CaCl$_2$.2H$_2$O and 4 MgCl$_2$.6H$_2$O. The pH was adjusted to 7.3–7.4 and the osmolality was between 295 and 305 mOsm/kg after bubbled with carbogen (95% O$_2$, 5% CO$_2$) gas. Throughout collection and transportation, the samples were placed on ice, connected to a portable container of carbogen gas, and transported to the laboratory (~15 min travel time).

## Processing and culture of human ex vivo brain tumor samples

The human ex vivo tumor slice culture method was based on established procedures[41,68,69]. The tumor specimens were mounted onto the vibrating microtome platform and were cut in sterile, ice-cold, oxygenated cutting aCSF slicing solution. Slices of 300 μm thickness were prepared using a Leica 1200S vibratome (Leica Microsystems, Germany). After incubating in the cutting aCSF (34 °C) for 30 min, the tumor slices were transferred to sterile low-calcium aCSF (15 °C) composed of (in mM): 130 NaCl, 3.5 KCl, 1 NaH2PO4, 24 NaHCO3, 10 D-glucose, 1 CaCl2.2H2O and 3 MgCl2.6H2O until transferred for culture. The low-calcium aCSF was equilibrated with 95% O2, and 5% CO2 with the pH 7.3–7.4 and osmolality 295–305 mOsm/kg. Afterwards, the tumor slices were placed onto membrane inserts (Millipore) onto 6-well plates with 800 μL culture media containing 8.4 g/L MEM Eagle medium, 20% heat-inactivated horse serum, 30 mM HEPES, 13 mM D-glucose, 15 mM NaHCO$_3$, 1 mM ascorbic acid, 2 mM MgSO$_4$· 7H$_2$O, 1 mM CaCl$_2$.4H$_2$O, 0.5 mM GlutaMAX-I, 1 mg/L insulin and 25 U/mL Penicillin/Streptomycin (pH 7.2–7.3). Within 30 min after the brain tumor slices were plated onto membrane inserts, infection with $4 \times 10^6$ PFU of VSVΔ51-RFP was performed by direct application of virus suspension over the tumor slice surface (total volume did not exceed 4 μl to sustain the access of tumor tissues to the air and prevent from overflowing). 4-OI (125 $\mu$M) or vehicle (DMSO) diluted in the slice culture media were added in this model as a co-treatment following infection to reduce the time of incubation in order to maintain the initial tissue characteristics in laboratory conditions.

## Drugs, cytokine, and plasmids

Dimethyl fumarate (DMF) and N-Acetyl-L-Cysteine (L-NAC) were purchased from Sigma-Aldrich. DMF was dissolved in DMSO (Sigma-Aldrich) and L-NAC in water. Interferon-β1A was obtained from Merck/Sigma-Aldrich. All plasmids used in the study were from the Lin laboratory and were previously described in ref. 59.

## Chemical Synthesis

All reactions were conducted in flame-dried glassware under an atmosphere of dry argon unless otherwise stated. CH$_2$Cl$_2$ was dried over activated 4 Å molecular sieves and reagents were used as received from commercial suppliers (Sigma-Aldrich, TCI, and Fluorochem). Concentration in vacuo was performed using a rotary evaporator with the water bath temperature at 40 °C, followed by further

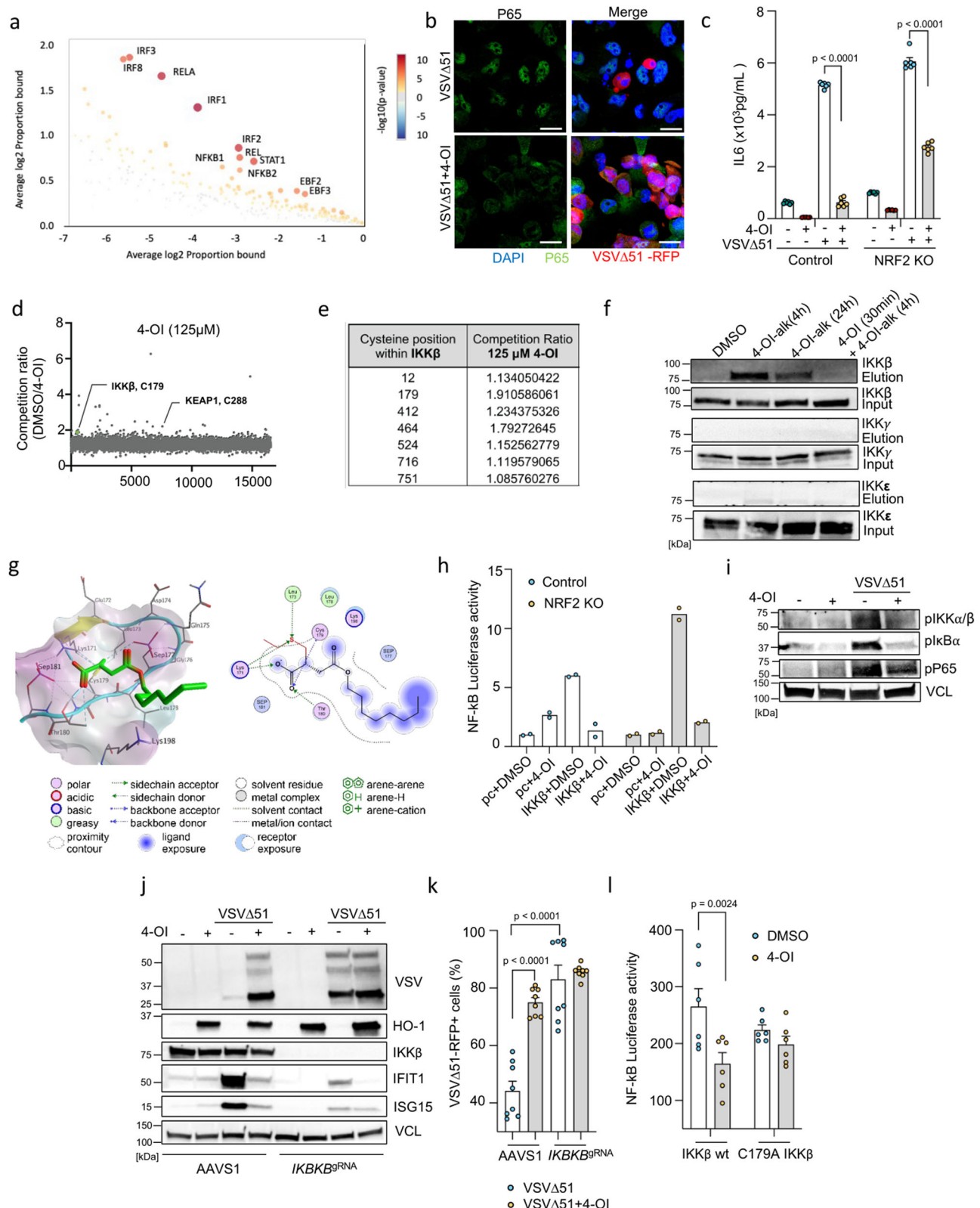

concentration using a high vacuum pump. TLC analysis was carried out on silica-coated aluminum foil plates (Merck Kieselgel 60 F254). The TLC plates were visualized by UV irradiation and/or by staining with KMnO4 stain. Purification was performed by automated flash column chromatography (AFCC) using an Interchim PuriFlash 420 instrument with 30 μm prepacked columns. Infrared spectra (IR) were acquired on a PerkinElmer Spectrum TwoTM UATR. Mass spectra (HRMS) were recorded on a Bruker Daltonics MicrOTOF time-of-flight spectrometer. Nuclear magnetic resonance (NMR) spectra were recorded on a Bruker BioSpin GmbH 400 MHz spectrometer, running at 400 and 101 MHz for $^1$H and $^{13}$C, respectively. The residual peak of the respective solvent was used as the internal standard: DMSO-$d_6$ (CD$_2$HSOCD$_3$ $\delta$H 2.50 ppm, CD$_3$SOCD$_3$ $\delta$C 39.5 ppm); CDCl$_3$ (CHCl$_3$ $\delta$H 7.26 ppm, CHCl$_3$ $\delta$C 77.16 ppm)

**Fig. 8 | 4-OI suppresses NF-κB activation via direct alkylation of IKKβ. a** CiiiDER analysis to identify overrepresented transcription factor binding sites in 786-O cells treated or not with 4-OI (75 μM) prior VSVΔ51 (MOI 0.01 for 17 h). **b, c** Confocal microscopy of p65 nuclear translocation in 786-O cells treated or not with 4-OI following VSVΔ51 (Scale, 20 μm) (**b**); IL-6 levels measured by ELISA (**c**). **d, e** Data from Runtsch et al.[29]. THP1 cells treated with 4-OI following IA-DTB, cell lysis, and LC-MS measurements of modified cysteines. **f** 786-O cells upon 4-OI-alk with or without 4-OI (both 125 μM). Immunoblotting of IKKβ, IKKγ, and IKKε before and after biotin enrichment. **g** Illustration of 4-OI binding (green) to IKKβ at Cys179 (left) and 2D ligand-protein interactions (right). Lipophilicity protein surface: lipophilic (cyan), hydrophilic (violet), neutral (white), α-helices (cyan), β-sheets (yellow), loops (cyan). **h** Luciferase NF-kB promotor activity in control and NRF2 KO 786-O treated or not with 4-OI prior to control (pc) or IKKβ plasmid transfection. **i** Immunoblotting of IKKα/β, IκBα, and P65 phosphorylation in 786-O cells treated

or not with 4-OI prior VSVΔ51. **j, k** IKKβ KO 786-O cells treated or not with 4-OI following VSVΔ51, immunoblotting (**j**) and flow (**k**) analyses. **l** Luciferase assay of NF-κB promotor activity in HEK293 cells stimulated or not with 4-OI prior to IKKβ wt or IKKβ C179A plasmid transfection. Data from one representative experiment in (**a**), two independent experiments in (**b**). Data are depicted as means ± SEM from two experiments in triplicates in (**c**), three independent experiments in (**d, e**). Data from one representative experiment out of three in (**f**). Data are depicted as means from one experiment in duplicates in (**h**). Data from one representative experiment performed twice in (**l, j**). Data are the means ± SEM from two experiments performed in quadruplicates in (**k**). Data are the means ± SEM from two experiments in triplicates in (**l**). Statistical significance by two-way ANOVA for (**c, l**) and one-way ANOVA for (**k**). Vertical stacks of bands are not derived from the same membrane in (**f, i, j**). Source data are provided in Source Data file.

## Synthesis of 4-octyl itaconate (4-OI)
Synthesis was performed as reported previously[25].

## Synthesis of 2-methylene-4-(oct-7-yn-1-yloxy)−4-oxobutanoic acid (4-OI-alk)
Itaconic anhydride (100 mg, 0.89 mmol, 1.0 eq.) was dissolved in anhydrous $CH_2Cl_2$ (0.5 mL). 7-Octyn-1-ol (338 mg, 2.67 mmol, 3.0 eq.) and conc. $H_2SO_4$ (5 drops) were added, and the reaction mixture was stirred for 16 h at room temperature $Et_2O$ (25 mL) was then added to the reaction mixture, and the organic phase was washed with an aqueous solution of $K_2CO_3$ (10 w%, 2 × 10 mL). The aqueous phase was then extracted with $Et_2O$ (2 × 20 mL) to remove the non-ionizable impurities. Conc. HCl was then added to the aqueous phase until pH = 1 and the aqueous phase was then extracted with $CH_2Cl_2$ (2 × 25 mL). The combined organic phases were dried over Na2SO4, filtered, and then concentrated under reduced pressure. The crude product was purified by flash chromatography on a silica gel column using 10−100% EtOAc in heptane as eluent to give **4-OI-alk** (70 mg, 0.30 mmol, 33%) as a white solid.

$R_f$ = 0.50 (Pentane/EtOAc 2:1); UV (254 nm) and $KMnO_4$
HRMS (ESI) m/z calcd for $C_{13}H_{17}O_4$ [M-H]$^-$ 237.1132, found 237.1131
IR $v_{max}$ (cm$^{-1}$) 3294, 2939, 1735, 1698, 1634, 1432, 1160, 960, 635
$^1$H NMR (400 MHz, DMSO-$d_6$) $\delta$ 12.62 (s, 1H), 6.15 (d, $J$ = 1.6 Hz, 1H), 5.76 (d, $J$ = 1.5 Hz, 1H), 4.00 (t, $J$ = 6.6 Hz, 2H), 3.30 (s, 2H), 2.75 (t, $J$ = 2.7 Hz, 1H), 2.15 (td, $J$ = 6.9, 2.7 Hz, 2H), 1.58−1.49 (m, 2H), 1.46−1.39 (m, 2H), 1.43−1.22 (m, 4H).
$^{13}$C NMR (101 MHz, DMSO-$d_6$) $\delta$ 171.0, 167.8, 135.3, 128.4, 85.0, 71.7, 64.5, 37.7, 28.4, 28.3, 28.2, 25.2, 18.1.
The NMR spectra are in agreement with the reported values[42].

## Synthesis of 1-octyl citraconate/4-octyl citraconate (1-OC/4-OC)
To a stirred solution of octanol (3.21 mL, 20.4 mmol, 1.0 eq.) in $CH_2Cl_2$ at 0 °C under an argon atmosphere was added $Et_3N$ (2.99 mL, 21.4 mmol, 1.05 eq.), citraconic anhydride (2.74 mL, 30.6 mmol, 1.5 eq.) and DMAP (0.125 g, 1.02 mmol, 0.05 eq.). On completion of the addition, the solution was warmed to RT and stirred for 30 min. Subsequently, additional amounts of $Et_3N$ (1.56 mL, 11.2 mmol, 0.55 eq.), citraconic anhydride (1.60 mL, 14.3 mmol, 0.7 eq.), and DMAP (0.125 g, 1.02 mmol, 0.05 eq.) were added and the resulting mixture was stirred for 3 h. Upon completion of the reaction, the mixture was partitioned against an aq. solution of HCl (1 M), water, and brine before being dried, filtered and concentrated to afford the crude product. FCC (pentane/EtOAc 9:1) was performed to isolate the pure mono-octyl ester products (2599 mg, 53%, least polar isomer, **1-OC**) and (480 mg, 10%, most polar isomer, **4-OC**).

## Least polar isomer (1-OC)
$R_f$ 0.21 (Pentane/EtOAc 2:1; $KMnO_4$)

$^1$H NMR(500 MHz, CDCl$_3$) $\delta_H$ 11.73 (s, 1H), 5.87 (d, $J$ = 1.9 Hz, 1H), 4.20 (t, $J$ = 6.8 Hz, 2H),2.08 (d, $J$ = 1.8 Hz, 3H), 1.72−1.61 (m, 2H), 1.29 (d, $J$ = 18.0 Hz, 10H), 0.87 (t, $J$ = 6.7 Hz, 3H).
$^{13}$C NMR(101 MHz, CDCl$_3$) $\delta_C$ 169.7, 169.1, 148.1, 120.8, 66.1, 31.9, 29.3, 28.4, 26.0, 22.8,21.1, 14.2.
IR (ATR)$v_{max}$/cm$^{-1}$ 2926, 2856, 1733, 1699, 1652, 1448, 1343, 1281, 1196, 1125.
HRMS (ESI)Calc. for $[C_{20}H_{24}Br_2N_2]^-$ 241.1445 found: 241.1445.

## Most polar isomer (4-OC)
$R_f$ 0.11 (Pentane/EtOAc 2:1; $KMnO_4$)
$^1$H NMR(500 MHz, CDCl$_3$) $\delta_H$ 12.00 (s, 1H), 6.13 (d, $J$ = 1.6 Hz, 1H), 4.18 (t, $J$ = 6.7 Hz, 2H),2.12 (d, $J$ = 1.6 Hz, 3H), 1.71−1.61 (m, 2H), 1.46−1.13 (m, 10H), 1.05−0.70 (t,3H).
$^{13}$C NMR(101 MHz, CDCl$_3$) $\delta_C$ 169.9, 166.9, 145.9, 123.5, 66.3, 31.9, 29.2, 28.4, 25.9, 22.7,21.9, 14.2.
IR (ATR)$v_{max}$/cm$^{-1}$ 2956, 2925, 2856, 1774, 1708, 1651, 1447, 1274, 1192, 1129, 1037.
HRMS (ESI)Calc. for $[C_{13}H_{21}O_4]^-$ 241.1445 found: 241.1445.

## Treatment of cells with itaconate, itaconate derivatives, and L-NAC
Itaconate derivatives were synthesized and kindly provided by the laboratory of Professor Thomas B. Poulsen, Aarhus University, Denmark (see Supplementary Methods section). 4-OI, 4-OC, and 1-OC were prepared at a concentration of 250 mM in DMSO and used at a working concentration of 75 μM, unless otherwise noted. L-NAC was prepared at a concentration of 1 M in water and used as a supplement in the growth media at a working concentration of 1 mM alone or in combination with 4-OI. For all in vitro experiments with itaconate derivatives, DMSO was used as a vehicle control. Itaconic acid (Sigma-Aldrich), mesaconic acid, and citraconic acid were prepared in DMEM media (Sigma-Aldrich) at a concentration of 25 mM, and pH was brought to 7.4 using 1 M NaOH.

## Plasmid mutagenesis
The MAVS and IKKβ point mutations were introduced with a Quick-change II site-directed mutagenesis Kit according to the manufacturer's instructions (Agilent). DNA sequencing was performed to confirm the mutations.

## Virus production, quantification, and infection
The Indiana serotype of VSV used in this study is referred to as wtVSV (wild type). VSVΔ51-expressing RFP, GFP, or eGFP is a recombinant derivative of the VSV Indiana serotype described in ref. 46. VSV WT was kindly provided by Professor Christian K. Holm (Aarhus University, Denmark), VSVΔ51-RFP and VSVΔ51-Luc by Dr. Tommy Alain (University of Ottawa, Canada), and VSVΔ51-eGFP by Dr John Hiscott

(Pasteur Institute, Rome, Italy). Both VSV WT and VSVΔ51 were propagated using Vero cells. In brief, $10 \times 10^6$ cells were seeded in T175 culture flasks and infected the following day at an MOI (multiplicity of infection) of 0.1 in 10 mL of 2% FCS DMEM medium. One hour after infection, the culture medium was increased with an extra 10 mL of DMEM containing 10% FBS medium supplemented with L-glutamine and penicillin/streptomycin. Virus propagation was continued until 24 h post-infection and to isolate the virus, cell culture supernatants were removed from the different flasks, centrifuged at $300 \times g$ for 5 min to remove the cell debris, and viruses were concentrated in Amicon Ultra-15 filter tubes (Sigma) by spinning at $4000 \times g$ for 30 min at 4 °C. The concentrated virus was further aliquoted and stored at −80 °C. The amount of infectious virus particles in the generated stock (PFU/mL) was determined using a standard plaque assay on Vero cells as in[20].

For the preparation of VSVΔ51 for animal experiments, Vero cells were infected at a multiplicity of infection (MOI) of 0.01. After incubation for a maximum of 24 h, when a cytopathic effect (CPE) of 90% was observed, the supernatant was collected and centrifuged at 10,000 rpm for 5 min to remove cellular debris, followed by filtration through a 0.45 µM filter. Subsequently, ultracentrifugation was performed using JLA-10.500 Beckman Coulter rotors to pellet the virus. For virus purification, a continuous gradient was prepared using OptiPrep at concentrations ranging from 10% to 40% in 12 ml tubes. The virus-containing supernatant was resuspended in PBS and carefully layered on top of the gradient. Ultracentrifugation was carried out using SW 41 Ti Beckman Coulter rotors at 36,000 rpm for 1.5 h at 4 °C, resulting in the formation of a virus ring in the middle of the tube. This virus ring was then extracted, aliquoted, and stored at −80 °C before further viral titer determination by plaque assay.

Sindbis virus expressing GFP was a gift from Dr Carolina Ilkow (Ottawa Hospital Research Institute). For experiments, 786-O cells were seeded and treated with DMSO or drug, then infected with Sindbis virus at a MOI of 0.01. Sindbis virus was quantified by standard plaque assay in Vero cells. Plaques were counted 3 days after infection.

Reovirus (Reolysin, Type 3 Dearing) and Vaccinia virus (Jennerex-594, strain Wyeth, Tk-deleted expressing GM-CSF) used in this study were previously reported in ref. [70]. Measles virus was previously reported in ref. [71]. In brief, PANC1 cells were seeded at $8 \times 10^5$ cells per well in a 6-well plate. 24 h later, cells were treated with 125 or 150 µM of 4-OI for 24 h. Cells were then infected with reovirus ($10^6$ TCID50/mL), measles virus (MOI = 0.1), or vaccinia virus (MOI = 0.01) for 48 h. Cells were then collected and freeze-thawed to release intracellular viruses and centrifuged at $1000 \times g$ for 5 min to remove cell debris. Supernatants were used for virus titration using plaque assay on a monolayer of Vero cells or TCID50 assay on a monolayer of L929 cells.

## Flow cytometry analysis of virus-positive and viable cells

The percentage of virus RFP-positive cells was quantified using flow cytometry. Briefly, 786-O or CT26WT cells were seeded in a 12-well plate ($2.5 \times 10^5$/well) and treated the next day with 4-OI (75 µM) for 24 h following an infection with VSVΔ51 at an MOI of 0.01. After one hour of infection, the supernatant was removed to eliminate the unbound virus and replenished with a complete growth medium containing 4-OI with the same concentration. At 17–18 h post-infection, cells were harvested and the LIVE/DEAD Fixable Green Dead Cell Stain kit with 488 nm excitation (Invitrogen) was used to determine the viability prior to fixation with 4% PFA for 30 min at room temperature. Flow cytometry analysis was performed using a NovoCyte Quanteon instrument, and data were analyzed using NovoExpress Software (v1.6.2). The gating strategy on virus-infected RFP+ cells and viable/dead cells is described in Fig. S18a, b.

## Flow cytometry analysis of LDLR surface protein level

The expression of LDLR upon escalating 4-OI stimulation (75–125 µM) was analyzed using flow cytometry. Briefly, $1.25 \times 10^5$ cells were washed with a staining buffer (PBS supplemented with 2% heat-inactivated FCS) and antibodies at a dilution of 1:80 were applied in staining buffer for 60 min at 4 °C in the dark (Mouse Anti-Human-LDLR Antibody, Clone C7, BV421-conjugated (BD Bioscience, Cat# BD744847) or Mouse IgG2 kappa Isotype Control BV421-conjugated, both from (BD Bioscience, Cat# BD569376)). Cells were then washed twice, and fixed using 1% paraformaldehyde, pH 7.4 (Morphisto). The fluorescence intensity was measured with a NovoCyte Quanteon flow cytometer and analyzed using FlowJo software (v10). Samples stained with the isotype control of the matched antibody were used as a control, and the background MFI obtained was subtracted from the MFI of the separate samples. The gating strategy on LDLR+ cells is described in Fig. S8a.

## Calcein green viability staining

To visualize viable cells by fluorescence microscopy staining with acetoxymethyl precursor of calcein (ViaStain Calcein AM, Nexcelom Bioscience) was performed. Calcein, a dye capable of permeating cell membranes, can be introduced into cells through incubation. Following the entry into the cells, endogenous esterase hydrolyzes calcein, transforming it into the intensely negatively charged green fluorescent calcein, which is exclusively retained in the cytoplasm of viable cells. 4× staining solution prepared in PBS was added to the culture media of wells containing cells or organoids and incubated for 30 min in the dark at 37 °C.

## Fluorescent plaque assay measurements

786-O cells were pretreated with increasing concentrations of 4-OI for 24 h and infected with VSVΔ51 (MOI, 0.0001). After 1 h of infection, 1× Hoechst 33342 (Bio-Rad) dissolved in RPMI was applied for 15 min at 37 °C to visualize nuclei. Then, the cells were rinsed with prewarmed DPBS and 1% Methyl cellulose overlay was added. Fluorescence microscopy of representative plaques was done and measured with EVOS M5000 Imaging System (Invitrogen) at 24 h after infection. Afterwards, plaques were fixed with 2% PFA and confirmed by staining with 1% Crystal Violet.

## Viral infection of colon tumoroids/organoids

For infection with VSVΔ51, organoids were seeded onto the BME layer and infected with $10^6$ PFU. For that, 48-well suspension culture plates (Greiner Bio-One) were moistened using culture medium and each well was evenly covered with 35 µL of undiluted BME and left overnight at room temperature to let it solidify. Confluent organoids were collected, pelleted, washed from old BME, and each confluent dome was resuspended in 500 µL of the respective culture medium but without N-acetyl-cysteine. This 500 µL of organoid suspension was carefully added to the center of the BME-covered well. Organoids were grown for 24 h before the addition of 4-OI (125 µM) and incubated for another 24 h before VSVΔ51 infection. On the day of infection, the media was carefully replaced with fresh media containing either DMSO (control), 4-OI alone (125 µM), VSVΔ51 ($10^6$ PFU) plus DMSO, or VSVΔ51 ($10^6$ PFU) plus 4-OI at the same concentration.

## GFP and luciferase-expressing tumor colon organoids

Tumor organoids labeled with GFP and luciferase reporter gene luc2 from plasmid pGL4.10[luc2] (Promega) were established by lentiviral transduction as described in ref. [72]. To quantify the luciferase signal, the supernatants were carefully removed at 96 h post-infection with VSVΔ51 and 500 µL of luciferase assay reagent (10% One-Glo Ex Reagent (Promega), 1% Triton X-100, 50 mM NaCl, 50 mM Tris-HCl pH 7.4) was added to each well and incubated for 60 min at room temperature in the dark. Lysates were resuspended and 100 µL from each well was transferred into an opaque 96-well microtiter plate and measured with luminescence microplate reader SpectraMax iD3 (Molecular Devices). Survival was defined as the difference between luciferase activity of organoids cultured alone (taken as 100%) to co-

culture either with 4-OI (125 μM), or VSVΔ51 (10⁶ PFU), or VSVΔ51 in combination with 4-OI at the same PFU and concentration.

## GFP and RFP detection within tumor colon organoids

Tumor colon organoids expressing GFP-luc2 were infected and treated as described before. At 48 h of post-infection with VSVΔ51-RFP, GFP and RFP fluorescence was visualized with BioTek Cytation C10 confocal imaging reader (Aglient) at 2.5X magnification. Images were stitched with Gen5 software to have an overview of the whole well in a 48-well plate. GFP/RFP fluorescence areas were quantified using the ImageJ software and the percentage of infectivity was calculated as the ratio of GFP-positive (total organoid area) to RFP-positive (infected) area. Organoids that were not transduced with pGL4.10[luc2] were stained with Calcein green (ViaStain Calcein AM, Nexcelom Bioscience) to quantify the initial GFP-organoid area. To assess the robustness of the method, the percentage of infectivity with VSVΔ51 in sensitive versus resistant to the infection tumor organoids was measured by microscopy and further processed by flow cytometry and compared side-by-side.

## Flow cytometry quantification of VSVΔ51 infection in tumor colon organoids

The percentage of VSVΔ51-RFP-positive cells within colon organoids was quantified using flow cytometry at 48 h of post-infection. Organoids were mechanically disrupted by pipetting, washed with PBS, and centrifuged for 5 min at 1200 rpm 4 °C. Then, the supernatant was removed and 1 mL of StemPro Accutase (Gibco) was added and incubation for 5 min at 37 °C with gentle vortexing was performed. Then, fresh DMEM media was quickly added and enzymatically digested organoids were centrifuged again for 5 min at 1500 rpm 4 °C. After one more wash in PBS, the pellet was resuspended in 4% PFA in PBS and filtered once or twice through 40 μm cell strainer using round bottom tubes with cell strainer cap (Stem cell Technologies). Flow cytometry counts were performed using BD LSRFortessa cell analyzer (BD Biosciences) and were analyzed with BD FACSDiva Software (v8.0.1).

## Ex vivo murine tumor core model

Female, 8-week-old mice were implanted subcutaneously in the right flank with $3 \times 10^5$ CT26WT in BALB/c mice or 76-9 cells in C57BL/6 mice. Upon reaching a tumor volume of 1000 mm³, mice were culled, and the tumors were extracted. In a sterile ex vivo environment, tumors were cut into 2 mm slices, and 2 mm diameter cores were extracted from the slices using a punch biopsy tool. Cores were maintained in a humidified incubator at 37 °C, 5% CO₂ in DMEM supplemented with 10% serum, 30 mM HEPES, 1% (v/v) penicillin-streptomycin and 0.25 mg/L amphotericin B. Cores were treated with 4-OI at indicated concentrations for 4 h, then infected with VSVΔ51-GFP (3e4 pfu/core). Fluorescence images were taken 24 hpi and supernatants were collected at 48 hpi and stored at −80 °C until viral titer determination by plaque assay.

## In vivo tumor model

Five- to six-week-old female BALB/c mice were purchased from Charles River (Kingston, New York, USA). The mice were housed at 5 animals per individually ventilated cage and had *ad libitum* access to water and chow (standard 18% protein rodent diet (Envigo: cat#:T.2018.15)). The ambient temperature was 21–23 °C and mice were undergoing a 12 h classical light:dark cycle. Mice were acclimated for one week before any manipulation. For tumor implantation, 10⁵ CT26WT cells resuspended in PBS were injected subcutaneously into the flank of animals. Experiments were performed in accordance with the University of Ottawa Animal Care and Veterinary Service guidelines for animal care under the protocol CHEOe-3084-R2 A1.

Five- to six-week-old female BALB/c mice were purchased from Janvier Labs (France). Animal facility conditions were as follows: 12-h

dark:light cycle with a temperature of 20–24 °C and relative humidity of 55% ± 10%. Animals were housed 4–6 per cage in individually ventilated cages (Tecniplast) prepared with wood chip bedding plus enrichment, including peanuts. Mice had *ad libitum* access to reverse osmosis chlorinated water and standard chow (Altromin, Lage, Germany, cat# 1328). Animal welfare was monitored daily by staff according to FELASA guidelines. Mice were acclimated for one week before any manipulation. For tumor implantation, 10⁵ CT26WT cells resuspended in PBS were injected subcutaneously into the flank of animals. Experiments were performed at Aarhus University, Department of Biomedicine in accordance with the license no. 2023-15-0201-01489 approved by the Danish Experimental Animal Expectorate.

In our survival and tumor growth studies, we chose to work exclusively with female subjects due to practical considerations surrounding animal handling and housing logistics. Females allowed for more efficient use of cage space, as we could comfortably house up to five individuals together, unlike males, who typically require individual cages due to territorial behaviors. We believed that minimizing stressors associated with isolation would lead to more reliable results and ensure the welfare of our experimental subjects.

## In vivo VSVΔ51 replication study

This part of the work was done at the University of Ottawa under the protocol CHEOe-3084-R2 A1. Two weeks after tumor implantation, mice were treated intratumorally with either 4-OI (25 mg/kg/dose) in 40% hydroxy-propyl-β-cyclodextrin (CDX) in PBS or only 40% CDX in PBS. After 24 h, 10⁸ PFU per tumor of VSVΔ51-Luc in 50 μL PBS was injected intratumorally. For visualizing luciferase expression from virus replication, D-Luciferin (GOLDBIO, CAT#: LUCK-1G) was prepared at 15 mg/mL in DPBS and injected intraperitoneally at 10 μL/g of body weight. Images were acquired after 5–8 min of luciferin injection. A series of 15 images, 1 min apart from each image, and 30 s of acquisition time were taken to determine the peak signal. Images were analyzed using Living Image® software (for IVIS® Spectrum images) V4.7. Following imaging, animals were sacrificed, and tumors were collected to titrate for intratumoral virus by plaque assay. Briefly, excised tumors were homogenized in PBS buffer with 2.0 mm zirconia beads (Thomas Scientific, CAT# 1197P96) and shaken at 20 Hz with a TissueLyzer II (QIAGEN). Lysates were centrifuged at $1000 \times g$ for 5 min to pellet the cell debris, and the supernatant was used for virus titration using plaque assay on a monolayer of Vero cells.

## Survival and tumor growth studies (Fig. 2)

To increase data reproducibility, survival and tumor growth studies in Fig. 2 were performed at two different locations (Ottawa University and Aarhus University). The data displayed in Fig. 2i, j resulted from pooled data of two experiments performed at different locations. Experimental details are appended below.

Experiment done at the University of Ottawa under the protocol CHEOe-3084-R2 A1. When implanted CT26WT tumors reached around 70–80 mm³, one intratumoral injection of 4-OI (25 mg/kg/dose) in 40% CDX in PBS or only CDX was performed every second day for a total of 2 injections (days 1 and 3). Injections of 10⁸ PFU of VSVΔ51-Luc in 50 μL PBS or PBS (vehicle) were performed one day after each 4-OI/vehicle injection (days 2 and 4). Tumor sizes were measured at the indicated time points using an electronic caliper, and tumor volume was calculated as length × (width)²/2. Mice were euthanized when tumor volume reached around 1500 mm³ or any other alternative humane endpoint, such as tumor ulceration. The maximal accepted size burden was not exceeded in the experiments.

Experiment done at Aarhus University, Department of Biomedicine in accordance with the protocol 2023-15-0201-01489 approved by the Danish Experimental Animal Expectorate. When implanted CT26WT tumors reached around 30–40 mm³, one injection of 4-OI (25 mg/kg/dose) in 40% CDX in PBS or only CDX was performed every

second day for a total of 2 injections (days 1 and 3). Injections of $5 \times 10^7$ PFU VSVΔ51-RFP in 50 μL PBS or PBS were performed one day after each 4-OI/vehicle injection (days 2 and 4). Tumors sizes were measured at the indicated time points using an electronic caliper, and tumor volume was calculated as length × (width)$^2$/2. Mice were euthanized when tumor volume reached around 1000 mm$^3$ or any other alternative humane endpoint, such as tumor ulceration. The maximal size burden was not exceeded in the experiments.

## Tumor rechallenge experiment

Experiment done at the University of Ottawa under the protocol CHEOe-3084-R2 A1. Cured mice from co-treatment experiment and 5 naive eleven-week-old female BALB/c mice were rechallenged with 10$^5$ CT26WT cells. The cells were injected on the opposite flank side from the original CT26WT tumor site. Tumor growth was monitored with calipers. Mice were euthanized when tumor volume reached 1500 mm$^3$ or any other alternative humane endpoint, such as deep tumor ulceration. The maximal size burden was not exceeded in the experiments

## Animal experiments (Fig. 6)

Experiment was performed in accordance with the institutional Animal Welfare Body of the LUMC and carried out under project license AVD1160020187004, issued by the competent authority on animal experiments in The Netherlands (named CCD). The experiment was performed following the Dutch Act on Animal Experimentation and EU Directive 2010/63/EU ("On the protection of animals used for scientific purposes") at the animal facility. Six- to eight-week-old male BALB/cByJ mice were purchased from Charles River (France). Mice were housed at 3 per individually ventilated cages and fed *ad libitum* with water and pelleted chow (Special Diets Services, RM3 (P), Cat# 801700). Animal facility conditions were as follows: 6.30–7 sunrise; 7–18 – daytime; 18–18.30 sunset; Temperature: 20–22 °C; Humidity 55%. Animals were acclimated for one week before any manipulation. Group size was calculated using the PS: Power and Sample Size Calculation program (Vanderbilt University, V.3.1.6). The calculated group size was 6 mice per group. However, unfortunately, we had to exclude three mice in the CDX/PBS group and one mouse in CDX-4-OI/PBS group before the start of treatment because of reaching humane endpoints. In this experiment, we chose to work exclusively with male subjects for practical/ethical reasons. Due to the stronger skin of male subjects, we expected fewer occasions of tumors breaking to the skin or ulcerations that led to the exclusion of mice from the experiment due to reaching humane endpoints, and hence reduced power.

For tumor implantation, 10$^5$ CT26WT cells resuspended in PBS/0.1% BSA were injected subcutaneously into the flank of animals. Animal welfare and tumor growth were assessed three times a week and tumor growth was measured using a caliper in a blinded manner concerning experimental treatment. Tumor volume was calculated as width × length × height. Cages were divided into groups based on average tumor and sample size. Three mice in the PBS group had to be excluded from the experiment due to severe discomfort before starting treatment. Treatment was started at the moment of palpable tumors and intratumoral injections were performed under isoflurane anesthesia. Every second day the mice received intratumoral injections with 40% cyclodextrin (CDX) in PBS or 4-OI in CDX in 50 μl on day 0 and day 2 followed by PBS or $1 \times 10^8$ PFU of VSVΔ51-eGFP in 50 μl PBS on day 1 and 3. The maximum accepted burden was 1500 mm$^3$, according to the experimental protocol. Other humane endpoints were stated as diarrhea (due to tumor model), loss of 20% body weight, Body Condition Score of 2 or lower, signs of pain (scale 2) according to the Grimace Scale for mice, skin irritation (redness, crusts) before tumor will break through the skin to avoid wounds. When humane endpoints were reached before the experimental endpoint, the mice were sacrificed and excluded from the analyses. This happened to

three mice, all in the CDX-PBS group. The maximal size burden was never exceeded in the experiments.

At the dedicated timepoint (day 8), the spleen, tumor-draining lymph node (T-DLN) and tumor were collected for flow cytometry analysis and a representative part of the tumor was snap-frozen for RNA analysis. Organs were dissociated into single-cell suspension as described in ref. 73. Cells were incubated with Zombie Aqua Fixable Viability dye (Biolegend) in PBS for 20 min followed by incubation with 2.4G2 FcR blocking antibodies (clone 2.4G2, BD Biosciences) in FACS buffer (PBS, 0.5% BSA and 1% sodium azide) for 20 min on ice followed by incubation with a mix of conjugated antibodies (Table S4) in FACS buffer for 30 min on ice in the dark. If necessary, cells were fixed and stained for nuclear proteins using the Foxp3 / Transcription Factor Staining Buffer Set (eBiosciences) according to manufacturers' instructions. After completing the staining protocol, samples were fixed in 1% paraformaldehyde and measured using a BD LSRFortessa X20 cell analyzer (BD Biosciences) at the Flow cytometry Core Facility (FCF) of Leiden University Medical Center (LUMC) in Leiden, Netherlands. Samples were excluded from analysis if no viable cells were acquired (indicated in figure legends). Four markers had to be excluded from analysis due to low signal, which may be due to aberrant expression in BALB/CByJ hosts compared to C57Bl/6. Flow cytometry data was analyzed using FlowJo™ Software (v10) (Becton, Dickinson and Company). Opt-SNE plots were generated using standard settings in OMIQ data analysis software (Omiq, Inc. www.omiq.ai)[74]. The gating strategy is described in Fig. S11a.

Snap-frozen bulk tumors were disrupted using stainless steel beads in the TissueLyser LT system (Qiagen) and total RNA was isolated using ReliaPrep RNA Tissue Miniprep System (Promega) according to the manufacturer's protocol. RNA quantity and quality were assessed using Nanodrop followed by generating cDNA using High-Capacity cDNA Reverse Transcription Kit (ThermoFisher Scientific, #4368814) according to the manufacturer's protocol. Two samples had to be excluded due to insufficient material (indicated in figure legends). Subsequent qPCR analysis was performed using the Bio-Rad iQ™SYBR® Green Supermix on a CFX384 Real-Time System machine (Bio-Rad). The used PCR program consisted of the following steps: (1) 3 min at 95 °C; (2) 40 cycles of 10 s at 96 °C followed by 30 s at 60 °C and plate read; (3) 10 s at 95 °C; (4) Melt curve 65–95 °C with an increment of 0.2 °C every 10 s, and plate read. The expression of target genes (Table S5) was normalized for reference genes Mzt2 and Ptp4a2 and relative expression was calculated using the 2-ΔΔCT method in the Bio-Rad CFX Maestro Software program (Bio-Rad) after normalizing for multiple runs.

## qPCR analysis

Gene expression was determined by real-time qPCR, using TaqMan detection systems (Applied Biosciences). RNA was extracted using the high pure RNA isolation kit (Roche, 11828665001) according to the manufacturer's instructions with RNA being eluted in 60 μL of Elution buffer from the kit. The RNA quantity and quality were assessed using a NanoDrop One (Thermo Scientific). mRNA levels were analyzed using the TaqMan RNA-to-Ct-1-Step kit according to the manufacturer's recommendations (Applied Biosciences, Thermo Fischer Scientific, cat. no. 4392938). We used the following commercially available predesigned gene expression assays: TBP Hs00427620, *HMOX1* Hs01110250, *IFIT1* Hs01675197, *AKR1B10* Hs00252524, and *ISG15* Hs00192713. For the VSV L gene (GeneBank accession number: J02428.1), primer and probe sequences were the following (purchased from LGC Biosearch): fw primer 5′-CGGGCTTGCACAGTTCTAC-3′, rev primer 5′-ATGCCCGACACTCCTCCAATG-3′, probe (FAM-CGCCATGTTG TATTTGGACC-BGH). Samples were analyzed with 250 nM of each primer and 200 nM of the probe, using the TaqMan kit described above. The analysis was performed on a QuantStudio3 Real-Time qPCR

system (Thermo Fischer Scientific) with program: 2′50 °C; 2′95 °C; 40× (1″95 °C; 20″60 °C). Ct values were extracted using the Thermo Fischer Connect Software.

### qPCR from Vaccinia virus, Reovirus, and Measles virus-infected cells

786-O cells were seeded in a 12-well plate (2.5 × 10⁵ cells/well) and treated the next day with 4-OI (125 μM) for 24 h. Cells were then infected with Vaccinia virus (MOI = 0.01), measles virus (MOI = 0.1) for 24 h, and reovirus (MOI = 1) for 48 h. Total RNA was extracted using QIAzol (QiaGen) and relative gene expression was quantified using RT-qPCR. Sequences of all primers used were listed in Table S6.

### qPCR from ex vivo brain tumor slices

After verification of the fluorescent signal from VSVΔ51-RFP in the ex vivo tumor slices, determined within two days post-infection using an EVOS M5000 (Thermo Fisher Scientific), samples were processed for RNA extraction. Briefly, tumor slices were carefully transferred using sterile forceps into a 2 mL Eppendorf tube containing 1 mL of lysis reagent (Qiagen, QIAzol Lysis Reagent) and a 5 mm stainless steel bead (Qiagen). Following homogenization of the tumor slices using the TissueLyser II (Qiagen) at 20 Hz for 3 min, RNA extraction was performed using the RNeasy Lipid Tissue Mini Kit (Qiagen) in accordance with the manufacturer's instructions. Subsequently, a qPCR analysis was conducted as described in "qPCR analysis" section.

### Sample preparation for metabolite profiling

786-O cells were seeded in 6-well plates (5 × 10⁵) and treated the next day with DMSO (Mock control), 4-OI (75 μM) or itaconic acid (10 mM). After 24-h incubation, cells were infected with VSVΔ51-RFP at an MOI of 0.01 for 1 h and afterwards virus-containing media was replaced with fresh complete RPMI containing the same concentration of the respective drugs. At 17 h post-infection, cells were washed once with 1 mL of ice-cold PBS, scraped from the surface with an S-size cell scraper (Starsted) in 2 mL ice-cold PBS, and transferred to falcon tubes. After spinning at 4 °C (300 G, 5 min), supernatant was removed and cells were resuspended with 800 μL of ice-cold methanol and incubated for 20 min at −20 °C. The procedure for further metabolite profiling is described in Supplementary Methods section.

### M8 RIG-I agonist stimulation of cells

786-O cells were pretreated for 24 h with 4-OI (75 μM) and subsequently transfected using Lipofectamine RNAiMAX transfecting reagent (Invitrogen) at a concentration of 2 μL/mL with a sequence-optimized RIG-I agonist M8[47] for 5 h at a concentration of 3.5 ng/mL. The antiviral and inflammatory levels were assessed by immunoblotting as described further.

### Protein extraction and denaturing gel electrophoresis

786-O cells (5 × 10⁵) were seeded into 6-well plate and treated as desired. Afterwards, each well was pelleted and lysed in 80 μL of ice-cold Pierce RIPA lysis buffer (Thermo Scientific) supplemented with 10 mM NaF, 1× protease inhibitors (Roche), and 5 IU/mL benzonaze (Sigma), respectively. The protein concentration was determined and equilibrated using a Pierce BCA protein assay kit (Thermo Scientific). 25 μg of whole-cell lysates was denatured for 5 min at 95 °C in the presence of 1× XT sample buffer and 1× XT reducing agent (both Bio-Rad). Separation of samples was performed by SDS-PAGE on 4–20% Criterion TGX precast midi protein gels (Bio-Rad) at 75 V until the dye front reached the separating gel and then at 120 V until it reached the bottom of the gel. Precision Plus Dual Color protein ladder (Bio-Rad) was used as separation control. Proteins exceeding a molecular weight of 200 kDa (such as TET2) were separated using 4–8% Criterion XT

Tris-acetate precast midi protein gels in the absence of SDS, employing a Tris-acetate buffer system at pH 7.0 (Bio-Rad).

### Immunoblotting

The immunoblotting onto PVDF membranes (1704157, Bio-Rad) was done using a Trans-Blot Turbo transfer for 7 min at 25 V. In case of high molecular weight proteins, the transfer was done for 10 min at 25 V. Membranes were blocked for at least 1 h with 5% BSA (A7906, Sigma-Aldrich) in PBS-T (PBS supplemented with 0.05% Tween-20). Then, membranes were fractionated according to the proteins of interest size and probed overnight at 4 °C with any of the following specific primary antibodies diluted 1:1000 in PBS-T (the lower dilutions for some antibodies are specified): VSV antisera (a gift from Dr. Jean-Simon Diallo, 1:10,000), anti-AKR1B10 (SC-365689, Santa Cruz), anti-Cleaved PARP (5625, Cell Signaling), anti-GAPDH (sc-47724, Santa Cruz), anti-HO1 (86806, Cell Signaling), anti-IFIT1 (14769, Cell Signaling), anti-IRF3 (11904, Cell Signaling), anti-IKKβ (2684, Cell Signaling), anti-ISG15 (2758, Cell Signaling), anti-Jak1 (29261, Cell Signaling), anti-KEAP1 (8047, Cell Signaling), anti-MAVS (3993, Cell Signaling), anti-NRF2 (ab62352, Abcam), anti-NF-κB p65 (8242, Cell Signaling), anti-P-IRF3 Ser396 (29047, Cell Signaling), anti-P-STAT1 Tyr701 (7649, Cell Signaling), anti-P-IKKα/β (2697, Cell Signaling), anti-P-IκBa (2859, Cell Signaling), anti-P-NF-κB p65 Ser536 (3033, Cell Signaling), anti-DYKDDDDK Tag (D6W5B) (FLAG) (14793, Cell Signaling), anti-TET2 (18950, Cell Signaling), anti-ATF3 (33593, Cell Signaling), anti-IκB-zeta (9244, Cell Signaling), anti-IKKγ (2685, Cell Signaling), anti-IKKε (2905, Cell Signaling), and anti-Vinculin (V9131, Sigma-Aldrich 1:10,000) used as a loading control. Membranes were then washed in PBS-T and incubated for 1 h in the corresponding secondary antibodies, peroxidase-conjugated F(ab)2 donkey anti-mouse IgG (H+L) (715-036-150, Jackson ImmunoResearch 1:10,000) or peroxide-seconjugated F(ab)2 donkey anti-rabbit IgG (H+L) (711-035-152, Jackson ImmunoResearch, 1:10,000) in PBS-T containing 1% skim milk powder (70166, Sigma-Aldrich). Blots were developed by adding 1:1 either the SuperSignal West Pico PLUS chemiluminescent substrate (34577, Thermo Scientific) or the SuperSignal West Femto maximum sensitivity substrate (34095, Thermo Scientific) in an iBright CL1500 Imager (Invitrogen). Vertical stacks of bands are not derived from the same membrane; each membrane was divided into smaller fragments and each piece of membranes incubated overnight at 4 °C with one of the following primary antibodies. Some of the membranes were extensively washed/stripped and blotted using another antibody. Uncropped images of the western blots and overall procedure are provided in Supplementary Fig. 19.

### Semi-native WB dimerization

IRF3 dimerization was assessed under semi-native conditions as described previously[25,34]. Cells were lysed as mentioned previously for protein extraction. Then, protein concentration was quantified and equilibrated following resuspension in 1× XT Sample Buffer (Bio-Rad). Samples were neither reduced nor heated before separation and immunoblotting was done as described in the section above.

### Quantitative profiling of 4-OI modified cysteinome in THP1 cells

The cysteinome in 4-OI treated THP1 cells was assessed as previously described[29]. In brief, THP1 cells were seeded into a 6 cm petri dish (4 million per dish) in 5 ml RPMI-1640 culture medium, and treated with DMSO or 125 μM 4-OI for 16 h at 37 °C and 5% CO₂. Ten dishes of cells were treated (4 dishes with DMSO, 3 with 125 μM 4-OI, and 3 with 250 μM 4-OI) in each replicate experiment for utilizing TMT-10plex isobaric label reagent (Thermo Fisher Scientific)[29]. Carbamidomethylation of cysteine residues was set as fixed modification, while oxidation of methionine residues, acetylation of protein N-term, and IA-DTB on cysteine residues were set as variable modifications. The competition ratio was calculated by dividing the TMT reporter

ion intensities from the control channel (DMSO) by the 4-OI treated channel.

### 4-octyl-itaconate alkyne (4-OI-alk) affinity pull-down assay

786-O WT cells were grown to 70% confluence in T75 (middle) flasks and treated with DMSO (negative control) or 125 μM 4-OI-alk for 4 and 24 h respectively. In the "competition" flask, 4-OI was applied first for 30 min followed by 4-OI-alk for 4 h. Then, cells were washed twice with ice-cold PBS, harvested using cell scrapers, and centrifuged at 400 g for 5 min at 4 °C. Cell pellets were resuspended in 200 μL lysis buffer (1% Triton X-100, 150 mM NaCl, 50 mM triethanolamine) and incubated 10 min on ice followed by centrifugation for 5 min (16,000 × g, 4 °C) to remove cell debris. Protein concentration was determined using a BCA protein assay kit and adjusted to 2 mg/mL with 2–4% SDS in PBS (aiming for the final concentration of 1% SDS). Afterwards, samples containing 300 μL of lysate (600 μg protein) were prepared. To each sample was then added 9 μL 50 mM CuSO$_4$ (Sigma-Aldrich, dissolved in MQ-H$_2$O), 21 μL 100 mM Tris(3-hydroxypropyltriazolylmethyl)amine (THPTA) ligand (TCI Chemicals, dissolved in MQ-H$_2$O), 9 μL 10 mM biotin azide (Sigma-Aldrich, dissolved in DMSO) and 15 μL 100 mM Sodium Ascorbate (Sigma-Aldrich, dissolved in MQ-H$_2$O), followed by incubation for 1 h at room temperature. The proteins were then precipitated with 2,7 mL methanol overnight at −20 °C. The next day, proteins were pelleted (7000 × g, 5 min, 4 °C), washed through resuspension in 1 mL 9:1 MeOH:MQ-H2O followed by re-pelleting, and finally redissolved in 100 μL PBS containing 0.5% SDS.

At this point, 20 μL (20%) of the protein solution was saved for the input control. The solutions were then incubated with 100 μL of a slurry of pre-washed streptavidin Dynabead in PBS containing 0.01% BSA (Invitrogen) for 2 h at room temperature with rotation. The streptavidin beads were washed with PBS-T six times, mixed with 1× XT sample buffer and 1× XT reducing agent (Bio-Rad) following heating to 95 °C for 5 min. Input and elution samples were resolved on 4–20% SDS-PAGE gel as described before. KEAP1, IKKβ, IKKγ, IKKε, MAVS, and JAK1 proteins were detected via western blot using antibodies listed in the Immunoblotting section (all from Cell Signaling Technologies).

To observe the enrichment of 4-OI alkyne labeling in MAVS wild-type protein compared to the C283A mutant MAVS, HEK293T cells were transfected with corresponding plasmids from "Plasmid mutagenesis" section. Briefly, HEK293T WT cells were grown to 70% confluency in T75 flask when pre-incubated for 30 min at room temperature mixture of X-tremeGENE TM HP DNA Transfection Reagent (Roche) (3 μL per 1 mL) with plasmids at a concentration of 1 μg per 1 mL was added in 500 μL of OptiMEM (Gibco). At 24 h post-transfection time, DMSO or 4-OI-alkyne probe were applied for 4 h to the flasks transfected either with Flag-MAVS wild type or Flag-MAVS C283A mutant plasmids. The remaining procedure was done as above, Flag tag was detected via western blot analysis, and ImageJ software was used to quantify all protein bands where TIFF images were converted to 8-bit format and pixel density was measured. Input controls were used for normalization to equalize the protein loading. The Precision Plus Protein Western C ladder (Bio-Rad) served as the separation control due to its lower chemoluminescence activity observed consistently during the blot development process.

### Confocal microscopy

786-O or CT26WT cells were seeded onto glass coverslips placed on the bottom of 12-well plates and treated with either DMSO (0.1%) or 4-OI (75 or 125 μM, respectively) for 24 h prior to infection with VSVΔ51 (MOI of 0.01). After 24 or 48 h, the cells were washed with PBS and fixed for 40 min in 4% paraformaldehyde (PFA, 11762 Morphisto), following 20-min permeabilization with 0.2% Triton X-100 in PBS. Next, blocking with 2% goat serum (G6767, Sigma) in PBS was performed for at least 40 min and a mouse VSV glycoprotein antibody (SAB4200695,

Sigma, 1:200) and a rabbit-cleaved caspase3 antibody ((Asp175) 5A1E, Cell Signaling, 1:400) or rabbit IFIT1 antibody (14769, Cell Signaling, 1:800) were applied for 1 h at room temperature in the blocking solution. To show NRF2 and P65 translocation into the nuclei rabbit NRF2 antibody (12721, Cell Signaling, 1:200) or rabbit-cleaved P65 antibody (8242, Cell Signaling, 1:200) were applied respectively. After three washes with PBS, cells were incubated with a goat anti-rabbit Alexa Fluor 488 nm fluorophore-conjugated secondary antibody (A11008, Invitrogen, 1:400), a goat anti-mouse Alexa Fluor 555 nm fluorophore-conjugated secondary antibody (A21424, Invitrogen, 1:400), Alexa Fluor Plus 647 Phalloidin (A30107, Invitrogen, 1:400), and PureBlu DAPI nuclear staining dye (1351303, Bio-Rad, 1:100) for 1 h at room temperature in the dark. Cells were then washed three times with PBS and mounted onto microscope slides using a ProLong gold anti-fade mounting medium (P36934, Invitrogen). Slides were air-dried in the dark and examined on the next day using a Zeiss LSM 710 inverted confocal microscope. Imaging was acquired using ZEN Black edition (v2.3 SP) and image analysis was done with Image J 1.53t Java 1.8.9_322 (64 bit). For P65 area quantification within a nuclear fraction, ImageJ software was applied. First, the mask (selection) was generated around the nuclei and then applied over the channel used to detect the P65 signal. Afterwards, the P65 area (% area) was quantified within the selected nuclear fraction.

### RNAseq and data analysis

RNA sequencing was performed in collaboration with BGI Europe Genome Center (Copenhagen, Denmark) following the standard operational procedures as described in[75]. Agilent 2100 Bioanalyzer (Agilent Technologies, Santa Clara, USA) confirmed the quality of RNA. Library construction, sequencing, and initial data filtering including adapter removal were performed by BGI Europe Genome Center. Total RNA was subjected to oligo dT-based mRNA enrichment. 100 bp paired-end read sequencing was performed on the DNBseq platform with more than 20 million clean reads obtained per sample. Reads were aligned to, respectively, the human genome (GRCh38 – Ensemble release 108) and the VSV genome (GenBank assembly accession: GCA_000850045.1) using HISAT2 aligner (v2.1.1)[76]. Transcript quantification was performed using StringTie (v2.2.1)[77] and the read counts were normalized for effective gene length, and sequencing depth to yield Transcripts Per Kilobase Million (TPM). Differentially expressed genes were determined from count tables using EdgeR (v3.36.0)[78] and reported after Benjamini–Hochberg false discovery rate (FDR) (5%) correction. Functional gene set analyses were performed with FDR Differentially expressed Gene (DEG) gene list as input using FUMA[79] and Hallmark gene set reported. Enrichment analysis of Transcription Factor Binding Sites (TFBSs) was carried out according to Gearing et al. using CiiiDER[80]. Briefly, promotor sequences (2000 bp upstream of Transcription Start Site (TSS) were extracted from the Homo sapiens GRCh38 – Ensemble release 108) genome file. Identification of TFBSs in these sequences was performed with JASPAR2020_CORE_vertebrates transcription factor position frequency matrices (downloaded from https://jaspar.genereg.net/) and a deficit cut-off of 0.15. CiiiDER enrichment analysis of overrepresented NR TFBSs in DEG query sequences compared to non-DEG query sequences (from 5000 genes with $p \sim 1$ and logFC$\sim$0) was determined by comparing the number of sequences with predicted TFBSs to the number of those without, using a Fisher's exact test.

### Immunoblot analysis following transfection with expression plasmids

HEK293 cells were grown in DMEM media (Wisent) supplemented with 10% (vol/vol) FBS, L-glutamine, and antibiotics (1%, Wisent, Cat#450201-EL,). Cells were transfected with 200 ng of GFP, GFP-RIG-IN, GFP-MAVS, GFP-TBK1 or GFP-IRF3(5D) expression plasmid using Lipofectamine 3000 according to the manufacturer's instructions

(Invitrogen) and treated with DMSO or 4-OI. 24 h after transfection, whole-cell lysates were resolved by 10% SDS-PAGE. After electrophoresis, proteins were transferred for 1 h 100 V at 4 °C to nitrocellulose membranes (0.45 μ, Bio-Rad) in a buffer containing 25 mM Tris, 192 mM glycine, and 10% ethanol. Membranes were blocked for 1 h at room temperature in 5% (wt/vol) dried milk in PBS and 0.05% (vol/vol) Tween-20 (PBS-T) and then were probed with the following primary antibodies: anti-RIG-I (1:5000, EMD Millipore, Cat# D14GG,), anti-ISG56 (1:5000, PA3-848, Thermo Fischer Scientific), anti-Actin (1:10000, EMD Millipore Cat# MAB1501), anti-GFP (1:3000, Santa Cruz Biotechnology, Cat# SC-9996) and monoclonal FLAG antibody M2 (1:5000, Sigma Cat#F1804). Antibody signals were detected by chemiluminescence using secondary antibodies conjugated to horseradish peroxidase (SeraCare Cat #5450 0011 anti-mouse HRP, 1:5000; SeraCare Cat# 5450 0010 anti-rabbit HRP, 1:5000) and an ECL detection kit (Millipore). Antibody signals were detected by chemiluminescence using secondary antibodies conjugated to horseradish peroxidise (Mandel Scientific) and an ECL detection kit (Millipore). Films were developed using a MINI-MED 90 X-Ray Film Processor (AFP Manufacturing Corporation)

## Co-Immunoprecipitation

HEK293 cells were co-transfected with expression plasmids as indicated in individual experiments and treated with DMSO or 4-OI. Whole-cell extracts (300 mg) were prepared 24 h after transfection and the extracts were incubated with 1 mg anti-Flag antibody M2(Sigma Cat#F1804). Precipitates were washed 5 times with lysis buffer, and eluted by boiling the beads for 3 min in 1× SDS sample buffer. Eluted proteins or 5% input whole-cell extracts were subjected to SDS-polyacrylamide gel electrophoresis (SDS-PAGE) in a 10% polyacrylamide gel. After electrophoresis, proteins were subjected to immunoblot analysis with anti-GFP (1:3000, Santa Cruz Biotechnology, Cat# SC-9996) and anti-Flag antibodies (1:5000, Sigma Cat#F1804). Antibody signals were detected by chemiluminescence using secondary antibodies conjugated to horseradish peroxidase (SeraCare Cat #5450 0011 anti-mouse HRP, 1:5000; SeraCare Cat# 5450 0010 anti-Rabbit HRP, 1:5000). Immunocomplexes were detected by using a chemiluminescence-based system (ECL). Films were developed using a MINI-MED 90 X-Ray Film Processor (AFP Manufacturing Corporation).

## Luciferase Assay

For the NF-κB luciferase assay, WT and NRF2 KO were stimulated with 4-OI (125 μM). Further, cells were transfected with 250 ng of pRLTK reporter plasmid, 250 ng of NF-κB-Luc reporter, and 250 ng of the expression plasmid encoding for IKKβ or the empty vector using Lipofectamine 3000 according to the manufacturer's guidelines. After 24 h of transfection, luciferase activity was measured with a dual-luciferase reporter assay and a GloMax 20/20 luminometer according to the manufacturer's recommendations (Promega).

## Enzyme-linked immunosorbent assay

Cytokines in the cell culture supernatants were measured using enzyme-linked immunosorbent assay (ELISA) for quantitative detection of IL-6 (R&D Systems Cat. No. DY206), CXCL10 (R&D Systems Cat. No. DY266), IL-1β (R&D Systems Cat. No. DY201), and IL-8 (R&D Systems Cat. No. DY208) all according to the manufacturer's instructions. Briefly, the plates coated with capture antibody overnight at room temperature (RT) were washed three times (PBS + 0.05% Tween). Samples and standards were then added to the wells and incubated overnight at 4 °C. Next, the wells were washed and incubated with detection antibody at RT for 2 h. Wells were washed again and conjugated with streptavidin HRP (R&D, 893975) at RT for 20 min, in the dark. After final washing, tetramethylbenzidine (TMB) substrate (Promega, 67431) was added till sufficient enzyme-substrate reaction was

visualized, which was then terminated by adding stop solution ((2 N) sulfuric acid (H₂SO₄), VWR Chemicals) and measured at 450 nm (570 nm for wavelength correction) using Synergy|HTX multimode reader.

## Supernatant transfer experiment

CT26 WT cells plated in 12-well dishes were pretreated with 4-OI or DMSO for 24 h and subsequently infected with VSVΔ51 at an MOI of 0.01. One hour after infection, the supernatant was removed to eliminate residual drug and virus and replaced with complete fresh media. The next day, supernatants were collected and transferred to fresh untreated CT26 WT cells. Infectivity was assessed by microscopy and flow cytometry as described earlier.

## Metabolites extraction

Metabolites were extracted from cells on ice using 800 μl 100% methanol for viral inactivation, followed by 200 μl of ice-cold HPLC-grade water. The cell solutions were vortexed for 10 s and incubated at −20 °C for 2 h. Then, the cell solutions were centrifuged at 4 °C, 16,000 × g for 20 min, and the supernatants were transferred to 1.5 ml Eppendorf tubes. The supernatants were dried down in a speedvac (Labconco Centrivap) at 8 °C. The dried metabolite extracts were resuspended in 100 μl of acetonitrile water (1:1, v/v) and sonicated for 10 min in ice water. The solution was centrifuged at 4 °C, 16,000 × g for 20 min. The supernatants were transferred to a 96-well plate (Greiner) and were subjected to targeted metabolomics LC-MS/MS analysis using a list of known itaconate-related metabolites. A quality control sample was prepared by pooling together 5ul of all samples and was used to observe the instrument performance during the run.

## Targeted metabolomics analysis and mass spectrometry

Targeted metabolomic analysis was performed on a triple quadrupole (QQQ) mass spectrometer (Agilent Triple Quadrupole 6495C, San Diego, CA), coupled to an ultra-high pressure liquid chromatography system (UPLC) system (1290 Infinity, Agilent Technologies) as previously described[81]. Data were acquired with Agilent MassHunter Workstation Data Acquisition (version 10.1). A CSH Phenyl-hexyl column (1.7 μm, 1.0 × 100 mm) (Waters, Taastrup, Denmark) was used for metabolites separation. Collision energies and product ions (MS2 or quantifier and qualifier ion transitions) were optimized. Electrospray ionization source conditions were set as follows: gas temperature, 200 °C; gas flow, 15 L/min; Nebulizer, 25 psi; sheath gas temperature, 325 °C; cap voltage, 3000 V; and nozzle voltage, 500 V. For the liquid chromatography, the following parameters were used. The gradient consisted of buffer A, and buffer B. Buffer A was 99.9% H₂O and 0.1% formic acid. Buffer B was 99.9% acetonitrile and 0.1% formic acid. The gradient with A/B ratios were as follows: T0, 99:1; T2.5, 99:1; T6, 86.9:13.10; T7, 1:99; T8.5, 1:99; T9, 99:1; T10, 99:1. The flow rate was 150ul/min. Three microliters of sample were injected. Multi-reaction monitoring (MRM) was used. A standard curve was recorded and integrated using the mass hunter platform (Agilent). The transitions used for 4-octyl-itaconic acid were 241.14 – >111 (quantifier, collision energy – 12 V), 241.14 – >67.1 (qualifier, collision energy – 24 V). The transitions for itaconic acid were the following: 129.02 – >85.1 (quantifier, collision energy – 8 V), 129.02 – >41.2 (qualifier, collision energy – 12 V). Retention time for 4-octyl itaconic acid was 8.2 min, and 2.0 min for itaconic acid.

## Targeted metabolomics data analysis

Transition lists, retention time, and raw data were loaded into Skyline (version 23.1)[82]. Then, peaks were evaluated and integrated, and intensities were exported. The area under the curve of the quantifiers was used for further analysis. Data were plotted using Python v3.9[83], and the packages matplotlib v3.5.1, numpy v1.22.2[83], pandas v1.4.1,

seaborn v0.12.0, statannotations v0.4.4. *T*-test independent was used for statistical analysis.

## Computational chemistry

All computational work was performed using Molecular Operating Environment (MOE), version 2020.09, Chemical Computing Group ULC, 910–1010 Sherbrooke St. W. Montreal, Quebec, H3A 2R7, Canada.

## Preparation of ligand and protein structures

The 2D structure of 4-OI was sketched using ChemDraw Professional 21.0 and was imported into the MOE window. The compound was subjected to an energy minimization up to a gradient of 0.001 kcal mol$^{-1}$ Å$^2$ using the MMFF94x force field and R-field solvation model, then it was saved as mdb file. The predominant protonation status of 4-OI in aqueous medium at pH 7 was calculated via the compute | molecule | wash command in the database viewer window. X-ray crystal structures of the human inhibitor of κB kinase β (IKKβ) in complex with the inhibitor K252a (PDB ID: 4KIK)[52] and a C-terminal fragment of human IKKβ in complex with NF-κB essential modifier (NEMO, or IKKγ) (PDB ID: 3BRV)[53] were used for the covalent docking studies. The potential was set up to Amber10:EHT as a force field and R-field for solvation. The addition of hydrogen atoms, removal of water molecules farther than 4.5 Å from ligand or receptor, correction of library errors, and tethered energy minimization of the binding site were performed via the QuickPrep module.

## Structural modeling

Covalent docking of 4-OI was performed into six binding sites encompassing the cysteine residues (12, 179, 412, 464, 524, and 716) of human IKKβ. Each cysteine residue was selected as the reactive site. The 1,4-Michael addition was set as the reaction. Placement trials were set to 100 poses with an induced fit refinement. The final scoring function was London dG with ten poses.

## Statistics and reproducibility

The data are shown as means of biological replicates ± standard error means (SEM). The number of replicates is indicated within figure legends. Statistical significance between groups was determined using a two-tailed Student's *t*-test when the data exhibited normal distribution and F Test to compare two variances data *p*-value > 0.1, two-tailed unpaired Student's *t*-test with Welch's correction when the data exhibited normal distribution but F Test *p*-value < 0.1, one-way or two-way ANOVA followed by Sidak's test for multiple comparisons, and two-tailed Mann–Whitney test when the data set did not pass the normal distribution test. All statistical analyses were performed using GraphPad Prism software (v.10.1.0) and *p*-values are reported directly on the figures.

## Data visualization

GraphPad Prism software (v.10.1.0), Microsoft PowerPoint (v16.83), and Adobe Illustrator (v28.3) were used for the graphical representation of the data. BioRender.com for creating scientific illustrations.

## Reporting summary

Further information on research design is available in the Nature Portfolio Reporting Summary linked to this article.

## Data availability

Sequencing data are uploaded to GEO with the accession number GSE232509 Metabolomics data are uploaded to MassIVE under the accession number MSV000094355 https://massive.ucsd.edu/ProteoSAFe/dataset.jsp?task=47883b17d55b400f89eeba94c6e6cf8d Source data are provided with this paper. All data generated in this study are provided in the Supplementary Information/Source Data File. Source data are provided with this paper.

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

## Acknowledgements

We thank Mohammed Mosa, Constantin Menche, and Sara Stier from Henner Farin's laboratory as well as Annette Trzmiel from Georg-Speyer-Haus Flow Core Facility (Frankfurt, Germany) for the experimental support with patient-derived tumor organoids. In addition, we would like to thank the Bioinformatics Core Facility at Aarhus University, Department of Biomedicine, and especially Per Qvist who helped with the analysis of the RNA, as well as the FACS Core Facility and the Bioimaging Core Facility at Aarhus University, Department of Biomedicine for their support with the use of the flow cytometers and confocal microscopes. The authors would like to thank Herdis Berg Johansen for technical assistance in the Olagnier laboratory. In addition, the authors would like to thank Kaare Meier, Nikola Mikic for handling the brain tumor database and neurosurgeons Ann Kathrine Sindby, Bo Bergholt, and Søren Ole Stigaard Cortnum (Aarhus University Hospital) for contributing to surgeries. This research work was supported by the Lundbeckfonden (R335-2019-2138), the Novonordiskfonden (NNF22OC0079512), the Danish Cancer Society (R279-A16218), the Independent Research Fund Denmark (1026-00003B), the Brødrene Hartman Fonden, the Hørslev Fonden, the Einar Willumsens mindelegat, the Eva og Henry Frænkels mindefond, the Beckett Fonden, the Lily Benthine Lunds fonden, the Aase of Ejnar Danielsens Fonden, the Frimodt-Heineke Fonden, the Sofus Carl Emil Friis Legat, the Danielsens fond, the børnecancerfonden, the Dagmar Marshalls fond, the Lykfekdts Legat, the Kræftfonden to D.O. This work was also supported by a grant from the Danish National Research Foundation (DNRF164). D.v.d.H. was supported by a PhD grant from the Dansk Kræftforsknings fond. L.d.l.V. and M.H. were supported by Cancer Research UK (C52419/A22869). A.R.K. was supported by the Danish Cancer Society (R304-A17698-B5570 and R295-A16770), the Lundbeck Foundation (R325-2019-1490), the Independent Research Fund Denmark (9039-00307B). This research work was supported by the Independent Research Fund Denmark (DFF-37741) to W.-H.H. This research work was supported by The Lundbeck Foundation (R325-2019-1490) to A.O. J.K. was supported by the Lundbeckfonden (R307-2018-3667) and Carlsberg Fonden (CF19-0687). T.B.P. was supported by the Novonordiskfonden (NNF19OC0054782). N.v.M. and P.K. were supported by Dutch Cancer Society Bas Mulder Award 11056 and Stichting Overleven met Alvleesklierkanker (SOAK 22.02). M.M.R. was supported by the Young Investigator Award from the Novo Nordisk Foundation, grant number NNF19OC0056043, the Carlsberg Young Investigator fellowship as well as Aarhus universitet forskningsfond. This research work was supported by funding from the Italian Association for Cancer Research (IG-2019-22891) to J.H. H.F.F. was supported by institutional funds from the Georg-Speyer-Haus and the LOEWE Center Frankfurt Cancer Institute (FCI) funded by the Hessen State Ministry for Higher Education, Research and the Arts [III L 5 – 519/03/03.001 - (0015)] and the EU/EFPIA/OICR/McGill/KTH/Diamond Innovative Medicines Initiative 2 Joint Undertaking (EUbOPEN grant n° 875510). W.A.M.E. is supported by the Arab-German Young Academy of Sciences and Humanities (AGYA) and the German Federal Ministry of Education and Research (BMBF) grant (01DL20003). T.A. was supported by grants from the Terry Fox Research Institute, the Canadian Cancer Society Research Institute, and the Cancer Research Society. Salaries were supported by a PhD fellowship to N.K. from the Graduate School of Health at Aarhus University and one-third of a PhD fellowship from the Department of Biomedicine at Aarhus University to D.v.d.H., a Kræftens Bekæmpelse postdoctoral fellowship (R306-A18092) to M.C.-T., a Research Year Grant from the Vissing Fonden to E.H. H.W. salary was supported by the European Union's Horizon 2020 research and innovation programme under the Marie Skłodowska-Curie grant agreement No 813343. This research was supported by grants to R.L. from the Canadian Institutes of Health Research (CIHR) (PJT-169663). This article reflects the views of the authors, and the joint undertaking is not liable for any use that may be made of the information contained herein.

## Author contributions

N.K. performed most of the experiments and wrote the material and methods sections. A.S., P.K., A.H.F.R., N.K., and M.v.d.H. performed ex vivo and in vivo experimentations. R.A. performed the revision experiments on cell lines shown in Figures 1 and 2, in particular, the virus titration and GFP measurements of virus infection in CT26wt and 76-9 with 4OI. B.W. performed all murine tumor core experiments and the in vitro experiments with Sindbis virus. A.K. helped with the establishment of the 3D tumoroid colon cancer model. X.H., Z.L., C.W., and R.L. performed the in vitro luciferase assay, co-immunoprecipitation experiment, generated the point mutants, and performed the Luciferase experiments with the mutants. A.S. and H.D.H. completed the in vitro experiments with Reovirus, Vaccinia virus and Measles virus. M.C.-T., D.v.d.H., C.F.K., A.B., and V.S. supported some of the in vitro

experimentations. E.K. and M.C. performed the metabolite measurements using mass spectrometry. E.H. performed the infection experiments on the organotypic glioblastoma slices. K.S.T. performed the CXCL10 and IL-6 ELISA measurements. R.N.O., E.B.S., F.B., F.E.H.K., and A.E.K. performed all the chemistry on the project. H.W. generated the figures representing the mass spectrometry data on MAVS and IKKβ. D.D.C. and M.M. performed the in vitro experiments on the pancreatic cell lines. M.H. and L.d.l.V. generated the NRF2 KO 786-O cells. T.T., A.O., and W.-H.H. helped with the establishment of the organotypic brain tumor model. L.d.l.V., C.K.H., and J.K. provided cells and reagents. Y.Z. and M.R. performed the mass spectrometry experiment on MAVS and IKKβ. W.A.M.E. performed all the structural modeling work on IKKβ. A.R.K. is the neurosurgeon who excised the brain tumor samples from the patients. N.v.M., T.B.P., J.H., L.A.O., D.G.R., M.M.R., J-S.D., H.F.F., and T.A. contributed to the development of the project, supervised the researchers involved in the project, and edited the manuscript. D.O. conceived and managed the project, designed the experiments, assembled the figures, and wrote the manuscript. All authors edited and approved the manuscript.

## Competing interests

M.M.R. reports research funding from Novo Nordisk A/S. The remaining authors declare no competing interests.

## Additional information

Naziia Kurmasheva [1], Aida Said[2,3], Boaz Wong [2,4], Priscilla Kinderman [5], Xiaoying Han [6], Anna H. F. Rahimic[1], Alena Kress [7,8,9], Madalina E. Carter-Timofte[1], Emilia Holm[1], Demi van der Horst[1], Christoph F. Kollmann [1], Zhenlong Liu[6], Chen Wang[6], Huy-Dung Hoang[2,3], Elina Kovalenko[1], Maria Chrysopoulou [1], Krishna Sundar Twayana[1], Rasmus N. Ottosen[10], Esben B. Svenningsen [10], Fabio Begnini[10], Anders E. Kiib [10], Florian E. H. Kromm[10], Hauke J. Weiss [11], Daniele Di Carlo[12], Michela Muscolini[12], Maureen Higgins[13], Mirte van der Heijden[5], Rozanne Arulanandam[4], Angelina Bardoul[14,15,16], Tong Tong [17,18,19], Attila Ozsvar[1,18], Wen-Hsien Hou[1], Vivien R. Schack [1], Christian K. Holm [1], Yunan Zheng[20], Melanie Ruzek[21], Joanna Kalucka [1,22], Laureano de la Vega [13], Walid A. M. Elgaher [23], Anders R. Korshoej [17,18,19], Rongtuan Lin[6], John Hiscott[12], Thomas B. Poulsen[10], Luke A. O'Neill [11], Dominic G. Roy[14,15,16], Markus M. Rinschen [1,24,25], Nadine van Montfoort[5], Jean-Simon Diallo [2,4], Henner F. Farin [7,8,26,27], Tommy Alain [2,3,27] & David Olagnier [1] ✉

[1]Department of Biomedicine, Aarhus University, 8000 Aarhus C, Denmark. [2]Department of Biochemistry Microbiology and Immunology, University of Ottawa, Ottawa, ON K1H 8M5, Canada. [3]Children's Hospital of Eastern Ontario Research Institute, Ottawa, ON K1H 8L1, Canada. [4]Ottawa Hospital Research Insitute, Ottawa, ON K1H 8L6, Canada. [5]Department of Gastroenterology and Hepatology, Leiden University Medical Center, Leiden, The Netherlands. [6]Lady Davis Institute, Jewish General Hospital and Department of Medicine, McGill University, Montreal, QC H3T 1E2, Canada. [7]Georg-Speyer-Haus, Institute for Tumor Biology and Experimental Therapy, Frankfurt am Main, Germany. [8]Frankfurt Cancer Institute, Goethe University, Frankfurt am Main, Germany. [9]Faculty of Biological Sciences, Goethe University, 60438 Frankfurt am Main, Germany. [10]Department of Chemistry, Aarhus University, 8000 Aarhus C, Denmark. [11]School of Biochemistry and Immunology, Trinity College Dublin, Trinity Biomedical Sciences Institute, Dublin 2, Ireland. [12]Pasteur Laboratories, Istituto Pasteur Italia - Fondazione Cenci Bolognetti, Rome 00161, Italy. [13]Jacqui Wood Cancer Centre, Division of Cellular Medicine, School of Medicine, University of Dundee, Dundee, UK. [14]Cancer Axis, CHUM Research Centre, Montreal, Canada. [15]Department of Microbiology, Infectious Diseases and Immunology, Faculty of Medicine, University of Montreal, Montreal, Canada. [16]Institut du Cancer de Montréal, Montreal, QC, Canada. [17]Department of Neurosurgery, Aarhus University Hospital, 8200 Aarhus N, Denmark. [18]Department of Clinical Medicine, Aarhus University, 8200 Aarhus N, Denmark. [19]DCCC Brain Tumor Center, Copenhagen University Hospital, Copenhagen, Denmark. [20]Small Molecule Therapeutics & Platform Technologies, AbbVie Inc., 1 North Waukegon Road, North Chicago, IL 60064, USA. [21]AbbVie, Bioresearch Center, 100 Research Drive, Worcester, MA 01608, USA. [22]Steno Diabetes Center Aarhus, Aarhus University Hospital, Aarhus, Denmark. [23]Department of Drug Design and Optimization, Helmholtz Institute for Pharmaceutical Research Saarland (HIPS), Helmholtz Centre for Infection Research (HZI), Saarland University, E8.1, 66123 Saarbrücken, Germany. [24]III. Department of Medicine and Hamburg Center for Kidney Health, Hamburg, Germany. [25]Aarhus Institute of Advanced Studies, Aarhus University, Aarhus, Denmark. [26]German Cancer Consortium (DKTK), Frankfurt/Mainz partner site and German Cancer Research Center (DKFZ), 69120 Heidelberg, Germany. [27]These authors contributed equally: Henner F. Farin, Tommy Alain. ✉e-mail: olagnier@biomed.au.dk

