## [Peer Review File · Nature Communications]

Octyl-Itaconate Enhances VSV Δ 51 Oncolytic Virotherapy by Multitarget Inhibition of Antiviral and Inflammatory PathwaysREVIEWER COMMENTS

Reviewer #1 (Remarks to the Author); expert in immunometabolism, itaconate:

Authors describe effect of pharmacological treatment by octyl-itaconate on the viral spread in the context of the tumor model. Overall, there appears to be some increase in viral titers which, however, does not translate to benefit in anti-tumor treatment (based on fig 2j).

Authors then go on to test the effect of oncolytic virus and OI cotreatment in organoids, which confirms viral load increase observed previously and seem to be synergistic in terms of cell killing (albeit only 5 out of 9 organoids are shown).

The following 4 figures are devoted to the investigation of the potential mechanism through which OI exerts the effects. Authors find that improvement in viral load is not driven by Keap1-Nrf2 pathway and show that generally OI inhibits antiviral immunity.

Confusingly, ATF3 is not considered as one of the potential mechanisms of action even though it is (a) known to be induced by electrophilic derivatives of itaconate in a Keap1-Nrf2 independent manner and (b) to affect directly interferon activation (see ref 21 in your paper).

Next the authors test IRF3, MAVS, IKKb and RELA as potential mediators of OI action. While they demonstrate that there is a degree of modification of cysteines of these proteins, the mechanistic involvement of these proteins is strongly overinterpreted against basic biochemical rules - the fact that Irf3/MAVS/etc KO does not have additive effect with OI action does not mean that they are part of the OI mechanism of action! It just means that knock-out of these proteins turns off the interferon pathway (well known fact) and hence leads to an effect that is very similar to pharmacological shut down of interferon pathway with octylitaconate - improvement in viral loads. To claim any mechanistic roles for the OI in the context of these proteins authors should overexpress proteins where corresponding cysteines are mutated and then subject to OI treatment.

Taken together, the manuscript reports no strong translationally important data in terms of survival, does not yield mechanistic insights beyond already known importance of IRF3/MAVS/IKKB/RELA in the interferon regulation and last but not the least misrepresents this work as if it is in any way relevant to the action of endogenous, non-derivatized itaconate. The latter is very sad and borderline unethical and below I expand on this point.

Panels 1b and 1c are the only two pieces of data with actual itaconate. Furthermore, these two panels have inconsistent results in terms of itaconate and consistent results only in terms of octylitaconate, and the rest of the paper deals exclusively with the effects of octylitaconate. The title of the paper is completely misleading as it mentions non-derivatized itaconate. The accurate name should be "Octyl-Itaconate enhances oncolytic virotherapy by multitarget inhibition of antiviral and inflammatory pathways".

This sleight of hand is very upsetting given that there are very significant differences between endogenous non-derivatized itaconate and octylitaconate or other electrophilic derivatives of itaconate.

Furthermore, authors mask their sleight-of-hand by using expression "itaconate family" but it is not clear from their data why it should be about itaconate. In fact, their own data show that other electrophilic compounds similar to octylitaconate such as DMF and sulforaphane demonstrate similar behavior.

Therefore, it appears to be a paper about one more artificial chemical compound that behaves in a manner similar to other electrophilic compounds described by this group.

In the opinion of this reviewer, this might be an interesting pharmaceutical study case expanding the list of compounds but does not add to the body of fundamental scientific knowledge about oncolytic viruses and

due to utilization of misrepresenting language the paper actually does a disservice to the immunological community studying itaconate.

some minor points

INTRODUCTION:

lines 135, 137: perhaps it would be worth citing original study that linked itaconate and its derivative and immunoregulation (Lampropoulou et al)
lines 138-139: this is directly false statement: In neither of the cited papers itaconate was shown to suppress inflammation through Nrf2. Mills et al study have shown that OCTYLITACONATE binds to Keap1 and suppresses inflammatory cytokines.
line 144: It is directly false statement: In neither of those papers it was shown that itaconate is highly electrophilic compound comparable to strongly electrophilic itaconate derivatives.
In fact, ref 21 directly provides the data that endogenous itaconate is not anywhere as strong as electrophilic derivative. Furthermore, later paper by Swain et al demonstrated significant difference in electrophilicity between itaconate and octylitaconate.

lines 145-147: the citation are misrepresenting scientific literature, mixing papers on octylitaconate and actual itaconate. Authors should directly and explicitly separate the statements about octylitaconate and non-derivatized itaconate since those have been shown to be different, for instance exogenous itaconic acid itself was shown to boost type I interferon at least in some contexts (e.g. long pretreatment prior to LPS)
lines 151-152: same as above - authors should be very careful in describing effects of itaconate vs its electrophilic derivatives

FIGURE 1:

Panels 1b and 1c once again demonstrate that non-derivatized itaconate and octylitaconate have two different effects - only 4OI have consistently yielded effect in different cell types.
In fact, for the many subsequent figures authors choose to work specifically on 786-O cell line from panel 1c where itaconate and OI behave very differently, again indicating that this is paper about octylitaconate not about itaconate as misleading title and introduction try to claim!

On a minor note, authors could/should add important controls - such as rescue with NAC of each and condition with malonate as control for Sdh inhibition effects of itaconate.

line 370: There are no references! It is inaccurate to claim that itaconate can modify NLRP3, KEAP1, STING and JAK1 given that studies have shown it for electrophilic derivatives of itaconate or similarly electrophilic constructs for pull down.
So, authors should really refer to the data that were published.

Reviewer #2 (Remarks to the Author); expert in itaconate:

The manuscript by Kurmasheva and colleagues highlights the impact of 4-OI to enhance tumor-specific VSVΔ51M replication and oncolysis in both in vitro and in vivo settings in various model systems. The study suggests that 4-OI inhibits the antiviral immunity of cancer cells via targeting MAVS, JAK1, and NF-κB pathways.

The study is well conducted and has an impact on the cancer and immunity community. The authors used "itaconate" in the title, but almost all the data were generated with 4OI treatment strategies. Itaconate and 4OI are distinct metabolites with distinct impacts on metabolism and cell function. Since the presented story is focused on 4OI the biological link to itaconate is missing. Additional data is needed to link the observation to itaconate.

1. The manuscript described the impact of 4OI on cell metabolism. Since itaconate and 4OI have distinct impacts on cell metabolism and function discussions on itaconate should be limited. The title and the discussion part should be adjusted accordingly. If the authors want to discuss itaconate, they may want to include additional data and repeat some of their studies with itaconate treatments as well as IRG1 KO models.

2. Previous studies reported that 4OI may inhibit virus replication through various mechanisms. This observation agrees with Fig.5f depicting that 4OI does not promote wtVSV replication (and may have anti-viral effects). This discrepancy could be explained in more detail. Does this observation phenocopies to itaconate treatments?

4. Work by Chen et al. 2023 indicates that endogenous itaconate influences TET2 activities to dampen inflammatory responses involving the NF- κ B pathway. Does 4-OI targets IKK β and promotes viral infection through NF- κ B by altering TET?

5. The impact on 4OI, itaconate, and other derivatives are different in Fig. 1b compared to Fig. 1c. Are the changes induced due to species differences (mouse cells CT26WT versus human 786-O cells)? Some discussions might be helpful to understand the differences.

6. Since IRG1 expression levels are upregulated in the tumor microenvironment, checking if metabolites (e.g. itaconate and succinate) are changed during 4-OI treatment and VSV Δ 51M infection would be informative.

Reviewer #3 (Remarks to the Author); expert in oncolytic virus:

Kurmasheva et al. describe the krebs cycle-derived metabolite itaconate and derivatives effects on oncolytic therapy with VSV Δ 51. The authors present a large amount of data that details the in vitro mechanism by which the metabolites interfere with the type I IFN and NF- κ B-mediated antiviral responses. Overall, the data is well presented, and conclusions are clear. The manuscript could just be significantly strengthened with more supporting in vivo data. And although the authors show nice mechanistic data using in vitro cancer cell lines and 3D organoids they data confirming these in vivo. The clinical application of 4-OI and VSV Δ 51 would be significantly strengthened by some additional data outlined below.

Major

- Can 4-OI be delivered orally? What does this do to efficacy? Maybe continuous in water?
- Does 4-OI treatment enhance survival in animals?
- Antiviral immunity only assessed in vitro cancer cells. What about the role of infection of lymphoid organs in vivo? And other non-cancer cells? Would be insightful to see serum cytokine analysis and RNA seq from tumors. Do these data agree with in vitro findings?

Minor

- Show actual p values
- Making clear the difference between infection (entering cells) and replication (new progeny)

Specific comments:

- Line 114: "highly sensitive to interferon (IFN)" Not the case for all OV's (Vaccinia Virus for example). Maybe discuss in the context of VSV?
- Line 118: "Because significant replication of OV's in tumor tissues is necessary for the highest efficacy" Is this true? References to support? Particularly with VSV in immunocompetent models? Or is stimulating an anti-tumor immune response (with or without direct tumor infection) more important?
- Line 145: Sentence is unclear.
- Line 191: Used the work "synergizing" I would remove unless evidence to show.
- Line 225: "intratumorally injected with 4-OI for 24 hours" What is meant here? Continuous for 24hr? Single injection 24h prior? Please clarify.
- Line 242: I would like more rational on benefits of "pathologically relevant 3D tumoroid models" over in vivo models?
- Line 245: Remove "significantly" since no stats are presented in Figure 3a.

Figure 1b and c – could enhanced infection also be due to increases entry (LDLR) receptor expression? This should be examined as a potential mechanism as well.
Figure 1f – I would like to see all cell lines treated with the same concentration of 4-OI and infected with same MOI of virus. Might be best to move different treatments to supplemental so a direct comparison can be made across all cell lines.
Figure 1g – should have separate letter for plaque area figure to make clearer.
Figure 1h – provide rationale for calcein green staining in text.
Why is 76-9 cell line not included in figure 1 if used as one of the main animal models? I would like to see how infection, replication, and cytotoxicity compares to CT26.

Figure 2b – images of 76-9 tumors?
Figure 2c – titers from 76-9 tumors?
Figure 2c-e – scales should be kept the same for easier comparison between models.
Figure 2f – Should include scale bar.
Figure 2j – Would like to see survival data.

Figure 3c – figure legend needs further details. What is TBP in figure?
Figure 3e-g – Figure legend needs significantly more info.

Figure 4c – this is difficult to interpret. Why not present like in Figure 1e?

Reviewer #4 (Remarks to the Author); expert in innate immunity:

In this manuscript, Kurmashewa et al. reported the enhancing effect of 4-octyl itaconate (4-OI) on the oncolytic virotherapy using VSV Δ 51M. Mechanistically, the authors propose that 4-OI targets multiple innate immune signaling pathways including the RIG-I-MAVS, JAK1-STAT1, and NEMO-IKK β -NF- κ B. Overall, the authors have performed a comprehensive evaluation demonstrating the efficacy of 4-OI in OV against cancer cell lines, engrafted tumors, patient-derived tumoroids, and brain tumor slices. On the other hand, the mechanisms underlying 4-OI's effect on viral replication and host innate immunity need more supporting evidences. My comments are as follows:

(1) The authors showed that 4-OI treatment only marginally promoted infection of VSV Δ 51M in normal cell lines (Figure S1). This seems to suggest that innate immunity is not the major factor determining the selective role of Δ 51M on cancer cells over normal cells, or the anti-IFN/NF- κ B roles of 4-OI as proposed by the authors do not work in normal cells. To address this question, the authors should check the IFN-beta production (e.g., by RT-PCR or ISG expression) upon VSV Δ 51M infection in normal cell lines (HUVEC, Fibroblasts), with or without the treatment of 4-OI.

(2) It is surprising that DMF treatment in normal cells drastically promoted the infection of VSV (nearly 100%, as shown in Fig S1c). This implicates that DMF may have other mechanisms (not NF- κ B) which promote VSV replication in normal cells. Could the author provide comments on this?

(3) Figure S2, as 4-OI targets multiple downstream innate immune signaling molecules that are shared by distinct sensing pathways, it is confusing to me why 4-OI restricted or had no effect on multiple types of viruses and oncolytic viruses, including WT-VSV, in the 786-O cancer cell line. This is suggesting that 4-OI has a unique role for VSV Δ 51M that cannot be explained by its anti-interferon or anti-inflammatory role. Again, I suggest the author to monitor the IFN and NF- κ B signaling activation in 4-OI treated cells during Sindbis virus, Reovirus, Vaccinia virus and Measles virus infections.

(4) Typo in the title, Figure S2c, 'Vaccinia virus'.

(5) Although the authors have identified C283 MAVS in the cysteine profiling, the competition ratio remained low (1.18) when compared to the sites of IKK β . To confirm that C283 is the major site of MAVS, the author should perform a 4-OI-alk binding assay with a MAVS C283 mutant (e.g. C283A).

(6) Figure 6e, the IPed MAVS for lane 5 and lane 6 needs to be balanced to justify the decrease of

binding between MAVS and TBK1 upon 4-OI treatment. To further validate the specificity of 4-OI on C283, the author should also check the binding of MAVS-C283A to TBK1 with or without the treatment of 4-OI.

(7) Figure 7b and S9, it looks like P65 mostly located in the nucleus in cells that were not infected with VSV (Row 3 of S9). The author should repeat the staining to make sure that the cells were in resting state before VSV infection.

(8) To avoid the potential off-target effect of 4-OI-Alk in the binding assay, the author should include a negative control in Figure 7g with a protein that is not known to be modified by 4-OI. For example, the closely related kinase IKK α may serve as a good control.

RESPONSE TO REVIEWERS' COMMENTS

Reviewer #1 (Remarks to the Author); expert in immunometabolism, itaconate:

Authors describe effect of pharmacological treatment by octyl-itaconate on the viral spread in the context of the tumor model. Overall, there appears to be some increase in viral titers which, however, does not translate to benefit in anti-tumor treatment (based on fig 2j). Authors then go on to test the effect of oncolytic virus and OI cotreatment in organoids, which confirms viral load increase observed previously and seem to be synergistic in terms of cell killing (albeit only 5 out of 9 organoids are shown).

The following 4 figures are devoted to the investigation of the potential mechanism through which OI exerts the effects. Authors find that improvement in viral load is not driven by Keap1-Nrf2 pathway and show that generally OI inhibits antiviral immunity.

Response: We would like to thank reviewer 1 for his/her evaluation of our manuscript as well as for the different suggestions made for improvement.

Confusingly, ATF3 is not considered as one of the potential mechanisms of action even though it is (a) known to be induced by electrophilic derivatives of itaconate in a Keap1-Nrf2 independent manner and (b) to affect directly interferon activation (see ref 21 in your paper).

Response: We fully agree with the reviewer that ATF3 should have been considered as one of the potential mechanisms through which 4-OI would drive its NRF2-independent proviral action. We have now completed a series of experiment using CRISPR/Cas9 gene editing of ATF3 and show that ATF3 is not responsible for the biological effects driven by 4-OI in our cellular system. Additionally, VSV Δ 51 did not lead to an induction of IKBZ. The new data are displayed in Fig. S15 a,b.

Next the authors test IRF3, MAVS, IKK β and RELA as potential mediators of OI action. While they demonstrate that there is a degree of modification of cysteines of these proteins, the mechanistic involvement of these proteins is strongly overinterpreted against basic biochemical rules - the fact that Irf3/MAVS/etc KO does not have additive effect with OI action does not mean that they are part of the OI mechanism of action! It just means that knock-out of these proteins turns off the interferon pathway (well known fact) and hence leads to an effect that is very similar to pharmacological shut down of interferon pathway with octylitaconate - improvement in viral loads. To claim any mechanistic roles for the OI in the context of these proteins authors should overexpress proteins where corresponding cysteines are mutated and then subject to OI treatment.

Response: In agreement with the reviewer's comment, we decided to improve the mechanistic aspects of our study. We originally demonstrated a degree of alkylation of different proteins involved in the antiviral and the inflammatory pathways including MAVS and IKK β , two newly identified targets of 4-OI. To claim some mechanistic roles for 4-OI in the context of these two proteins, as suggested by the reviewer, we overexpressed MAVS and IKK β where cysteine residues 283 and 179 were mutated into alanine, respectively. Affinity pull-down experiments with MAVS mutant showed some specificity of binding to the identified cysteine target residue. Additionally, ISRE luciferase and NF- κ B luciferase reporter assays with the different mutant proteins also confirmed the role of the reported cysteine

residues in the regulation of antiviral and inflammatory responses. The new data are displayed in Fig. 7 and 8.

Taken together, the manuscript reports no strong translationally important data in terms of survival, does not yield mechanistic insights beyond already known importance of IRF3/MAVS/IKKB/RELA in the interferon regulation and last but not the least misrepresents this work as if it is in any way relevant to the action of endogenous, non-derivatized itaconate. The latter is very sad and borderline unethical and below i expand on this point.

Response: Considering the reviewer comments on our manuscript:

1. no strong translational important data,
2. no yield of novel mechanistic insights beyond already known regulatory mechanisms,
3. misrepresentation of the work, we extensively revised our manuscript as follows.
 1. We repeated a series of new *in vivo* experiment where we evaluated the therapeutic potential of combining 4-OI with VSV Δ 51 *in vivo*. Mice given the combination therapy were successful in controlling tumor burden as tumor volumes were in this treatment group significantly smaller compared with either monotherapy. The combined treatment also significantly prolonged survival of animals as compared with all other treatment conditions. Remarkably, the combination therapy produced complete remission in 87.5 % of the mice. The cured CT26WT-bearing mice that had received the combination regimen subsequently became immune to rechallenge with the same cancer cells. Data are displayed in figure 2.
Additionally, we would like to mention that our original manuscript also contained data on the combinatorial treatment (VSV Δ 51+4-OI) in 3D patient-derived colon tumoroids and patient-derived organotypic glioblastoma slices. We are confident that the data presented report some strong translational significance.
 2. As detailed in the previous comment, we improved the mechanistic aspects of our study by overexpressing MAVS and IKK β where cysteine residues 283 and 179 were mutated into alanine, respectively. Antiviral and inflammatory responses using luciferase reporter assays demonstrated the role of the reported cysteine residues targeted by 4-OI in the regulation of these signaling pathways. The new data are presented in Fig. 7i and 8l.
 3. It was never our intention to misrepresent our work, but the reviewer makes a valid point that some confusion may have arisen from the original version of the manuscript. For that reason and as suggested by the reviewer, we decided to focus our work on the activity of the itaconate derivative 4-OI in promoting oncolytic virotherapy. The new version of the manuscript reflects this change. The title, abstract, text and figures have been adjusted accordingly.

Panels 1b and 1c are the only two pieces of data with actual itaconate. Furthermore, these two panels have inconsistent results in terms of itaconate and consistent results only in terms of octyl itaconate, and the rest of the paper deals exclusively with the effects of octylitaconate. The title of the paper is completely misleading as it mentions non-derivatized itaconate. The accurate name should be "Octyl-Itaconate enhances oncolytic virotherapy by multitarget inhibition of antiviral and inflammatory pathways".

Response: We agree, and the title has been edited accordingly.

This sleight of hand is very upsetting given that there are very significant differences between endogenous non-derivatized itaconate and octylitaconate or other electrophilic derivatives of itaconate. Furthermore, authors mask their sleight-of-hand by using expression "itaconate family" but it is not clear from their data why it should be about itaconate. In fact, their own data show that other electrophilic compounds similar to octylitaconate such as DMF and sulforaphane demonstrate similar behavior. Therefore, it appears to be a paper about one more artificial chemical compound that behaves in a manner similar to other electrophilic compounds described by this group.

In the opinion of this reviewer, this might be interesting pharmaceutical study case expanding the list of compounds but does not add to the body of fundamental scientific knowledge about oncolytic viruses and due to utilization of misrepresenting language the paper actually does misservice to the immunological community studying itaconate.

Response: We fully agree with the reviewer that itaconate and 4-OI represent two different chemical entities acting biologically quite distinctly one from the other. We reorganized and rewrote some sections of our manuscript to highlight this point specifically. Future studies will be needed to study the possible impact of the IRG1/itaconate axis in affecting oncolytic virotherapy.

We do feel that that this collection of extensive data reports more than just another electrophilic artificial chemical compound enhancing the replication of an oncolytic virus. First, the drugs mentioned by the reviewer, SFN and DMF, work quite differently than 4-OI in promoting VSV Δ 51 oncolytic virotherapy. For example, SFN does promote OV therapy through NRF2 engagement, which is not the case of 4-OI. DMF does promote OV therapy but through an unknown mechanism. Here, we report, two novel proteins and cysteine residues targeted by 4-OI that regulate antiviral and inflammatory responses. Additionally, we bring an extensive amount of *in vivo* (animal models) and *ex vivo* (patient material) data showing the strong translational potential of the combination 4-OI+VSV Δ 51. Altogether and with the modifications applied to the revised version of the manuscript, we are confident that this study adds significantly to the body of fundamental new knowledge in the cancer/OV/immunology community studying these different elements.

some minor points

INTRODUCTION:

lines 135, 137: perhaps it would be worth citing original study that linked itaconate and its derivative and immunoregulation (Lampropoulou et al)

Response: The work by Lampropoulo *et al* is now cited and included in the revised version of the manuscript.

lines 138-139: this is directly false statement: In neither of the cited papers itaconate was shown to suppress inflammation through Nrf2. Mills et al study have shown that OCTYLITACONATE binds to Keap1 and suppresses inflammatory cytokines.

Response: The reviewer is correct, and we adjusted all statements related to the use of itaconate vs 4-octyl-itaconate in the revised version of the manuscript. Literature has also been modified accordingly.

line 144: It is directly false statement: In neither of those papers it was shown that itaconate is highly electrophilic compound comparable to strongly electrophilic itaconate derivatives. In fact, ref 21 directly provides the data that endogenous itaconate is not anywhere as strong as electrophilic derivative. Furthermore, later paper by Swain et al demonstrated significant difference in electrophilicity between itaconate and octylitaconate.

Response: The work by Swain *et al* is now referenced and included in the revised version of the manuscript. The text has been adjusted accordingly.

lines 145-147: the citation are misrepresenting scientific literature, mixing papers on octylitaconate and actual itaconate. Authors should directly and explicitly separate the statements about octylitaconate and non-derivatized itaconate since those have been shown to be different, for instance exogenous itaconic acid itself was shown to boost type I inteferon at least in some contexts (e.g. long pretreatment prior to LPS)

Response: The text is now adjusted accordingly.

lines 151-152: same as above - authors should be very careful in describing effects of itaconate vs its electrophilic derivatives

Response: The text is now adjusted accordingly.

FIGURE 1:

Panels 1b and 1c once again demonstrate that non-derivatized itaconate and octylitaconate have two different effects - only 4OI have consistently yielded effect in different cell types. In fact, for the many subsequent figures authors choose to work specifically on 786-O cell line from panel 1c where itaconate and OI behave very differently, again indicating that this is paper about octylitaconate not about itaconate as misleading title and introduction try to claim!

Response: We fully agree with Reviewer 1 and followed his/her suggestion to focus our manuscript on 4-OI specifically. In line, the title and the introduction part have been adjusted accordingly.

On a minor note, authors could/should add important controls - such as rescue with NAC of each and condition with malonate as control for Sdh inhibition effects of itaconate.

Response: This is a valid point. We performed a NAC rescue experiment where cells were treated with the ROS scavenger prior to 4-OI and virus challenge. The new data are displayed in Fig. 4k and show that a pre-treatment with NAC did not alter 4-OI capacity to promote VSV Δ 51 infection in 786-O cells. Considering the refocus of our manuscript on 4-OI and the new metabolic measurements displayed in Fig. S1, a rescue experiment using malonate in 4-OI-treated cells, that did not alter succinate levels, appeared irrelevant.

line 370: There are no references! It is inaccurate to claim that itaconate can modify NLRP3, KEAP1 STING and JAK1 given that studies have shown it for electrophilic derivatives of itaconate or similarly electrophilic constructs for pull down. So, authors should really refer to the data that were published.

Response: The text has been adjusted accordingly and a reference has been added.

Reviewer #2 (Remarks to the Author); expert in itaconate:

The manuscript by Kurmasheva and colleagues highlights the impact of 4-OI to enhance tumor-specific VSV Δ 51 replication and oncolysis in both in vitro and in vivo settings in various model systems. The study suggests that 4-OI inhibits the antiviral immunity of cancer cells via targeting MAVS, JAK1, and NF- κ B pathways.

The study is well conducted and has an impact on the cancer and immunity community. The authors used “itaconate” in the title, but almost all the data were generated with 4OI treatment strategies. Itaconate and 4OI are distinct metabolites with distinct impacts on metabolism and cell function. Since the presented story is focused on 4OI the biological link to itaconate is missing. Additional data is needed to link the observation to itaconate.

Response: We are delighted that Reviewer 2 found our study “well conducted” and with a possible “impact on the cancer and immunity community”.

1. The manuscript described the impact of 4OI on cell metabolism. Since itaconate and 4OI have distinct impacts on cell metabolism and function discussions on itaconate should be limited. The title and the discussion part should be adjusted accordingly. If the authors want to discuss itaconate, they may want to include additional data and repeat some of their studies with itaconate treatments as well as IRG1 KO models.

Response: As Reviewer 1 and 2 suggested, a refocus of the entire study towards 4-OI has been applied. In line, the title and text have been adjusted accordingly. We only preserved two important pieces of data comparing itaconate and 4-OI in the newly submitted version of the manuscript showing their discrepancy of action. These data demonstrate that 4-OI and itaconate act distinctly in terms of cellular metabolism induction (Fig. S1) and biological activity on VSV Δ 51 in 786-O cells (Fig. 1b), thus justifying a focus on 4-OI. Further studies will be needed to understand the possible impact of itaconate/IRG1 on oncolytic virotherapy.

2. Previous studies reported that 4OI may inhibit virus replication through various mechanisms. This observation agrees with Fig. 5f depicting that 4OI does not promote wtVSV replication (and may have anti-viral effects). This discrepancy could be explained in more detail.

Response: Indeed, our previous work (Olagnier et al, 2020) demonstrated some antiviral action of 4-OI against a broad range of viruses including DNA and RNA viruses. This observation agrees with Fig. 5f depicting that 4-OI also has an antiviral action against WT VSV. However, 4-OI is promoting VSV Δ 51 infection in cancer cells which is rather counterintuitive. A section was already dedicated to this observation in the original version of the manuscript.

Does this observation phenocopies to itaconate treatments?

Our manuscript is now focused towards 4-OI and not itaconate and we decided not to include this piece of data in the main or supplementary figures of the manuscript. However, we performed the comparative experiment suggested by reviewer 2, and appended the data below. While itaconate did not have a significant proviral impact on VSV Δ 51 in cancer cells, it led to some antiviral action against WT VSV in the same cells.

4. Work by Chen et al. 2023 indicates that endogenous itaconate influences TET2 activities to dampen inflammatory responses involving the NF- κ B pathway. Does 4-OI targets IKK β and promotes viral infection through NF- κ B by altering TET2?

Response: We fully agree with Reviewer 2 that TET2 should have been considered as a potential mechanism of action through which 4-OI would drive its NRF2-independent proviral action. We have now completed a series of experiment using CRISPR/Cas9 gene editing of TET2 and show that TET2 is not responsible for the biological effects driven by 4-OI in our cellular system. These new data are displayed in Fig. S15e,f.

5. The impact on 4OI, itaconate, and other derivatives are different in Fig. 1b compared to Fig. 1c. Are the changes induced due to species differences (mouse cells CT26WT versus human 786-O cells)? Some discussions might be helpful to understand the differences.

Response: As part of the refocus of our manuscript on 4-OI, this panel has now been removed from the revised version of the manuscript.

6. Since IRG1 expression levels are upregulated in the tumor microenvironment, checking if metabolites (e.g. itaconate and succinate) are changed during 4-OI treatment and VSV Δ 51M infection would be informative.

Response: This is an excellent point, and we have now included metabolite measurements (itaconate, octyl-itaconate and succinate) by LC/MS from 786-O cells stimulated with itaconate or 4-OI and infected with VSV Δ 51. The new data are displayed in Fig. S1.

Reviewer #3 (Remarks to the Author); expert in oncolytic virus:

Kurmasheva et al. describe the krebs cycle-derived metabolite itaconate and derivatives effects on oncolytic therapy with VSV Δ M51. The authors present a large amount of data that details the in vitro mechanism by which the metabolites interfere with the type I IFN and NF- κ B-mediated antiviral responses. Overall, the data is well presented, and conclusions are clear. The manuscript could just be significantly strengthened with more supporting in vivo data. And although the authors show nice mechanistic data using in vitro cancer cell lines and

3D organoids they data confirming these *in vivo*. The clinical application of 4-OI and VSVΔM51 would be significantly strengthened by some additional data outlined below.

Response: We are delighted that Reviewer 3 found our “large amount of data” to be “well presented with clear conclusions”

Major

- Can 4-OI be delivered orally? What does this do to efficacy? Maybe continuous in water?

Response: We haven't tried to give 4-OI orally as the treatment worked quite nicely intratumorally. We performed qPCR measurements of HO-1 gene induction, an NRF2-induced gene and showed a potent induction within tumors following administration of the molecule (Fig. 5c). In previous studies, 4-OI was administered intraperitoneally (Mills et al, 2018). Here, demonstrating some activity following intratumoral administration in this cancer model is already promising. That being said, the point from Reviewer 3 is extremely valid, and further studies will be needed to compare the activity of intratumoral vs oral administration of 4-OI in combination with VSVΔ51.

- Does 4-OI treatment enhance survival in animals?

Response: We repeated a series of new *in vivo* experiment where the treatment protocol was slightly altered. We reduced the number of VSVΔ51 challenge from 3 to 2. We next evaluated the therapeutic potential of combining 4-OI with VSVΔ51 *in vivo*. Mice given the combination therapy were successful in controlling tumor burden as tumor volumes were significantly smaller compared with either monotherapy (Fig. 2i). The combined treatment also significantly prolonged survival of animals as compared with all other treatment conditions (Fig. 2j). Notably, the combination therapy produced complete remission in 87.5 % of the mice (Fig. 2j). In addition, the cured CT26WT-bearing mice that had received the combination regimen subsequently became immune to rechallenge with the same cancer cells (Fig. 2k-l).

- Antiviral immunity only assessed *in vitro* in cancer cells. What about the role of infection of lymphoid organs *in vivo*? And other non-cancer cells? Would be insightful to see serum cytokine analysis and RNA seq from tumors. Do these data agree with *in vitro* findings?

Response: To address this comment from Reviewer 3, we performed a large *in vivo* immunophenotyping experiment where various parameters were assessed at different time points following 4-OI/virus treatment including immune gene expression within the tumor and immune cell (lymphoid vs myeloid cells) infiltration/activation within the tumor, spleen and tumor-draining lymph nodes by flow cytometry (5 days after the last virus injection) (Fig. 6 and Fig. S10-S11). Overall, our data indicate no negative influence of 4-OI on infiltrating lymphocytes nor systemic manipulation of the main T-cell and myeloid populations especially in this model that already has a significant baseline T-cell infiltration. The dampening of antiviral and inflammatory responses within the tumor after combinational treatment might primarily benefit the oncolytic properties of VSVΔ51. The *in vivo* data align with *in vitro* findings where a trends towards a dampening of antiviral and inflammatory markers at the gene level is also observed intratumorally *in vivo* at early days post infection/treatment.

Minor

- Show actual p values

Response: Considering the large number of main and supplementary figures, we felt that it would fill the figures too much to display the actual p values for each significant comparison. We have a dedicated section included in the material and methods and the p values are also mentioned in the figure legends. However, we decided to display the actual p values in the figure legends for the in vivo data in Fig. 2.

- Making clear the difference between infection (entering cells) and replication (new progeny)

Response: We checked throughout the revised version of the text that no confusion can be made between viral infection and viral replication. In our assays, the term viral infection is usually reserved for data coming from flow cytometry measurements when the term viral replication is devoted for data coming from plaque assay measurements.

Specific comments:

- Line 114: “highly sensitive to interferon (IFN)” Not the case for all OV_s (Vaccinia Virus for example). Maybe discuss in the context of VSV?

Response: The portion of the text related to the “highly sensitivity to IFN” has been removed from the revised version of the text.

- Line 118: “Because significant replication of OV_s in tumor tissues is necessary for the highest efficacy” Is this true? References to support? Particularly with VSV in immunocompetent models? Or is stimulating an anti-tumor immune response (with or without direct tumor infection) more important?

Response: We agree with the reviewer that some viruses, especially oHSV, do not need to highly replicate within the tumor to trigger strong immunotherapeutic efficacy. The portion of the text related to this statement has been removed from the revised version of the text.

- Line 145: Sentence is unclear.

Response: The sentence has been edited.

- Line 191: Used the work “synergizing” I would remove unless evidence to show.

Response: The word synergizing has been removed and the sentence edited.

- Line 225: “intratumorally injected with 4-OI for 24 hours” What is meant here? Continuous for 24hr? Single injection 24h prior? Please clarify.

Response: The sentence has been edited.

- Line 242: I would like more rationale on benefits of “pathologically relevant 3D tumoroid models” over in vivo models?

Response: The use of human 3D tumor models in cancer research offers several advantages over traditional murine in vivo models. Here are some of the rationale and benefits.

1. **Relevance to human biology** - Human 3D tumor models better mimic the complexity and heterogeneity of human tumors compared to murine models. This is crucial in understanding the mechanisms of cancer development, progression, and response to treatment in a context that closely resembles human biology.

2. **Patient-specific characteristics** - Human 3D tumor models can be derived from patient samples, allowing for the development of patient-specific models. This personalized approach takes into account individual variations in genetic makeup, tumor microenvironment, and response to treatment.
3. **Therapeutic testing and development** - Testing therapies in human 3D tumor models provides more relevant information about their efficacy and toxicity in a human context. This can improve the success rate of drug development by filtering out candidates that may have shown promise in murine models but fail to translate well to humans.
4. **Reduction of animal use and ethical considerations** - Using human 3D tumor models reduces the need for large numbers of animals in research. This is in line with ethical considerations and the principles of the 3Rs (Replacement, Reduction, Refinement) in animal research. It helps minimize the ethical concerns associated with the use of animals in experiments.
5. **Cost and time efficiency** - Human 3D tumor models can be more cost-effective and time-efficient compared to murine models. The shorter generation time and reduced complexity of working with *in vitro* models can accelerate research progress and drug discovery.

While human 3D tumor models offer significant advantages, it's worth noting that they are not without limitations. Researchers often use a combination of *in vitro* models, animal models, and 3D patient-derived tumoroid models to gain a comprehensive understanding of cancer biology and develop effective therapies. This is the strategy that we have decided to use in the study. A sentence has been added to the text to highlight the rationale for using 3D tumoroid models.

- Line 245: Remove “significantly” since no stats are presented in Figure 3a.
Response: The word significantly has been removed

Figure 1b and c – could enhanced infection also be due to increases entry (LDLR) receptor expression? This should be examined as a potential mechanism as well.

Response: We fully agree with Reviewer 3 that the increased surface levels of LDLR should have been considered as a potential mechanism of action through which 4-OI could have been driving its NRF2-independent proviral action. We have now completed a series of experiment measuring the surface protein levels of LDLR in 4-OI-treated 786-O cells. We also used a CRISPR/Cas9 gene editing approach to KD LDLR and show that: 1- LDLR surface levels are not varying in response to 4-OI treatment in 786-O cells and 2- Knocking-Down LDLR did not affect the capacity of 4-OI to promote VSVd51 infection in 786-O cells. Altogether this justifies that LDLR is not a target of 4-OI in this specific setup. These new data are displayed in Fig. S8.

Figure 1f – I would like to see all cell lines treated with the same concentration of 4-OI and infected with same MOI of virus. Might be best to move different treatments to supplemental so a direct comparison can be made across all cell lines.

Response: The cell lines used in the study have a different sensitivity profile to the virus, some being quite resistant while some others are more permissive, thus justifying the use of different MOI. Here, we included more cell lines to our original panel (H4, DK-MG, 76-9) and

decided to create different graphs based on the sensitivity to VSV Δ 51 of the different groups of cell lines studied (Fig. 1e-g).

Figure 1g – should have separate letter for plaque area figure to make clearer.

Response: The two panels have now a separate letter.

Figure 1h – provide rational for calcein green staining in text.

Response: Calcein is a widely used green, fluorescent viability cell marker. Calcein is membrane-permeant and thus can be introduced into cells via incubation. Once inside the cells, calcein is hydrolyzed by endogenous esterase into the highly negatively charged green fluorescent calcein, which is retained in the cytoplasm. Only viable cells can be stained with this fluorescent dye. A sentence has been added to the methodology section to justify the use of calcein as a viability tracking dye.

Why is 76-9 cell line not included in figure 1 if used as one of the main animal models? I would like to see how infection, replication, and cytotoxicity compares to CT26.

Response: We have now new data showing the comparison of VSV Δ 51 infection in presence or not of 4-OI in CT26 and 76-9 cells. The data are now displayed in Fig. 1f and Fig. 2a.

Figure 2b – images of 76-9 tumors?

Response: The images of 76-9 tumor cores have been added as Fig. 2c.

Figure 2c – titers from 76-9 tumors?

Response: The titers in Fig. 2f are the ones from the pictures now displayed in Fig. 2c.

Figure 2c-e – scales should be kept the same for easier comparison between models.

Response: We decided to leave the scale as they were since we are not trying to compare both models here but simply to demonstrate that 4-OI can increase the replication of VSV Δ 51 in tumor cores derived either from CT26WT or 76-9 tumors.

Figure 2f – Should include scale bar.

Response: Scale bar is now included

Figure 2j – Would like to see survival data.

Response: Survival data and survival data from cured and re-challenged animals are now included in the revised version of the manuscript. The data are displayed in Fig.2.

Figure 3c – figure legend needs further details. What is TBP in figure?

Response: TBP stands for TATA box-Binding Protein. The gene is here used as an invariant housekeeping gene to normalize the VSV L gene data by qPCR. This is clarified in the text.

Figure 3e-g – Figure legend needs significantly more info.

Response: A section entitled “GFP and RFP detection within tumor colon organoids” was already integrated in the original version of the manuscript. We have added a sentence in the figure legends linking to this specific section in the material and methods.

Figure 4c – this is difficult to interpret. Why not present like in Figure 1e?

Response: We have now changed the graphical representation of this figure so that it looks the same as Fig. 1e

Reviewer #4 (Remarks to the Author); expert in innate immunity:

In this manuscript, Kurmashewa et al. reported the enhancing effect of 4-octyl itaconate (4-OI) on the oncolytic virotherapy using VSV Δ 51M. Mechanistically, the authors propose that 4-OI targets multiple innate immune signaling pathways including the RIG-I-MAVS, JAK1-STAT1, and NEMO-IKK β -NF- κ B. Overall, the authors have performed a comprehensive evaluation demonstrating the efficacy of 4-OI in OV_s against cancer cell lines, engrafted tumors, patient-derived tumoroids, and brain tumor slices. On the other hand, the mechanisms underlying 4-OI's effect on viral replication and host innate immunity need more supporting evidences. My comments are as follows:

Response: We thank the reviewer for his/her constructive evaluation of our manuscript.

(1) The authors showed that 4-OI treatment only marginally promoted infection of VSV Δ 51M in normal cell lines (Figure S1). This seems to suggest that innate immunity is not the major factor determining the selective role of Δ 51M on cancer cells over normal cells, or the anti-IFN/NF- κ B roles of 4-OI as proposed by the authors do not work in normal cells. To address this question, the authors should check the IFN-beta production (e.g., by RT-PCR or ISG expression) upon VSV Δ 51M infection in normal cell lines (HUVEC, Fibroblasts), with or without the treatment of 4-OI.

Response: We completed a series of experiments where the levels of different immune markers and antiviral proteins were assessed by immunoblotting in primary HUVECs from different donors upon VSV Δ 51 infection, with and without 4-OI. At the dosage of 4-OI used, only a moderate dampening of antiviral markers could be observed in primary HUVECs, thus explaining the limited VSV Δ 51 replication in these cells (Fig. S9c). As to why 4-OI suppresses antiviral immunity more firmly in cancer cells, there are several aspects to take into consideration such as the penetrance of the molecule in cancer vs normal cells, the binding affinity of the different targets to 4-OI (MAVS, JAK1, IKK β , etc...) in normal/cancer cells, and the magnitude of the immune response to VSV Δ 51 to overcome by the drug in normal/cancer cells. These are different parameters that could influence the final virus replication outcome observed in normal/cancer cells. This is further discussed in our revised manuscript.

(2) It is surprising that DMF treatment in normal cells drastically promoted the infection of VSV (nearly 100%, as shown in Fig S1c). This implicates that DMF may have other mechanisms (not NF- κ B) which promote VSV replication in normal cells. Could the author provide comments on this?

Response: We performed the same experiment as suggested in point (1) from Reviewer 4. Here, DMF at the dosage used had a greater capacity than 4-OI to suppress the induction of antiviral proteins such as ISG15 and IFIT1 upon VSV Δ 51 infection in primary HUVECs. The data are now displayed in Fig. S9d.

(3) Figure S2, as 4-OI targets multiple downstream innate immune signaling molecules that are shared by distinct sensing pathways, it is confusing to me why 4-OI restricted or had no effect on multiple types of viruses and oncolytic viruses, including WT-VSV, in the 786-O cancer cell line. This is suggesting that 4-OI has a unique role for VSV Δ 51 that cannot be explained by its anti-interferon or anti-inflammatory role.

Response: Our previous work (Olagnier et al, 2020) demonstrated some antiviral action of 4-OI against a broad range of viruses including DNA and RNA viruses. This observation agrees with Fig. 5f depicting that 4-OI also has an antiviral action against WT VSV. The highly IFN-sensitive nature of VSV Δ 51 may be one of the main reasons why 4-OI has such a positive effect on the virus, an effect that is not further promoted for the WT VSV. It is also possible that 4-OI has direct alkylating effect on other viruses, limiting their replication. This is currently one line of research we are pursuing.

Again, I suggest the author to monitor the IFN and NF- κ B signaling activation in 4-OI treated cells during Sindbis virus, Reovirus, Vaccinia virus and Measles virus infections.

Response: The gene levels of different antiviral and inflammatory markers upon oncolytic Measles virus, vaccinia virus and Reovirus infection, with or without 4-OI is now reported in Fig. S9a,b.

(4) Typo in the title, Figure S2c, 'Vaccinia virus'.

Response: This typo has been corrected in the revised version of Fig. S2c now S3c.

(5) Although the authors have identified C283 MAVS in the cysteine profiling, the competition ratio remained low (1.18) when compared to the sites of IKK β . To confirm that C283 is the major site of MAVS, the author should perform a 4-OI-alk binding assay with a MAVS C283 mutant (e.g. C283A).

Response: We have performed the experiment suggested by the reviewer 4. The data are now included in Fig. 7h. Using luciferase reporter assays, we also demonstrated the role of the reported cysteine residues targeted by 4-OI in MAVS and IKK β in the regulation of the antiviral and inflammatory signaling pathways. The data are displayed in Fig. 7i and 8l.

(6) Figure 6e, the IPed MAVS for lane 5 and lane 6 needs to be balanced to justify the decrease of binding between MAVS and TBK1 upon 4-OI treatment.

Response: A quantification has been applied and is now reported in the result section of the manuscript.

To further validate the specificity of 4-OI on C283, the author should also check the binding of MAVS-C283A to TBK1 with or without the treatment of 4-OI.

Response: We performed and quantified the experiment suggested by the reviewer 4. The data are now included in Fig. 7j.

(7) Figure 7b and S9, it looks like P65 mostly located in the nucleus in cells that were not infected with VSV (Row 3 of S9). The author should repeat the staining to make sure that the cells were in resting state before VSV infection.

Response: There must have been some confusion by the reviewer here as the cells really were in a resting (non-activated) state. We appended the different pictures from the original experiment showing the different channels as Fig. S14a. Additionally, we quantified the

presence of p65 in the nucleus for the different conditions. The data are also displayed in Fig. S14b.

(8) To avoid the potential off-target effect of 4-OI-Alk in the binding assay, the author should include a negative control in Figure 7g with a protein that is not known to be modified by 4-OI. For example, the closely related kinase IKK α may serve as a good control.

Response: This is an extremely valid point. We have included IKKgamma and IKKeppilon as non-targeted control proteins in our assay. The data are displayed in Fig. 8f.

REVIEWERS' COMMENTS

Reviewer #1 (Remarks to the Author):

the authors have responded to all the comments. Congratulations on a very thorough and detailed revision work.

Reviewer #2 (Remarks to the Author):

The manuscript has been reoriented to center around 4-OI, prompting adjustments to the title. The majority of the data related to itaconate has been omitted, as it presents distinct effects compared to 4OI. Given the divergence between 4OI and itaconate in terms of their metabolic and functional impact, discussions about itaconate should be limited in the manuscript. Specifically, the introduction is misleading as it does not pertain to the focus of this study. Additionally, itaconate should be removed from both the keyword list and the short title. Discussions on itaconate hold lesser relevance to the scope of this study and should be restricted accordingly.

Reviewer #3 (Remarks to the Author):

Thank you for the detailed responses to all reviewer comments. I am satisfied with the revised manuscript and recommend it for publication.

Reviewer #4 (Remarks to the Author):

The authors have adequately addressed most of my concerns with additional experiments and explanations. My minor comments were listed as follows:

- (1) Please provide the line numbers for the discussions on the differences between cancer and normal cells in response to 4-OI.
- (2) Figure 7J, it seemed that the overall IPed MAVS was quite low compared to the bands presented in the input. Moreover, the IPed MAVS showed significantly lower signals in the 4-OI treated groups for both WT and C283A, which largely determined the quantification of the interaction. Could the author comment on this? Does TBK-1 promote the stability of MAVS upon binding?

RESPONSE TO REVIEWERS' COMMENTS

Reviewer #1 (Remarks to the Author):

The authors have responded to all the comments. Congratulations on a very thorough and detailed revision work.

Reviewer #2 (Remarks to the Author):

The manuscript has been reoriented to center around 4-OI, prompting adjustments to the title. The majority of the data related to itaconate has been omitted, as it presents distinct effects compared to 4-OI. Given the divergence between 4-OI and itaconate in terms of their metabolic and functional impact, discussions about itaconate should be limited in the manuscript. Specifically, the introduction is misleading as it does not pertain to the focus of this study. Additionally, itaconate should be removed from both the keyword list and the short title. Discussions on itaconate hold lesser relevance to the scope of this study and should be restricted accordingly.

Response: Although we fully agree with Reviewer 2 on the fact that 4-OI and itaconate are quite divergent in terms of their metabolic and functional impact, we tend to disagree on the point that the introduction is misleading and does not pertain to the focus of the study. Indeed, we feel that our introduction in its current stand properly introduces 4-octyl-itaconate. As 4-OI, is a molecule derivatized from the endogenous metabolite itaconate, we felt legit to also introduce itaconate itself. Additionally, most of the text and references in the section going from lines 134-157 have been specifically requested by the Reviewer 1 who is now satisfied with the edits/additions to the text. In agreement with Reviewer 2, we removed the word itaconate in the keyword list and replaced itaconate derivative by 4-OI in the short title. The discussion has been modified and sections on itaconate removed to reflect the scope of the study better on itaconate derivatives.

Reviewer #3 (Remarks to the Author):

Thank you for the detailed responses to all reviewer comments. I am satisfied with the revised manuscript and recommend it for publication.

Reviewer #4 (Remarks to the Author):

The authors have adequately addressed most of my concerns with additional experiments and explanations.

My minor comments were listed as follows:

(1) Please provide the line numbers for the discussions on the differences between cancer and normal cells in response to 4-OI.

Response: It is a mistake of ours that a discussion section was not included in the revised version of our manuscript. We have now added this section to the finalized version of the text (lines 664-674).

(2) Figure 7J, it seemed that the overall IPed MAVS was quite low compared to the bands presented in the input. Moreover, the IPed MAVS showed significantly lower signals in the 4-OI treated groups for both WT and C283A, which largely determined the quantification of the interaction. Could the author comment on this? Does TBK-1 promote the stability of MAVS upon binding?

Response: The overall low immunoprecipitated (IPed) levels of MAVS can be attributed to both a limited amount of whole cell extracts utilized in the immunoprecipitation process and low protein loads. Notably, the IPed MAVS displayed significantly reduced signals in the group treated with 4-OI, both in wild-type (WT) and C283A variants. This suggests that the binding of 4-OI might interact with additional target sites on MAVS, inducing a conformational change that decreases the exposure of the 5' flag-tag to the antibody.

MAVS undergoes phosphorylation by the kinase TBK1, and this phosphorylation can influence MAVS stability. However, we did not observe a significant increase in MAVS levels upon TBK1 treatment in the original film. In Fig. 7J, it appears that the increase in MAVS levels in the input sample is primarily due to overexposure and size reduction. TBK1 triggers phosphorylation of MAVS, and in the input sample, the phosphorylated and unphosphorylated bands are not well separated, giving the impression that TBK1 increases MAVS levels.